# Assessing spatio-temporal variability of melt–refreeze patterns in firn over Greenland with CryoSat-2

Weiran Li [1], Stef Lhermitte [2, 1], Bert Wouters [1], Cornelis Slobbe [1], Max Brils [3, 5], and Xavier Fettweis [4]

[1]Department of Geoscience and Remote Sensing, Delft University of Technology, Delft, the Netherlands
[2]Department of Earth & Environmental Sciences, KU Leuven, Leuven, Belgium
[3]Institute for Marine and Atmospheric Research, Utrecht University, Utrecht, the Netherlands
[4]Laboratory of Climatology, Department of Geography, SPHERES research unit, University of Liège, Liège, Belgium
[5]Geography and Environmental Sciences Department, Northumbria University, Newcastle upon Tyne, UK

**Correspondence:** Weiran Li (w.li-7@tudelft.nl)

**Abstract.** In recent decades, satellite radar altimetry has been widely used to assess volume changes over the Greenland Ice Sheet. Especially, melt events result in drastic changes in volume scattering of firn, which induces a pronounced change in parameters derived from radar altimeter data. Due to the recent and increasingly frequent melt events over Greenland, the impacts of these events on the firn condition i.e. formation of ice lenses and reduction of firn air content, need to be better understood. This study therefore exploits the ability of long-term CryoSat-2 data in indicating changes in firn volume scattering, in order to assess the spatio-temporal firn condition variations in Greenland. More specifically, this study utilises the leading edge width (LeW) parameter derived from CryoSat-2 Low Resolution Mode data, which has been proven to be a parameter strongly sensitive to changes in volume scattering, and assesses its variation between September 2010 and September 2024. With a combined analysis of remote sensing observations, in situ observations and outputs from regional climate models, our study demonstrates that the LeW drop induced by extreme melt events in the interior of Greenland experiences a gradual recovery, which can potentially be explained by new snow deposition. However, in many high-elevation regions of Greenland where firn layers were originally dry, the recent recurrence of extensive melt has prevented a full recovery of firn volume scattering to pre-2012 conditions, indicating a persistent increase in firn density under a changing climate. Finally, our study also confirms the utility of radar altimeter data for long-term monitoring of the impact of melt and refreezing events on the properties of the upper firn layer.

## 1 Introduction

Over the recent decades, the Greenland Ice Sheet has experienced a notable increase in the frequency and intensity of melt events (Tedesco et al., 2011, 2013; Nilsson et al., 2015; Tedesco et al., 2016; Tedesco and Fettweis, 2020). These events are particularly prevalent in low-elevation regions, where they contribute to runoff towards the ocean. This runoff negatively impacts the surface mass balance of the ice sheet and may lead to irreversible ice loss and sea-level rise (Lenton et al., 2008; Sasgen et al., 2012). In contrast, at higher elevations, meltwater can infiltrate and refreeze within the porous firn (Harper et al., 2012). This refreezing process releases latent heat, which accelerates firn compaction and diminishes the firn's capacity to store

additional meltwater, consequently speeding up runoff from the ice sheet's interior (van den Broeke et al., 2016; Machguth et al., 2016; Vandecrux et al., 2019). Surface meltwater also drains towards the bedrock through crevasses and moulins, altering basal frictions and ice velocities (Zwally et al., 2002; Sundal et al., 2011; Meierbachtol et al., 2013). Furthermore, studies suggest that the runoff and melt will continue to increase (Vizcaíno et al., 2009; Huybrechts et al., 2011), underscoring the importance of assessing the impact of melt and refreezing on Greenland's firn layer for understanding the ice sheet's overall stability and response to climate change (Heilig et al., 2018).

Given the limited spatial and temporal coverage of in situ data (Hall et al., 2008; Koenig et al., 2016; Castelao and Medeiros, 2022), remote sensing techniques are indispensable for assessing this impact. Radar altimetry is one of them, although so far it has mainly been used to measure surface elevation changes (e.g., Helm et al., 2014; Slater et al., 2018). The underlying principle is based on the fact that over firn-covered regions of the ice sheet—primarily at higher elevations—radar pulses at frequencies commonly used in satellite altimetry penetrate into the firn (e.g., Ridley and Partington, 1988). According to Ridley and Partington (1988) and Davis and Zwally (1993), the penetration depth may range from a few centimetres to several metres, depending on the firn status (e.g. dry, wet, refrozen; Slater et al., 2019) and the retracker (Michel et al., 2014; Simonsen and Sørensen, 2017; Li et al., 2022). Consequently, recorded waveforms contain signals from both surface scattering and volume (or subsurface) scattering caused by inhomogeneities within the underlying firn layers. As explained by Ridley and Partington (1988), surface scattering dominates the start of the waveform, while volume scattering becomes predominant beyond the point at which the illuminated surface area becomes constant (regarding the latter, see (Chelton et al., 2001, Sect. 2.4.1)). The rise of the backscattered power from volume scattering depends on firn parameters, including firn density, firn air content (FAC), and grain size (Ridley and Partington, 1988; Vandecrux et al., 2019; Brils et al., 2022). For example, larger grain radii and higher firn densities lead to a faster increase in backscattered power (i.e., a steeper leading edge) (Ridley and Partington, 1988, Figs. 9, 10). Melt and refreezing events alter firn parameters and form refrozen layers, modifying scattering behaviour and waveform shape. Depending on thickness and density, refrozen layers can substantially reduce radar penetration, diminishing volume scattering (Nilsson et al., 2015; Otosaka et al., 2020). While these changes in waveform shape are observable, attributing them to variations in volume scattering—and thus to changes in firn properties—requires distinguishing them from variations in surface scattering, particularly those driven by surface roughness. Indeed, a decrease in surface roughness also results in steeper leading edges (Ridley and Partington, 1988, Fig. 8).

Several studies have employed waveform shape parameters to gain insight into the impact of melt and refreezing on Greenland's firn layer, as well as to estimate the bias in radar altimeter-derived elevations caused by radar penetration into the firn. Nilsson et al. (2015), for example, used Ku-band (13.575 GHz) altimeter data acquired by CryoSat-2 to track the formation of ice lenses following melt events. Simonsen and Sørensen (2017) explored the same data to investigate the impact of volume scattering properties on elevation estimates. Both Nilsson et al. (2015) and Simonsen and Sørensen (2017) showed that a large leading edge width (LeW) is indicative of volume scattering of the signal in the upper parts of the firn, while Nilsson et al. (2015) in particular observed the impact of the 2012 Greenland melt event and its subsequent refreezing on waveform-derived parameters, including LeW, trailing edge slope (TeS) and peakiness, backscatter intensity, and height. The Simonsen and Sørensen (2017) study indicated that within the region of the Greenland Ice Sheet covered by Low Resolution Mode (LRM)

data (i.e., the LRM zone), LeW could be effectively used to correct for elevation biases caused by volume scattering. In addition to waveform shape parameters, other radar altimeter-derived variables have also been utilised to infer firn properties. For instance, Scanlan et al. (2023) leveraged surface echo powers in Ku-band CryoSat-2 and Ka-band SARAL radar waveforms to derive monthly maps of Greenland Ice Sheet's wavelength-scale surface roughness and density between January 2013 and December 2018. Furthermore, several studies have estimated radar penetration depths by combining radar and laser altimetry data. Michel et al. (2014), for example, analysed the differences between radar (ENVISAT) and laser (ICESat) altimeter heights over Antarctica to derive Ku-band radar penetration biases into firn and compared these height differences with LeWs. The study provides insights into opportunities for similar approaches to study Greenland's firn.

Despite these advances in using radar altimetry to monitor Greenland's firn properties—particularly in assessing the effects of melt and refreezing—existing studies have largely been confined to a period without extensive melt (e.g., January 2013 to December 2018; Scanlan et al., 2023), or to the short time frame immediately following the 2012 melt event (e.g., up to 2014; Nilsson et al., 2015). The impact on the long term, especially following the 2019 melt (Tedesco and Fettweis, 2020), remains insufficiently monitored. The availability of more than a decade of CryoSat-2 data presents a valuable opportunity to address this gap. By assessing how melt–refreezing processes affect the CryoSat-2 LeW, we aim to improve the understanding of the stability of the Greenland Ice Sheet and its response to climate change, and to explore the potential of using radar altimeters as a complementary tool for providing a comprehensive observation of Greenland firn properties.

The main objective of this paper is to assess the impact of melt and refreezing events on the properties of Greenland's upper firn layer, using LeWs derived from CryoSat-2 radar waveforms acquired between 2010 and 2024. To support the interpretation of the results, we complement the assessment by comparing the LeWs with: (i) the surface roughness dataset derived by Scanlan et al. (2023) to analyse under which circumstances the LeW variation is dominated by surface scattering; (ii) the ArcticDEM (Porter et al., 2023) standard deviation to assess the impact of macro-scale roughness due to topographic variation on LeW; (iii) laser-radar height offsets obtained by differencing ICESat-2 laser altimeter elevations and CryoSat-2 radar altimeter elevations to gain further insights into volume scattering; and (iv) firn densities and FAC from the Modèle Atmosphérique Régionale (MAR) (Fettweis et al., 2011, 2017; Lambin et al., 2022) and the Firn Densification Model from the Institute for Marine and Atmospheric research Utrecht (IMAU-FDM v1.2G; Brils et al., 2022) to analyse how spatial and temporal variations in firn properties affect the LeW and to improve the interpretation of radar altimeter scattering properties for future research.

The paper is organised as follows. The details of data and coverage are described in Section 2. The derivation of LeWs and elevation differences between ICESat-2 and CryoSat-2 will be detailed in Section 3. Sections 4 and 5 present, analyse and discuss the results. Finally, our main findings and outlook are presented in Section 6.

## 2   Data

### 2.1   Reference digital elevation model

The ArcticDEM digital elevation model (DEM) is utilized for three purposes: (i) correcting CryoSat-2 elevation estimates for slope-induced errors using the leading edge point-based (LEPTA) correction method (Li et al., 2022), (ii) computing macro-

scale surface roughness resulting from topographic variation, and (iii) segmenting the Greenland Ice Sheet into ten distinct elevation bands for spatio-temporal analysis. These groups include one below 1500 m, eight between 1500 m and 3000 m evenly spaced according to elevation, and one above 3000 m.

ArcticDEM is constructed from recent stereo satellite imagery (Porter et al., 2023) and has a systematic error of less than 5 m (Noh and Howat, 2015). The model is available at various resolutions, ranging from 2 m to 1 km (Porter et al., 2023). Consistent with Li et al. (2022), we employ the 100 m resolution ArcticDEM for slope-induced error correction in CryoSat-2 elevation estimates, balancing accuracy and computational efficiency compared to the higher-resolution 2 m dataset.

For macro-scale roughness estimation and elevation-based segmentation of the Greenland Ice Sheet, we define the roughness metric as the standard deviation of elevations within a 10 km $\times$ 10 km grid. Prior to segmenting the Greenland Ice Sheet in distinct elevation bands, we first average elevations over the same 10 km $\times$ 10 km grid to ensure consistency with the grid used for firn model output.

## 2.2 CryoSat-2 observations

CryoSat-2's primary payload, the SAR Interferometric Radar Altimeter (SIRAL), operates in three measurement modes (Wingham et al., 2006):

1. Low Resolution Mode (LRM): analogous to pulse-limited radar altimetry, this mode is used over relatively flat ice sheet regions and the open ocean.

2. Synthetic Aperture Radar (SAR) Mode: utilising coherent echo processing, this mode provides higher-resolution along-track data from sea ice and sea-ice-contaminated ocean surfaces.

3. SAR Interferometric (SARIn) Mode: combining coherent echo processing with interferometry, SARIn delivers high-resolution along-track data along with across-track echo directions, primarily for ice sheet margins.

This study focuses on LRM data acquired over the interior of the Greenland Ice Sheet, specifically using Baseline E L1b data spanning from September 1, 2010, to September 30, 2024.

Both LeWs and elevations ($h_C$) are estimated using the Offset Centre of Gravity (OCOG) retracker (Wingham et al., 1986; Bamber, 1994; Gommenginger et al., 2010; Rinne and Similä, 2016). The retracker is applied to waveforms that are normalized based on their peak power. The LeW, expressed in metres, is computed as:

$$\text{LeW} = \frac{1}{2}c\Delta t(b_{0.95} - b_{0.05}),$$

(1)

where $b_{0.95}$ and $b_{0.05}$ are the range bins at which the normalized waveform power equals 95% and 5% of the OCOG amplitude ($A_{\text{OCOG}}$), respectively, determined via linear interpolation. Here, $c$ represents the speed of light, and $\Delta t$ denotes the CryoSat-2 LRM waveform sampling interval (3.125 ns). $A_{\text{OCOG}}$ is defined as:

$$A_{\text{OCOG}} = \sqrt{\sum_{n=n_1}^{n_2} y^4(n) / \sum_{n=n_1}^{n_2} y^2(n)},$$

(2)

where $y(n)$ denotes the normalised power at bin $n$. The parameters $n_1$ and $n_2$ define the range of bins included in the computation, which serves to mitigate the impact of noise typically present at the beginning and end of the waveform (Frappart et al., 2021; Ronan et al., 2024). In this study, $n_1$ and $n_2$ are empirically set to 10 and 125, respectively. To ensure reliable LeW estimates, an additional data editing step is applied: waveforms for which the normalized power in the initial part of the waveform exceeds $5\%$ of $A_{OCOG}$ are excluded from LeW computation. In these cases, $b_{0.05}$ cannot be determined reliably. This editing step explains the discrepancy between the total number of LeW and elevation estimates reported below. A diagram showing a typical waveform over the Greenland Ice Sheet and the OCOG processing is shown in Fig. 1.

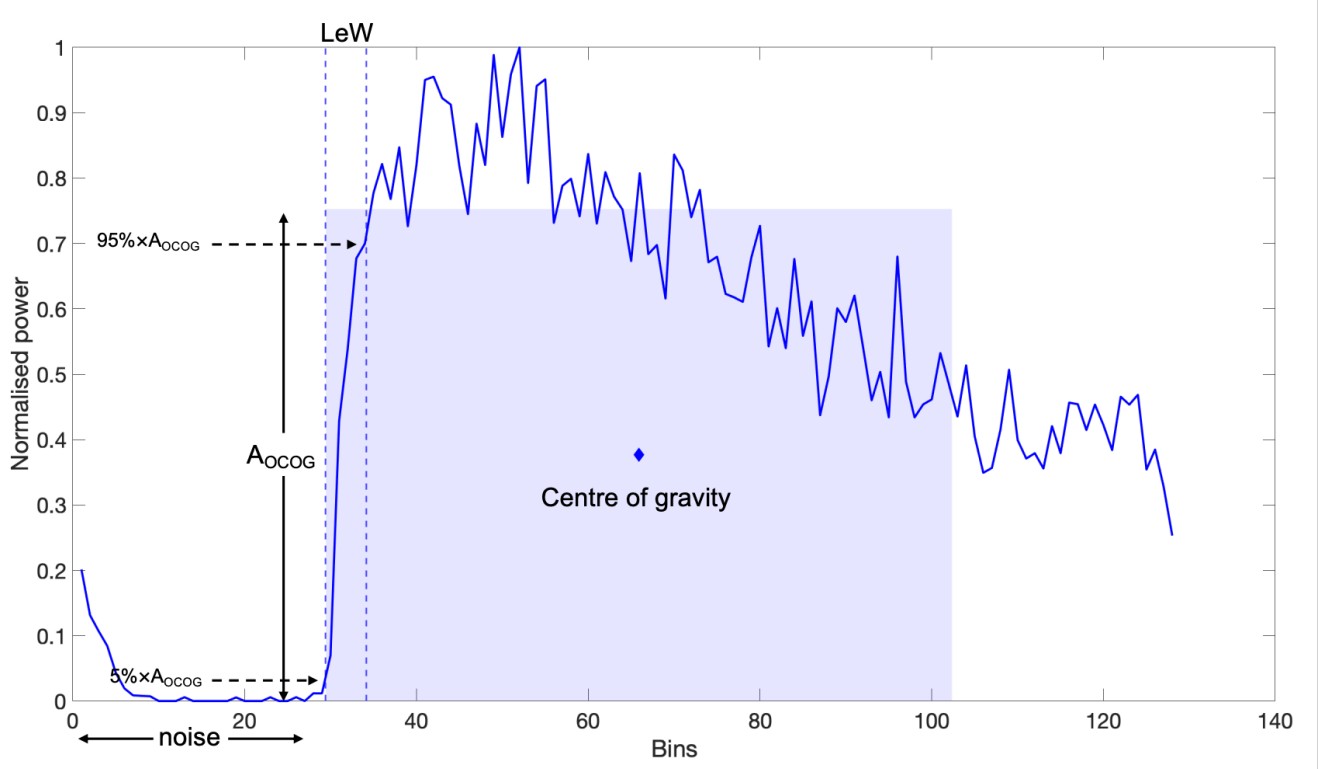

**Figure 1.** A waveform acquired over the Greenland Ice Sheet indicating the OCOG amplitude and LeW.

The computation results in approximately $3.07 \times 10^7$ LeWs over the entire investigated period in total. Monthly mean LeWs are computed on a $10\,\mathrm{km} \times 10\,\mathrm{km}$ grid. The time series of monthly mean LeW serves as the basis for analysing temporal LeW variations along a simple north-south transect (connecting the NEEM site, Summit Camp and the southernmost locations of the LRM coverage) and conducting a Greenland-wide assessment of LeW variations across elevation bands.

A total of approximately $3.13 \times 10^7$ CryoSat-2 elevation estimates ($h_C$) are derived using the OCOG retracker with a $50\%$ threshold. As discussed by Davis (1997), the half-power point best represents the mean surface elevation (i.e., the mean elevation of the firn–air interface) within the altimeter's pulse-limited footprint, assuming surface scattering dominates the waveform. However, in firn-covered areas, $h_C$ corresponds to an elevation within the firn layer. By comparing $h_C$ to ice

sheet elevations from ICESat-2, we obtain laser-radar height offsets as a proxy for the Ku-band radar penetration depth (see Sect. 3.3).

Based on the results of Li et al. (2022), we chose to apply the LEPTA method to correct $h_C$ for slope-induced errors. The study, which introduced and compared LEPTA against slope- and point-based correction methods using CryoSat-2 LRM data over Greenland in 2019, demonstrated that LEPTA provides stable performance across both flat interior regions and areas with more complex topography when compared to ICESat-2 observations.

## 2.3 ICESat-2 observations

The laser altimeter-derived elevations of the firn-air interface, which are used to calculate a proxy for the Ku-band radar penetration depths, are obtained from the ICESat-2 L3A Land Ice Height (ATL06) product, Version 6 (Smith et al., 2023a). ICESat-2 operates the Advanced Topographic Laser Altimeter System (ATLAS) that transmits pulses with a wavelength of 532 nm (Abdalati et al., 2010). The along-track resolution of the ATL06 product is approximately 40 m (Smith et al., 2023b), and its geolocation bias is less than 10 m (National Snow and Ice Data Center (NSIDC), 2021).

In this study, we used ATL06 data from January 1, 2019, to September 30, 2024, to obtain ICESat-2 elevations ($h_{ICE2}$) for each CryoSat-2 point. Following the approach described in Li et al. (2022), we compared CryoSat-2 elevations acquired in a given month to ICESat-2 elevations from the same month. For each CryoSat-2 point, we first identify all ICESat-2 points within a 50 m radius. If ICESat-2 points are available in each quadrant surrounding the CryoSat-2 point, ICESat-2 elevations are interpolated to the CryoSat-2 location using natural-neighbor interpolation ($h_{ICE2}$). Otherwise, nearest-neighbour interpolation is applied. The search for ICESat-2 points acquired within the same month and within a 50 m radius of each CryoSat-2 measurement yields a total of approximately $4.53 \times 10^5$ points.

## 2.4 Firn models

To support the interpretation of altimeter-derived LeWs and laser-radar height offsets, we use firn density, melt water content (MWC), and firn air content (FAC) from two regional climate and firn models: the Modèle Atmosphérique Régional (MAR, Section 2.4.1) and the IMAU Firn Densification Model (IMAU-FDM, Section 2.4.2).

### 2.4.1 Modèle Atmosphérique Régional (MAR)

Layered firn densities and meltwater content (MWC) over the study period are obtained from version 3.14 of the Modèle Atmosphérique Régional (MAR) forced by the ERA5 reanalysis (Fettweis et al., 2017; Lambin et al., 2022; Grailet et al., 2024; Machguth et al., 2024). The MAR outputs have a horizontal resolution of 10 km and a daily temporal resolution, whereas the vertical resolution of the snowpack varies over depth, from 10 cm near the surface to 5 m at the bottom of the snowpack. The MAR model resolves only the upper 20 metres of the snowpack.

We use the time series of modelled firn density profiles to compute the weighted average density of the upper firn column, from the surface to the maximal proxy for Ku-band radar penetration depth. Thickness is used as a weighting factor

to account for the model's uneven vertical resolution. The maximal proxy for Ku-band radar penetration depth is empirically determined by analysing the differences between ICESat-2 laser altimeter elevations and CryoSat-2 radar altimeter elevations (see Sect. 3.3).

The firn density profile provides insights into volume scattering. For example, the presence of a refrozen layer (i.e., a layer with high density; Nilsson et al., 2015; Otosaka et al., 2020) prevents radar signal penetration, thereby reducing volume scattering. The MWC is used to restrict our analysis to non-melt conditions, minimising the impact of meltwater production on Ku-band radar and 532 nm laser measurements. When MWC$> 0$, meltwater is present in the firn layer; thus, the altimeter-derived parameters are primarily influenced by meltwater content rather than firn properties. The MWC value at each CryoSat-2 location is obtained using nearest-neighbour interpolation.

To assess the timing and extent of melt–refreeze patterns, we use the MAR-derived meltwater production and refreezing outputs (expressed in mmWE day$^{-1}$) as a reference. To provide insights into volume scattering variations, we adopt the total snow height change (i.e., the snowfall accumulation) from MAR. The accumulated total snow height change is calculated as the sum of daily changes from the intensive melt in July, 2012 (Nghiem et al., 2012) to September 30, 2024. The time series of these outputs is visualised in Appendix A. For consistency with the CryoSat-2 time series, we compute monthly averages for density, melt, and refreezing from the daily data. Similarly, monthly snow height changes are computed as the sum of daily changes.

### 2.4.2 Firn Densification Model (IMAU-FDM)

Alternatively, firn density, MWC, and FAC with a 10-day temporal resolution are obtained from the firn density model IMAU-FDM v1.2G (Brils et al., 2022). This is a Lagrangian 1D firn model that simulates the evolution of the firn thickness, density, temperature, and water balance. It uses a "bucket method" to compute meltwater percolation into the firn. The model's ability to accurately represent firn properties has been validated in Brils et al. (2022). At its surface, IMAU-FDM is forced by output from the polar version of the Regional Atmospheric Climate Model (RACMO2; Noël et al., 2018). Model results are available from October 1957 to December 2020. As with the MAR data, we compute the weighted average density of the upper firn column, from the surface to the maximal proxy for Ku-band radar penetration depth.

The FAC represents the vertically integrated porosity of the firn (Kuipers Munneke et al., 2015), expressed in metres. It is computed over the entire firn column and serves as a measure of total firn porosity, indicating the firn's capacity to retain meltwater (Vandecrux et al., 2019). Although CryoSat-2 signals primarily penetrate the upper firn layers, we leverage the modelled FAC time series to assess whether the observed melt–refreeze patterns notably influence broader firn conditions.

This dataset serves two main purposes. First, to determine whether firn conditions changed considerably during the CryoSat-2 observation period, we use IMAU-FDM firn density and FAC as a reference dataset for comparison with the monthly LeW time series from January 2011 to December 2020. Second, a time series of yearly averaged IMAU-FDM density and FAC from 1961 to 2020 is used to examine recent changes in the Greenland Ice Sheet in the context of previous decades.

## 2.5 In situ density profiles

In addition to firn model data, we incorporate various in situ density profiles from Vandecrux et al. (2024) at different locations to assess the impact of melt events on firn layers. In situ measurements provide valuable insights into the persistence and distribution of refrozen layers within the Greenland firn, supporting the interpretation of LeW variations.

We include available and published in situ density profiles in our analysis if they meet the following criteria: (i) the acquisition site falls within the CryoSat-2 LRM coverage; (ii) the acquisition time is within the CryoSat-2 operational time, and (iii) the acquisition is vertically continuous rather than a single measurement at a specific depth.

The adopted in situ measurements include profiles collected along a trajectory between NEEM and the East Greenland Ice-core Project (EGRIP) (Schaller et al., 2016a) and along the Expédition Glaciologique Internationale Au Groenland (EGIG) line (Otosaka et al., 2020). Additional data come from the Summit and Saddle stations (MacFerrin et al., 2022), the Greenland Traverse for Accumulation and Climate Studies (GreenTrACS; Lewis et al., 2019), ice cores from central west Greenland (CWG; Trusel et al., 2018), and from Vandecrux et al. (2023). The locations and corresponding densities are shown in Fig. 2. Among the adopted in situ density profiles, the inclusion of the 2012 melt layer is particularly important, as it offers strong evidence that the extreme melt event in that year produced a distinct high-density layer which can also be identified in modelled firn densities (Schaller et al., 2016a; Otosaka et al., 2020; MacFerrin et al., 2022). This layer becomes progressively buried in subsequent years, enabling the observed recovery in LeW. Similarly, recent melt events can appear in the recently acquired in situ density profiles (Vandecrux et al., 2023), as they show similar spikes (approximately 25 % higher than the average density over the top 5 m) as the 2012 melt event.

## 2.6 Wavelength-scale surface roughness data

To assess the extent to which changes in LeW can be attributed to variations in wavelength-scale surface roughness, we use the wavelength-scale surface roughness dataset derived by Scanlan et al. (2023) from Ku-band CryoSat-2 and Ka-band SARAL radar altimeter data. The data are provided as monthly estimates on a 5 km×5 km grid for the period January 2013 to December 2018. Surface roughness estimates are obtained via the Radar Statistical Reconnaissance (RSR) technique (Grima et al., 2012, 2014), which statistically characterises the distribution of surface echo powers extracted from individual radar waveforms. The RSR method fits a homodyned K-distribution to the observed echo power histograms to recover relative, dataset-dependent coherent and incoherent powers, which are converted to absolute powers by calibration. Surface roughness is subsequently inferred through inversion using a backscattering model. Spatially, roughness estimates derived from both SARAL and CryoSat-2 LRM reveal a smooth interior becoming progressively rougher toward the margin. The assessment of the roughness time series at six locations revealed an annual roughness cycle at one site, with a marked increase in surface roughness during summer, likely associated with surface melting (Scanlan et al., 2023). In order to assess the LeW variations with respect to roughness variations, the roughness dataset is averaged to the 10 km×10 km, consistent with the LeW grid.

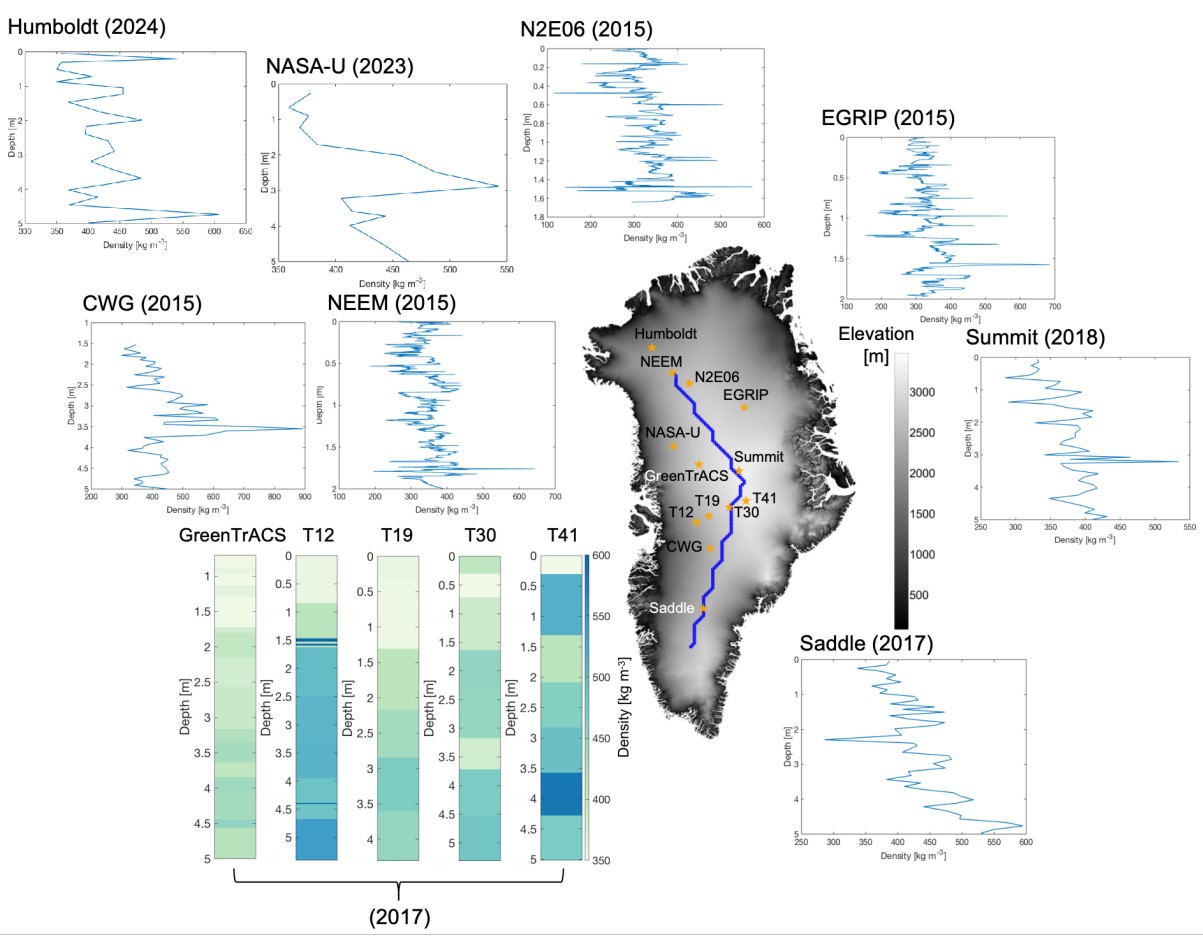

**Figure 2.** In situ density profiles and their measurement locations acquired from Schaller et al. (2016b); Otosaka (2020); Vandecrux et al. (2024). The background is the 1 km×1 km ArcticDEM. Blue rectangles with a resolution of 10 km×10 km represent the geographic transect used to present the results.

## 3  Methods

### 3.1  Assessing LeW variability and recovery after the 2012 melt event

This assessment evaluates the spatio-temporal variability of the changes in LeW, with particular emphasis on the recovery of LeW following the extreme 2012 melt event (Tedesco et al., 2013). To this end, we generate a series of maps showing the changes in average LeW over non-melt seasons (October–April) relative to (i) the non-melt seasons of the previous year, and (ii) the baseline period of October 2010 to April 2011. These non-melt season averages are computed using the time series of monthly mean LeW values on a 10 km×10 km grid described in Sect. 2.2. In addition to the maps, we present changes in monthly mean LeW time series—expressed relative to the mean LeW from October 2010 to April 2011—along

a geographic transect connecting the NEEM site (Nilsson et al., 2015; Schaller et al., 2016a), Summit Camp, and South Greenland (see Fig. 2). Finally, we examine these changes across ten elevation bands, ranging from below 1500 m to above 3000 m (Section 2.1).

## 3.2 Assessing the effects of surface roughness (changes) on LeW (changes)

To assess the extent to which changes in LeW can be attributed to variations in surface roughness, time series of both wavelength-scale and macro-scale surface roughness covering the same period as the CryoSat-2 LRM data (i.e., September 1, 2010, to September 30, 2024) would be required. Unfortunately, such time series are not available. In the case of macro-scale surface roughness, no time series exist. For wavelength-scale surface roughness, the available data are limited to the period from January 2013 to December 2018 (Sect. 2.6). This constrains the scope of our analysis.

To gain insights into the potential influence of wavelength-scale surface roughness on LeW, we compare time series of monthly mean wavelength-scale surface roughness values with corresponding changes in monthly mean LeW. In both cases, anomalies are computed by subtracting the mean over the January 2013 to December 2018 period. As in the analysis of LeW variability and post-2012 melt recovery, the results are presented along the geographic transect and as averages across ten elevation bands.

In addition, we examine the relationship between the absolute values of LeW and surface roughness—both wavelength-scale and macro-scale. These results are also presented along the geographic transect and averaged over the ten elevation bands.

## 3.3 Assessing the correlation between LeW and laser-radar height offsets

This analysis aims to evaluate the extent to which changes in LeW can be attributed to variations in volume scattering. We assess this indirectly by examining the correlation between time series of monthly mean LeW and those of a proxy for Ku-band radar penetration depth. Following the approach of Michel et al. (2014), this proxy is defined as the difference ($\Delta h$) between elevations obtained from the ICESat-2 laser altimeter and the CryoSat-2 radar altimeter. It is computed as:

$$\Delta h = h_{\text{ICE2}} - h_C - (h_{\text{DEMI}} - h_{\text{DEMC}}), \tag{3}$$

where $h_{\text{DEMI}}$ and $h_{\text{DEMC}}$ represent the elevations of the 100 m resolution ArcticDEM at the ICESat-2 and CryoSat-2 measurement locations, respectively. The subtraction of their difference accounts for terrain-induced elevation variations between the two measurement locations.

All negative $\Delta h$ values are excluded from further analysis. As shown in Fig. 3, these values predominantly occur near the margins of the LRM zone, particularly around Jakobshavn Glacier. From the overall probability distribution function, also shown in Fig. 3, we conclude this affects 7% of all values. Negative $\Delta h$ values can be explained in different ways. Among others as the result of measurement errors, errors in data processing (notably in the corrections for slope-induced errors), potential scattering errors from ICESat-2 measurements (Smith et al., 2018; Fair et al., 2024) and the difference in resolution between laser and radar altimetry (ICESat-2 might resolve topographic variability that cannot be resolved from radar altimeter data).

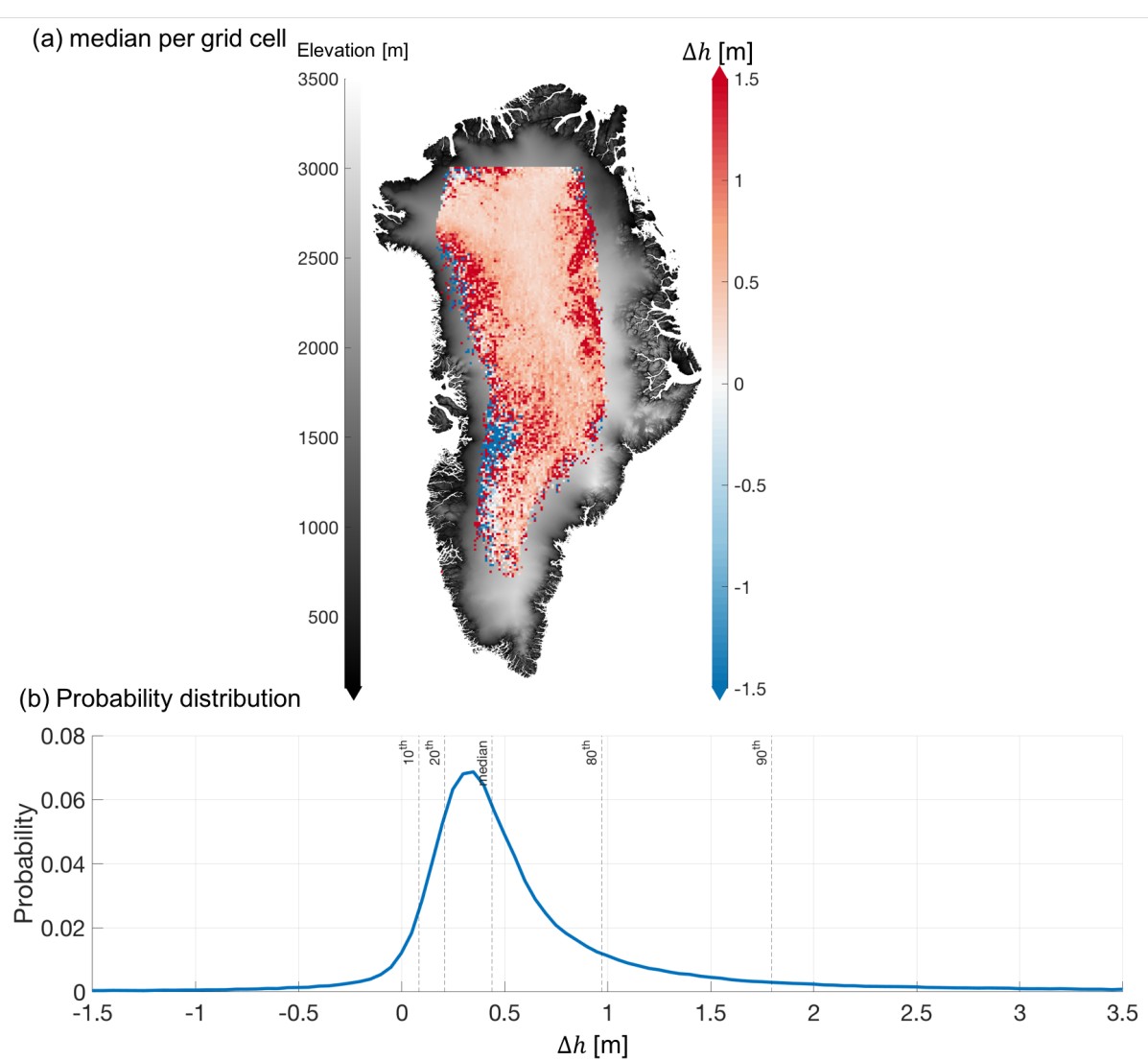

**Figure 3.** Statistics of elevation differences between ICESat-2 and CryoSat-2 ($\Delta h$). (a) Median $\Delta h$ per grid cell at a 10 km×10 km grid; the background shows the 1 km ×1 km ArcticDEM. (b) Probability distribution function of $\Delta h$, with vertical lines indicating selected percentiles.

To assess the correlation between LeW variations and $\Delta h$, we first obtain all the $\Delta h$ and LeW over the entire period within each grid cell of a 10 km×10 km grid. Correlations are quantified using the Pearson correlation coefficient. Statistical significance is evaluated via $p$-values, with correlations considered significant when $p \leq 0.05$ (Bermudez-Edo et al., 2018).

### 3.4 Comparison with modelled firn properties

The comparison with modelled firn properties aims to examine (i) the capability of LeW to indicate firn processes and (ii)
whether these processes are recent, providing insights for future studies. In doing so, we compare LeW variations against variations in modelled firn properties over two different time periods. The first comparison is conducted over the period for which CryoSat-2 Baseline E data are available (between 2010 and 2024). The second comparison covers the longer period when IMAU-FDM data are available (between 1960 and 2020). To ensure consistency with the LeW variability in Sect. 3.1, for the former analysis (using data between 2010 and 2024), we present the monthly mean density and FAC time series relative
to the mean density and FAC from October 2010 to April 2011. This comparison is also performed over the north–south transect and across ten elevation bands, introduced in Sect. 3.1. For the analysis over the longer period, we present the yearly mean density (including summer months) and FAC relative to the density and FAC from 2011.

In addition, for the analysis where we compare and interpret the spatio-temporal variations with firn properties, we need to select a maximal proxy for penetration depth that corresponds to the vertical resolution of the available firn models (i.e., MAR
and IMAU-FDM) to ensure a fair comparison. Figure 3 shows that $\Delta h$ is generally between 0 and 1.5 m on the interior of the ice sheet, therefore this range is used to select the max $\Delta h$ that can be used as an indicator of volume scattering. Due to the uneven vertical resolution of MAR, the weighted average density of the upper 1.5 m is calculated using the thickness per layer as the weighting factor.

## 4   Results

### 4.1   LeW variability and recovery after the 2012 melt event

Figure 4a presents the changes in non-melt season (October–April) average LeWs between successive seasons. It shows a notable reduction (greater than 2 m) in LeW over the interior of the ice sheet between the non-melt seasons of 2011 and 2012, followed by increases between 2013 and 2015, with an increase of approximately 0.5 m per year. This initial reduction indicates a decrease in volume scattering, corresponding to the extreme melt event and subsequent ice-lens formation in 2012 (Tedesco
et al., 2013; Nilsson et al., 2015). The recovery between 2013 and 2015 suggests a return to stronger volume scattering, likely due to new snow deposition hence the downward movement of ice lenses (Rennermalm et al., 2021).

Similarly, between 2018 and 2019, LeW experiences another minor drop of about 1 m, which coincides with the early melt observed in April and May 2019 (Tedesco and Fettweis, 2020) and the strong melt events between June and December 2018 (Fig. 6 in Houtz et al., 2021) that reduced volume scattering. Between 2019 and 2023, LeW increases and decreases alternately
occur over the northern and southern parts of the interior of Greenland.

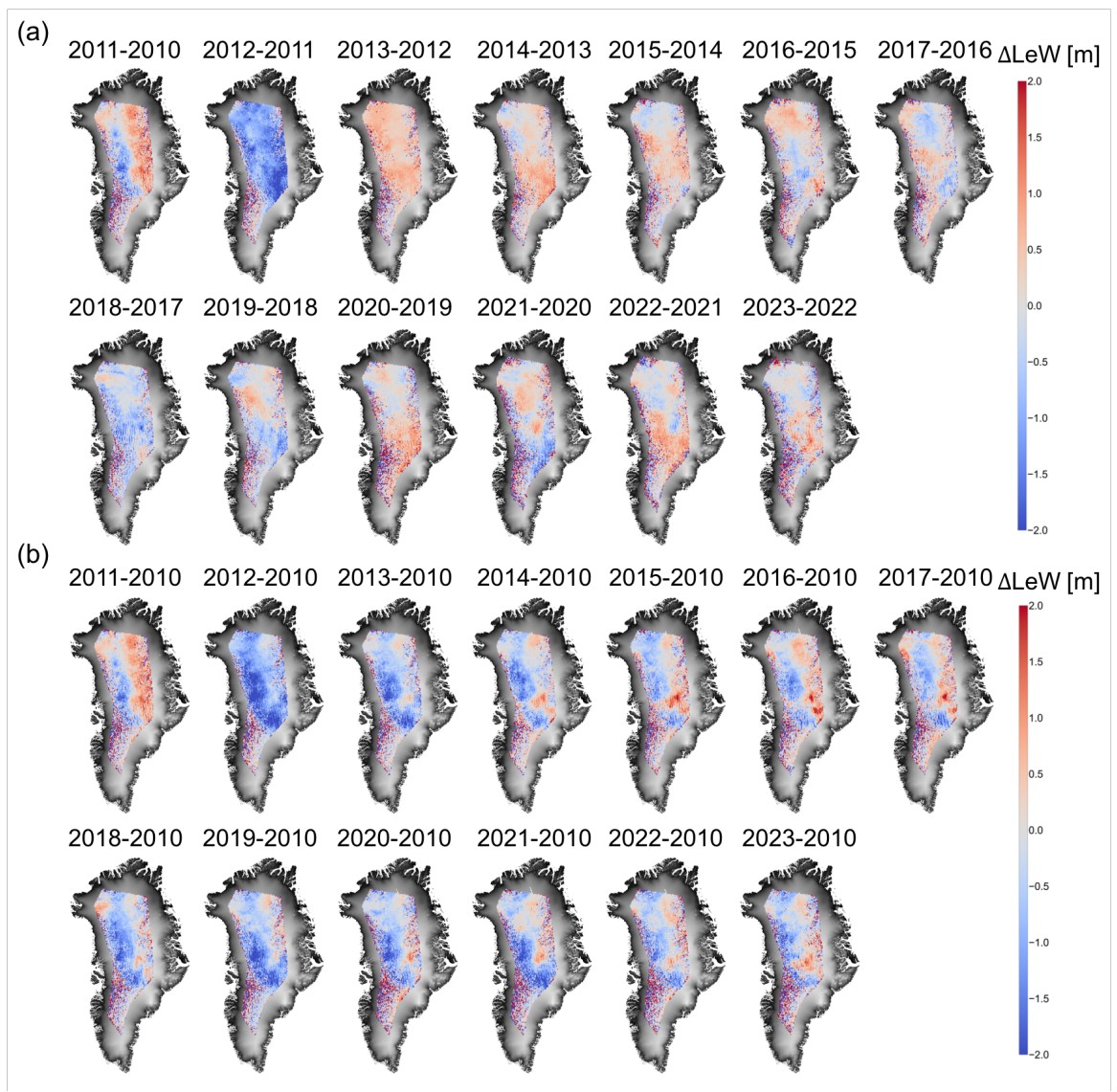

**Figure 4.** (a) Changes in non-melt season (October–April) average LeWs between successive seasons, and (b) changes in non-melt season average LeWs for the seasons 2011–2023 relative to the 2010 non-melt season average. Years refer to the start of each non-melt season.

Figure 4b compares the non-melt season average LeWs of each year with the 2010 non-melt season average, highlighting long-term variations relative to the extreme melt year of 2012. All years show a negative difference in the interior to western side of the ice sheet, except for the difference between 2010 and 2011, indicating that after the 2012 melt, LeW does not fully recover. It remains approximately 1.5 m below 2010 levels in the centre-west of the ice sheet. The area with negative values (i.e. no LeW recovery since 2012) shrinks between 2013 and 2018, expands again in 2019, remains stable until 2021, and recovers again until 2023.

When combining Figs. 4a and b, a notable recovery in LeW since 2012 is evident. Our interpretation of the LeW patterns suggests that in the region that shows this reduction–recovery pattern, the snow was initially dry before the 2012 melt, resulting in LeW patterns dominated by large volume scattering. The 2012 melt and the subsequent refreezing reduced volume scattering, causing a drop in LeW. LeW gradually recovered as the refrozen layer was buried, leading to the restoration of volume scattering. However, this recovery can be interrupted by more recent and frequent melt events, depending on the specific region. This interpretation can be supported by in situ measurements shown in Fig. 2. In most of the in situ firn profiles acquired after 2012, an elevated firn density ($\geq 500 \text{ kg m}^{-3}$) can be observed between $0.4$ m and $5$ m depth, depending on the time and location of the acquisitions. Typically, if the acquisition time of the in situ measurement is closer to the 2012 melt event (e.g. the Schaller et al. (2016a) acquisition), the high-density layer is shallower.

### 4.2 Effects of surface roughness (changes) on LeW (changes)

The comparison of LeW and wavelength-scale surface roughness time series between 2013 and 2019 (Figs. 5a and c) reveal a clear spatial correspondence between these variables. Zones of high roughness ($> 6 \times 10^{-4}$ m) coincide with elevated LeW values ($> 5$ m) around pixels C and D. The region south of pixel C corresponds to Fig. 4 of van den Broeke et al. (2023), where the modelled melt extent exceeds $10 \text{ kg m}^{-2} \text{ yr}^{-1}$. Similarly, in Figs. 5f and h, regions with high LeW ($> 7$ m) are associated with increased wavelength-scale surface roughness ($> 8 \times 10^{-4}$ m) at elevations below 1800 m.

We also compare the spatial LeW variations with macro-scale roughness where the wavelength-scale surface roughness data are not available. Figure 5e shows that macro-scale surface roughness follows a similar pattern to LeW (Fig. 5a), with a high standard deviation of elevation per grid cell ($> 50$ m) around pixel D. Fig. 6e illustrates that the macro-scale surface roughness decreases with increasing elevation, mirroring the LeW pattern in Fig. 6a, where LeW generally increases as elevation decreases.

The aforementioned combined analysis of wavelength-scale and macro-scale surface roughness indicates that, spatially, LeW tends to be higher in areas with increased roughness. Figures 5–6b and d investigate how the melt–refreeze events and the subsequent snow deposition affect the temporal variations in LeW and wavelength-scale surface roughness (the macro-scale roughness is temporally invariant in our study). Figures 5b and 6b show a generally increasing LeW following the 2012 melt event, both in regions north of pixel C and in elevation bands higher than 2000 m. On the contrary, the wavelength-scale surface roughness shown in Figures 5d and 6d varies rather arbitrarily. Among all the investigated pixels along the north–south transect, the temporal correlation coefficients between LeW anomalies and wavelength-scale surface roughness anomalies vary between -0.67 and 0.46. Among the ten elevation bands, the correlation coefficients vary between -0.12 and 0.26. Both sets of the

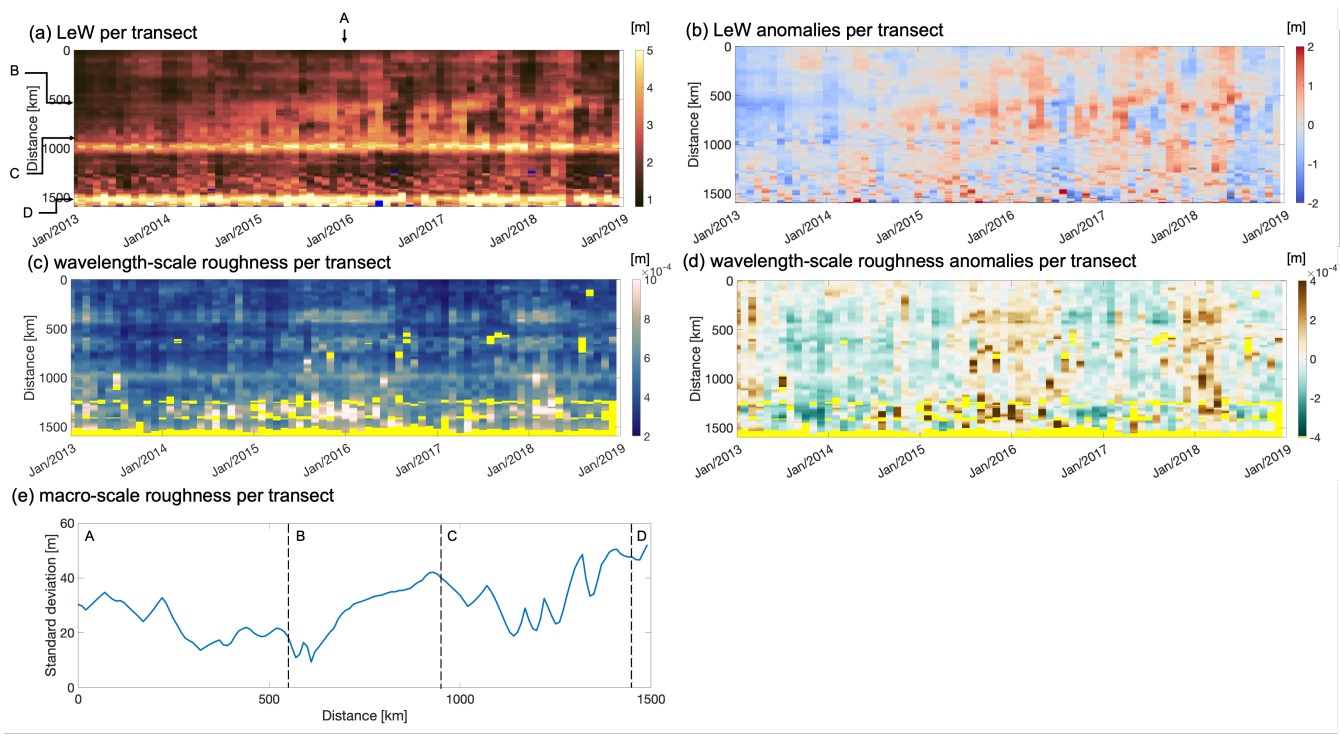

**Figure 5.** (a),(c),(e) LeW, wavelength-scale surface roughness (represented by surface root mean square height in unit of metres), and macro-scale roughness (represented by the standard deviation of ArcticDEM per 10 km cell); (b),(d) LeW anomalies, and wavelength-scale surface roughness anomalies along the north–south transect, respectively. A–D correspond to the inspected grid cells highlighted in Fig. 2. Blue colour in (a), grey colour in (b), and yellow colour in (c) and (d) indicate that the data are not available.

correlation coefficients indicate low temporal correspondence. This comparison shows that the snow deposition following the melt–refreeze events mainly effects the LeW by affecting volume scattering, while the surface scattering temporal variations play a minimal role in LeW temporal variations.

### 4.3 Correlation between LeW and laser-radar height offsets

Figure 7 presents the correlation coefficients between the LeW and $\Delta h$ time series for each 10 km$\times$10 km grid cell. Across the CryoSat-2 LRM zone, the two parameters generally exhibit a positive correlation, with a median value of approximately 0.6. Fig. 7b shows, however, that most correlation coefficients at the margin of the LRM zone are not significant ($p > 0.05$). Excluding them has minimal effect on the median correlation coefficient; it remains around 0.6. The lower correlation and significance towards the margins can likely be attributed to the compromised performance of ICESat-2 elevation measurements due to the large slopes, rough surfaces (Smith et al., 2023c) and scattering biases in the low-elevation regions (Smith et al., 2018). These biases can be propagated to the derived $\Delta h$, which may not properly indicate volume scattering variations.

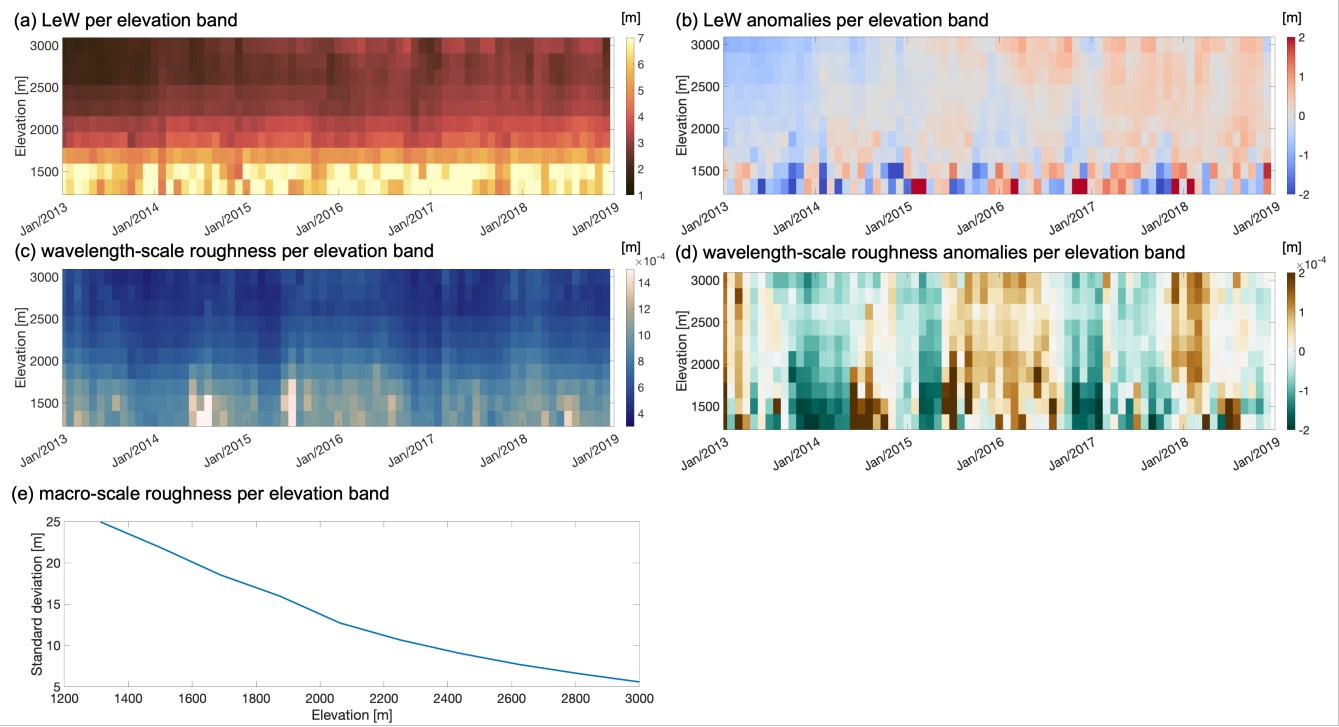

**Figure 6.** (a),(c),(e) LeW, wavelength-scale surface roughness, and macro-scale roughness; (b),(d) LeW anomalies, wavelength-scale surface roughness anomalies grouped by the ten elevation bands, respectively.

The positive correlation suggests that as LeW increases, the laser-radar height offset also increases, indicating a rise in volume scattering. However, most correlation coefficients remain below 0.9. This aligns with the findings of Nilsson et al. (2015), which showed that LeW was more sensitive to ice-lens formation and volume scattering variations than $\Delta h$. Therefore, LeW was used as an indirect parameter to assess volume scattering, particularly when ICESat-2 data were not available.

## 4.4 LeW variations versus model-derived firn property variations

Figure 8 shows the differences (with respect to the mean between October 2010 and April 2011) in monthly mean LeW, MAR density, IMAU-FDM density, and FAC time series along the transect highlighted in Fig. 2 (the absolute time series of these parameters are provided in Fig. A1). The most pronounced reduction in LeW (by 1–3 m compared to the non-melt season of 2010–2011) in Fig. 8a aligns with the sharp decrease in 2012, also shown in Fig. 4.

Along the investigated transect, in the pixels north of pixel C, a clear recovery in LeW can be observed after the 2012 melt event. Between pixels B and C, the recovery is disrupted again in summer 2019, corresponding to another extensive melt event (Tedesco and Fettweis, 2020) and the observation in Fig. 4.

In the pixels south of pixel C along the transect, the monthly mean LeW since 2010 does not show notable anomalies (e.g. reduction by 1–3 m compared to the non-melt season of 2010–2011, as observed for pixels north of pixel C) related to melt

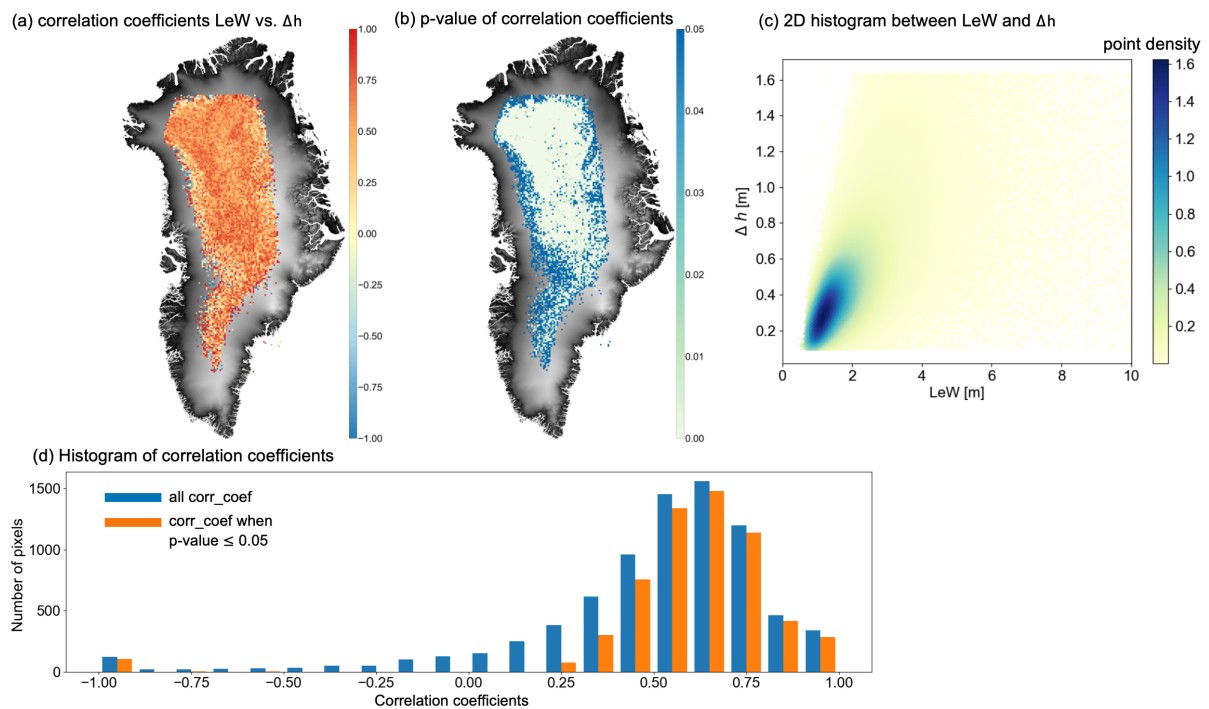

**Figure 7.** (a) Map of correlation coefficients between LeW and $\Delta h$. (b) Corresponding map of $p$-values for the correlation coefficients between LeW and $\Delta h$. The background is the 1 km $\times$1 km ArcticDEM. (c) 2D histogram showing the relationship between all LeW and $\Delta h$ data points, with point density estimated using a Gaussian kernel (Węglarczyk, 2018). (d) Histograms of all correlation coefficients (blue) and statistically significant correlation coefficients ($p \leq 0.05$, orange).

events (Fig. A2b). These pixels are characterised by recurrent melting each year (Fig. A2b) and higher snow accumulation rate (Fig. A2d). However, moments of LeW reduction can be observed in summer 2018, 2019, 2021 and 2023, corresponding to the high meltwater production and refreezing ($\geq$ 5 mmWE per month) in Figs. A2b–c.

Figure 9 shows the differences (with respect to the mean between October 2010 and April 2011) in LeW, MAR density, IMAU-FDM density, and IMAU-FDM FAC between September 2010 and September 2024 for different elevation bands (the absolute time series are shown in Fig. A3). In the low-elevation regions (below 1600 m), the LeW variation does not show a regular temporal pattern. On the contrary, these low-elevation areas demonstrate more notable seasonal variations in density and FAC (large density increases and FAC decreases in June) compared to other elevation bands. These density increases and FAC decreases coincide with the annual melt season (Fig. A4b–c).

Above the 1600 m elevation, a reduction in LeW is observed across all elevation bands following the 2012 melt event. The largest reduction is approximately 2 m, occurring in regions above 2500 m. This LeW reduction in mid-2012 corresponds to the extensive melting, increased densities in both MAR and IMAU-FDM models, and a decline in FAC. Smaller increases in

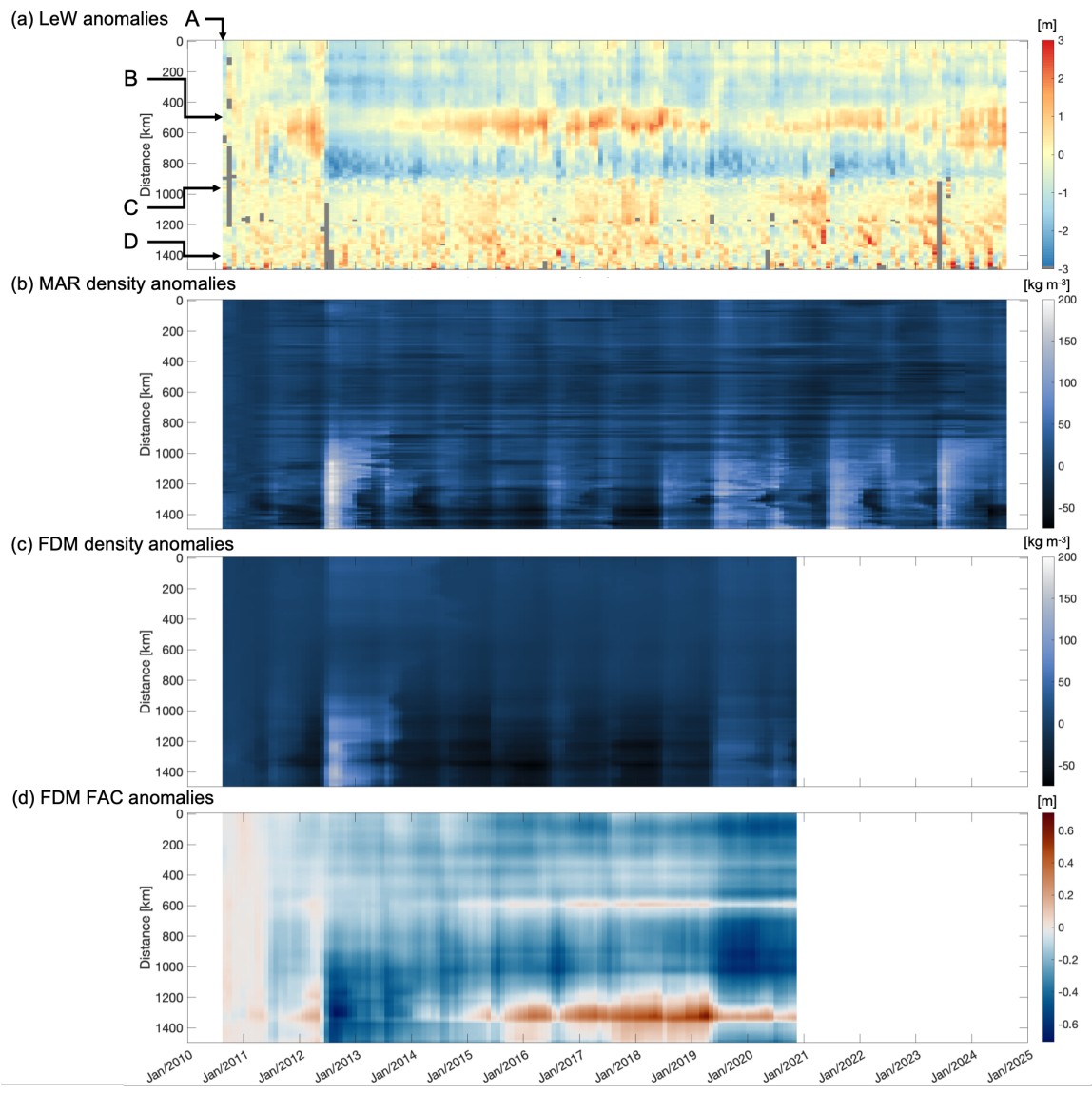

**Figure 8.** Differences (with respect to the mean between October 2010 and April 2011) in monthly mean LeW, MAR density of the top 1.5 m of firn, IMAU-FDM density of the top 1.5 m of firn, and IMAU-FDM firn air content (FAC) time series per pixel along the transect visualised in Fig. 2. The FAC time series are normalised with respect to the long-term mean of each pixel. The y-axes refer to the distance from the northernmost pixel. Arrows indicate the inspected pixels A–D. Grey colour in (a) indicates the values that are not available.

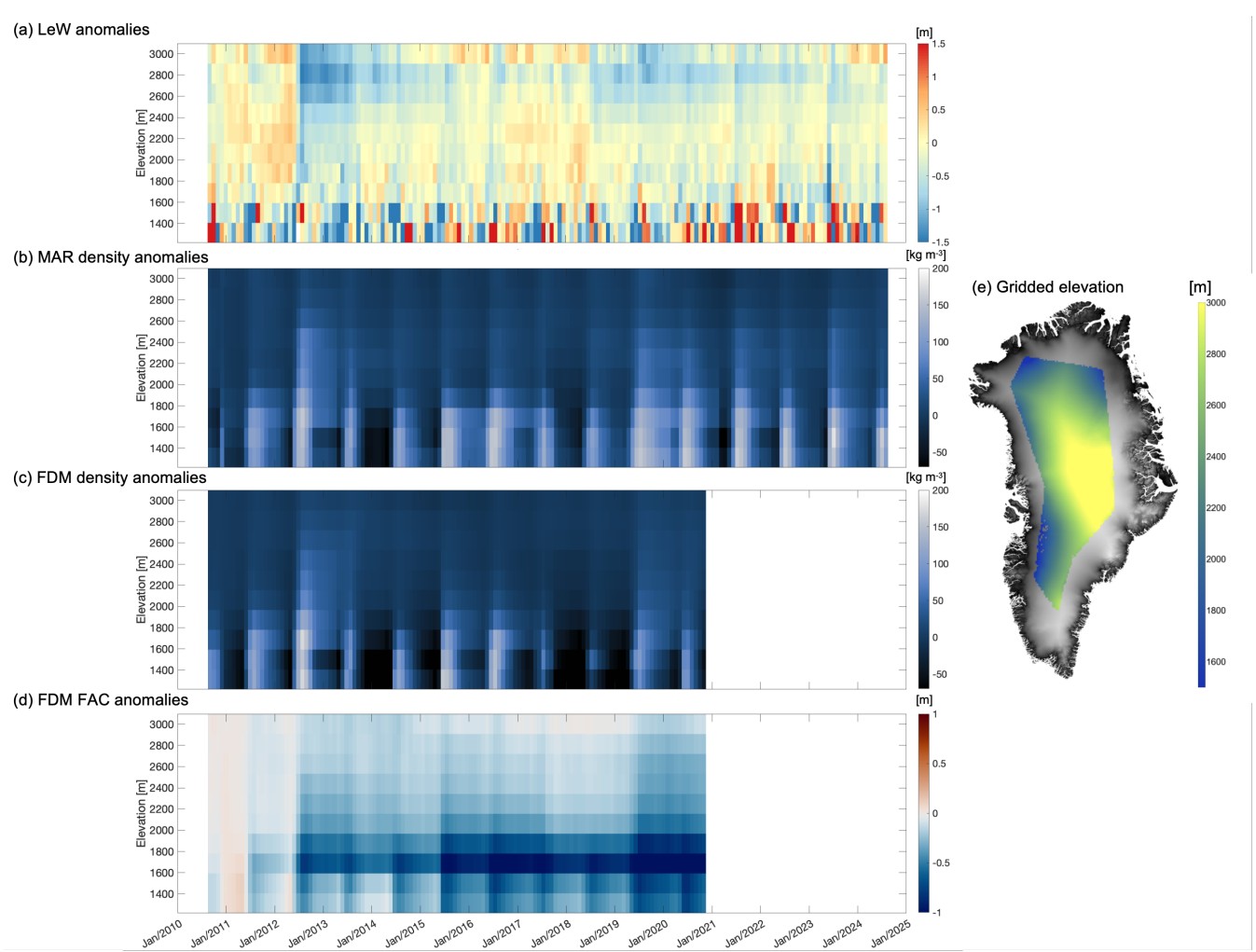

**Figure 9.** Differences (with respect to the mean between October 2010 and April 2011) in time series of monthly mean LeW, density from MAR, density from IMAU-FDM, and FAC from IMAU-FDM, grouped by a down-sampled (gridded) DEM. A map of the gridded DEM is provided on the right (e), with the original 1 km ×1 km ArcticDEM as background.

density and decreases in FAC were also observed in mid-2015, mid-2016, and mid-2019 above 1800 m, with a slight decline in LeW during mid-2015, mid-2016, and mid-2018.

From the time series, we also observe that in the 1800 m–2200 m range, the LeW recovery rate is faster than in regions above 2200 m (approximately 0.6 m yr$^{-1}$ versus 0.2 m yr$^{-1}$). By spring 2018, in areas above 1800 m, the LeW had nearly returned to pre-2012 levels (about 1 m lower). However, it temporarily dropped in mid-2018 by about 0.5 m and experienced another recovery process. This trend is consistent with the observations in Fig. 4b. Notably, our time series indicate that aside from the well-documented 2019 melt event (Tedesco and Fettweis, 2020) (characterised by density increases and FAC decreases), a slight decrease in LeW above 2200 m has been observed in mid-2018, suggesting reduced volume scattering. However, unlike other events, the LeW decrease in mid-2018 does not correspond to a density increase or FAC decrease in the models.

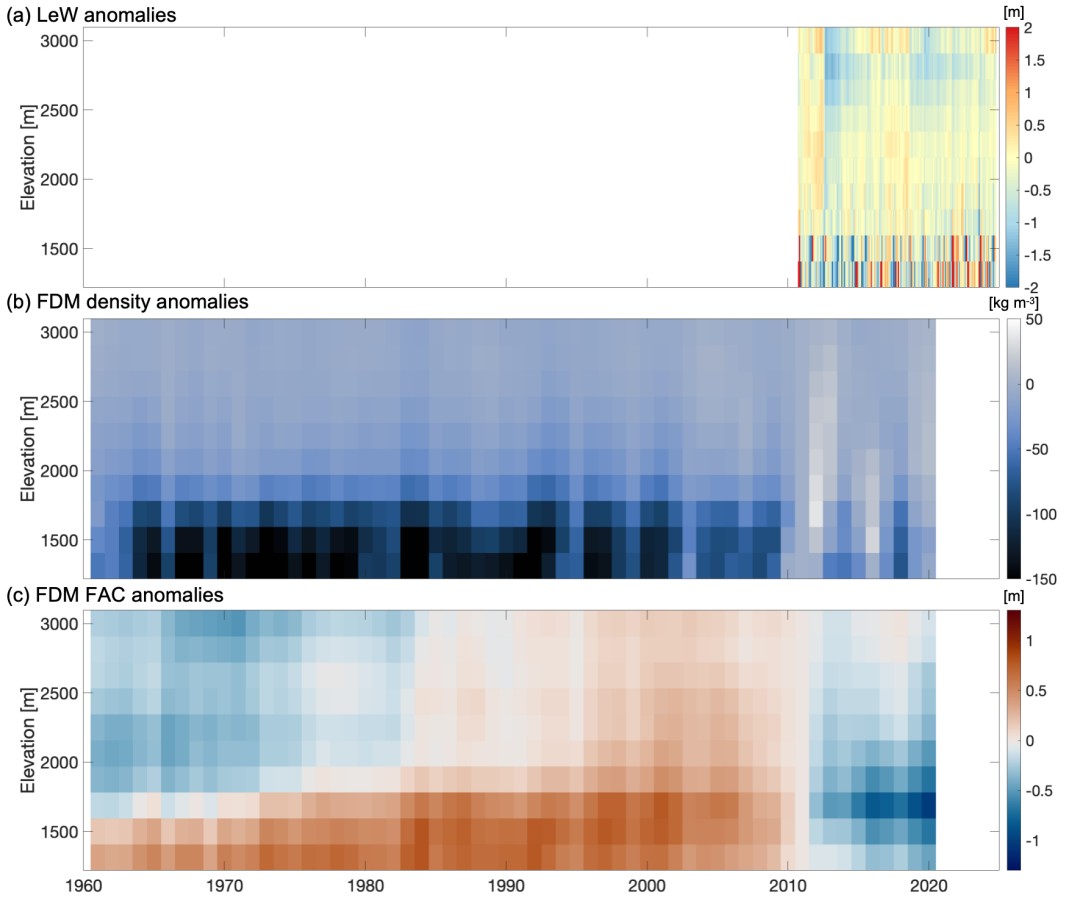

**Figure 10.** (a) Difference (with respect to the mean between October 2010 and April 2011) in time series of monthly mean LeW; (b),(c) difference (with respect to the annual mean of 2011) in time series of yearly mean density and FAC from IMAU-FDM, grouped by a down-sampled (gridded) DEM.

Finally, Fig. 10 displays the difference (with respect to the mean between October 2010 and April 2011) in monthly mean LeW between 2010 and 2024, and the difference (with respect to the annual mean of 2011) in annual mean IMAU-FDM density and FAC time series from 1961 to 2020 for different elevation bands. This comparison aims to determine whether the changes in volume scattering observed in CryoSat-2 LeW show a recent instability in Greenland's firn. Since 2012, the firn density of the upper 1.5 m has been notably higher—by up to $50 \text{ kg m}^{-3}$)—compared to the period prior to 2012. FAC below 2000 m dropped considerably compared to earlier periods.In the elevation bands above 2000 m, between 2012 and 2020, FAC experienced drops and recoveries that correspond to the LeW pattern between 2012 and 2018, followed by another decline between 2018 and 2020, which also aligns with LeW trends. However, in these elevation bands, the FAC level overall remains at the same level as the FAC level before 1980. Therefore, we cannot yet conclude that the firn layer above 2000 m elevation has experienced fundamental altering in the past decade.

## 5  Discussion

This study explores the capability of using LeW derived from CryoSat-2 waveforms to assess the spatial and temporal variations of firn in Greenland caused by melt–refreeze events. By first showing the yearly evolution of LeW, we demonstrate that LeW can successfully capture the strong melt events such as in 2012, 2019 and 2021. It also demonstrates the recovery of the firn layer due to the deposition of new snow, which is consistent with the observation and prediction from Nilsson et al. (2015).

Outside of the central dry snow zones, LeW is constantly high, as surface features such as roughness and topography begin to dominate its spatial variability (also noted by Ronan et al. (2024)). This was confirmed through comparisons with ArcticDEM, firn models, and the data from Scanlan et al. (2023), where a high LeW spatially corresponds to a high wavelength-scale and macro-scale surface roughness. To distinguish whether the temporal variations in LeW are also caused by surface roughness, we derived the temporal anomalies in LeW and wavelength-scale surface roughness with respect to their mean between 2013 and 2018 (due to the limitation in data availability). This experiment shows that contrary to the spatial variations, the temporal variations in LeW are independent of variations in surface roughness. It is important to note that this experiment has not yet considered the temporal variability of macro-scale roughness, as it is derived from the static ArcticDEM mosaics. Future studies are encouraged to investigate this by computing the standard deviation of elevations within different CryoSat-2 footprint at different acquisition time.

To further investigate LeW as a measure of firn volume changes, we compared it with laser-radar height offsets (calculated as the height difference between ICESat-2 and CryoSat-2). Overall, the relationship is similar to the conclusion of Michel et al. (2014), who proposed a linear function between LeW and penetration depth in flat regions of Antarctica. Our study, although not explicitly defining a "flat" region, delineated the interior of Greenland where the correlation between LeW and laser-radar height offsets is positive (around 0.6) and is significant ($p \leq 0.05$). In this region, it is most probable that the temporal LeW variations can be caused by variations in volume scattering. Therefore, our study provides a valuable framework for understanding firn response to melt events across different regions of Greenland.

This study is the first known demonstration of a Greenland-wide (within the CryoSat-2 LRM data coverage) LeW time series analysis, supported by two different firn models. The time series between 2010 and 2024 show that the 2012 melt event had a more prolonged impact than any other following melt events, resulting in a reduction in LeW that persisted until 2018, especially over the central-west Greenland. Recurrent melt events since 2018 have interrupted the firn recovery that was expected by Nilsson et al. (2015), underscoring the importance of monitoring these processes over long timescales. These findings highlight the value of CryoSat-2 LeW in assessing post-melt firn evolution, particularly at higher elevations.

When observed over a longer time scale (between 1960 and 2020), firn models indicate that the most pronounced firn changes, such as increases in density and decreases in FAC, occur below 1600 m elevation. At these lower elevations, LeW changes are less pronounced, not showing any clear temporal pattern compared to the mean LeW over the non-melt seasons between 2010 and 2011. In contrast, at elevations above 2000 m, LeW is comparably sensitive to the long-term effect of refreezing layer as FAC and density, while exhibiting higher sensitivity in 2018. This sensitivity suggests that LeW data could play a crucial role in refining firn models and improving radiative transfer models. For example, currently, radiative transfer modelling has been most successful in understanding firn property variations in Antarctic dry-snow zones (Adodo et al., 2018; Larue et al., 2021). How the refrozen layers in high-elevation zones in Greenland act as a reflective layer and hence affect the radar altimeter signal could be better represented in the modelling. Subsequently, the density and grain size changes following the melt–refreeze events and the potential new-snow deposition could also be derived with the combination of radiative transfer modelling and radar waveform information. Such a method has the potential of improving firn models through data assimilation (Weng, 2007), especially for higher elevations, where existing models may underestimate the impacts of melt events on volume scattering.

Our findings indicate that in southern and low-elevation regions of Greenland, where surface scattering dominates and melt events are more frequent, LeW is less effective in capturing firn changes. Future studies could address these limitations by incorporating additional altimeter-derived parameters such as trailing edge slope (TeS), waveform peakiness, and backscatter coefficients, which are more sensitive to surface scattering processes (Nilsson et al., 2015). These parameters, combined with LeW, would offer a more complete picture of surface and volume scattering interactions.

To better simulate the complex contributions of surface and volume scattering, radiative transfer models (Adodo et al., 2018; Larue et al., 2021) can be employed. These models, when taking into account the varying viewing geometry of the satellite, can enable more accurate representations of how melt–refreeze processes (characterised by varying temperature, firn density, microstructure and grain size) impact firn properties.

The ongoing ICESat-2 mission should also continue to provide the opportunity to indicate Ku-band radar penetration abilities, which also allows to continuously monitor changes in subsurface firn due to melt–refreeze processes. This could complement CryoSat-2 data, offering a higher-resolution view of firn structure over time. Combining radar altimeter data from different frequencies can also help derive volume scattering information from different subsurface layers. According to Lacroix et al. (2008) who compared waveform parameters from S-band and Ku-band radar altimeters, the impact of surface scattering as well as from snow grain size decreases with an increasing radar frequency. According to Scanlan et al. (2023) who derived firn properties using both Ku-band and Ka-band radar altimeters, radar altimeters operating in a lower frequency are sensitive to

firn densities at a larger depth. For future dual-frequency radar altimeters, e.g. the Copernicus Polar Ice and Snow Topography Altimeter (CRISTAL) mission which operates in both Ku- and Ka-bands (Kern et al., 2020), the different penetration abilities and sensitivities to firn properties offer the potential of a multi-layered analysis approach. For a higher frequency such as Ka-band, the penetration depth is smaller, hence we expect a quicker recovery of LeW after a melt event than that of Ku-band. This different recovery rate can help future studies to locate the subsurface refrozen layers and derive accumulation rate. Such a multi-layered approach can be particularly useful in regions where surface and volume scattering overlap, offering more nuanced insights into firn changes.

To enhance future firn studies, LeW data can be integrated into firn models to improve predictions of melt impacts on volume scattering, particularly at higher elevations. Although not presented in this study, the Goddard Space Flight Center (GSFC) firn model (Medley et al., 2022) and Glacier Energy and Mass Balance (GEMB) firn model (Gardner et al., 2023) can also be incorporated in the satellite time series analysis by both qualitatively indicating the presence of subsurface ice lenses and by quantitatively deriving firn properties with radiative transfer models. By using interdisciplinary approaches, we can deepen our understanding of how melt events affect firn properties over the long term, improving our ability to predict future ice sheet dynamics.

## 6   Conclusions

This study explored and demonstrated the possibility for using the LeW derived by CryoSat-2 to assess spatio-temporal changes in Greenland firn status caused by melt–refreeze events. While previous studies indicated a recovery pattern in LeW when new snow is deposited on top of the refrozen layers, our study further investigated whether this process can be observed with the help of LeW time series. Our analysis showed that the recovery speed can be related to the elevation and new snow deposition. However, the recovery is hampered by more recent melt events (although not as severe as the 2012 event). In central-west Greenland, the LeW never recovered to the pre-2012 level; in regions where LeW managed to recover, new melt events resulted in new LeW reductions, which indicates a reduction in volume scattering hence a reduction in its capacity to store meltwater. Such alternation can also be confirmed using the long-term time series, which showed a decreasing LeW, an elevated density and a decreasing FAC in recent decades.

Finally, this study has demonstrated the reliability and limitations for using LeW from radar altimeter to understand the melt–refreeze processes and the subsequent volume scattering variations in Greenland firn, paving the way for the study of subsurface firn processes in a changing climate. The use of a combination of CryoSat-2 and ICESat-2 height measurements can also contribute to the study of Ku-band penetration ability hence volume scattering variations induced by melt–refreeze events.

*Code and data availability.* Software for the in-house processing of CryoSat-2 data from L1b to L2 is available on request from Cornelis Slobbe. The MAR datasets are available on http://ftp.climato.be/fettweis/MARv3.14/Greenland/. The IMAU-FDM datasets are available on

request from Max Brils. ArcticDEM is provided by the Polar Geospatial Center under NSF-OPP awards 1043681, 1559691, and 1542736. The CryoSat-2 L1b and L2I data are provided online by ESA and the ICESat-2 L3A data are provided online by NSIDC (https://nsidc.org/data/atl06). The surface roughness data is available on https://doi.org/10.11583/DTU.21333291.v1.

## Appendix A: LeW and MAR melt, refreeze and total snow height time series

Figure A1 shows the absolute time series of LeW, MAR density and IMAU-FDM density, and the normalised IMAU-FDM FAC with respect to the long-term mean, along the north–south transect shown in Fig. 2. Over Greenland, FAC increases with elevation and exhibits substantial spatial variability, ranging from approximately 0 m at the ice sheet margins to over 20 m in the interior. However, when juxtaposing FAC over the entire study area, spatial variations are pronounced, whereas temporal variations are more subtle. To enhance the temporal signal relative to spatial variation, we subtract the long-term mean of each pixel (using the same grids as the resampled CryoSat-2 LeW) from the time series. Finally, monthly mean density and FAC are computed to align with the CryoSat-2 time series.

Figure A2 illustrates the time series of monthly mean LeW, MAR meltwater production, meltwater refreezing and accumulated total snow height change along the transect highlighted in Fig. 2. The meltwater production and refreezing correspond to the density increases in Fig. A1b. Figure A2a indicates that the overall decrease in LeW in 2012 corresponds to both the extensive meltwater production as well as refreezing. A slight LeW decrease in mid-2019 between pixels A and C also corresponds with the melt–refreeze event. By comparing Fig. A2a and Fig. A2d, we notice that pixels between pixels A and B show a slower LeW recovery than between pixels B and C, which corresponds to a lower cumulative total snow height change (i.e snow accumulation) since the melt event in July 2012. However, around pixel B, the LeW recovery between 2014 and 2018 is higher that that between pixels A and B, despite a lower snow accumulation. This may be attributed by the difference in properties and structures of the upper firn layer (Ronan et al., 2024).

Figure A3 shows the absolute time series of LeW, MAR density and IMAU-FDM density, and the normalised IMAU-FDM FAC with respect to the long-term mean, grouped by the ten elevation bands. Similarly, we present the time series per elevation band in Fig. A4. The melt–refreeze patterns correspond with the density increases in Fig. A3b. However, differently from Fig. A2a and Fig. A2d, Fig. A4a and Fig. A4d do not show a high correspondence between LeW recovery and total accumulated snow: the total accumulated snow height from July 2012 shows the highest increase between 2014 and 2024 at around 2400 m elevation, while the fastest LeW recovery occurs between 1800 m and 2200 m. This discrepancy could be attributed to the significantly larger snow accumulation in the South of Greenland than the North, despite being at the same elevation.

*Author contributions.* WL, SL and BW designed the study. WL conducted data management, processing, and analysis; produced the figures; and provided the manuscript with contributions from the other co-authors. SL provided support on data visualisation and analysis. BW provided expertise for data analysis. CS provided expertise and software for radar altimetry processing and data analysis. XF generated the MAR outputs. MB generated the IMAU-FDM outputs.

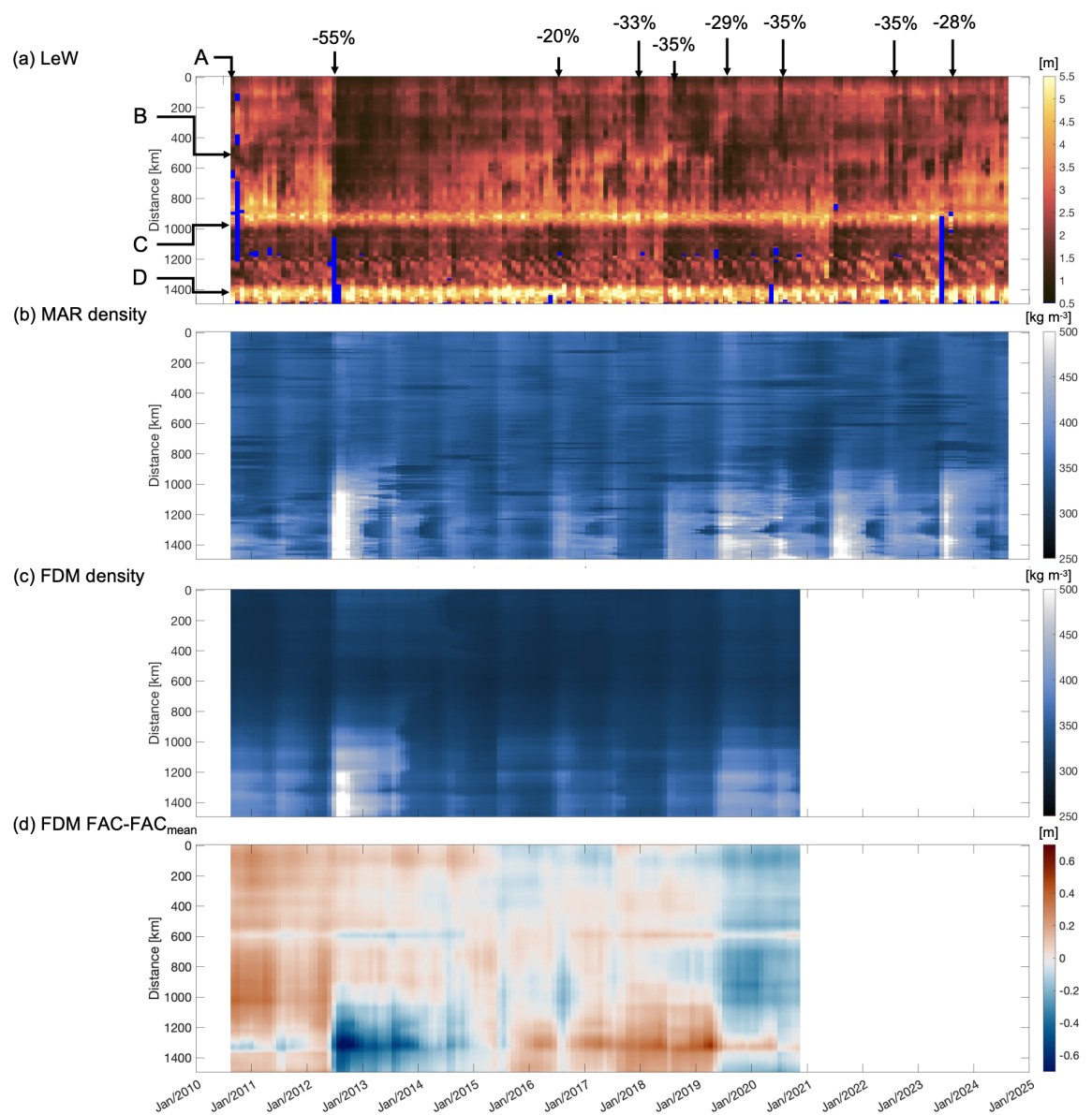

**Figure A1.** Monthly mean LeW, MAR density of the top 1.5 m of snow, IMAU-FDM density, and IMAU-FDM firn air content (FAC) time series per pixel along the transect visualised in Fig. 2. The FAC time series are normalised with respect to the long-term mean of each pixel. The y-axes refer to the distance from the northernmost pixel. Arrows indicate the inspected pixels A–D. Large LeW decreases with respect to the previous month for pixels north of pixel C are labelled. Blue (a) colour indicates the values that are not available.

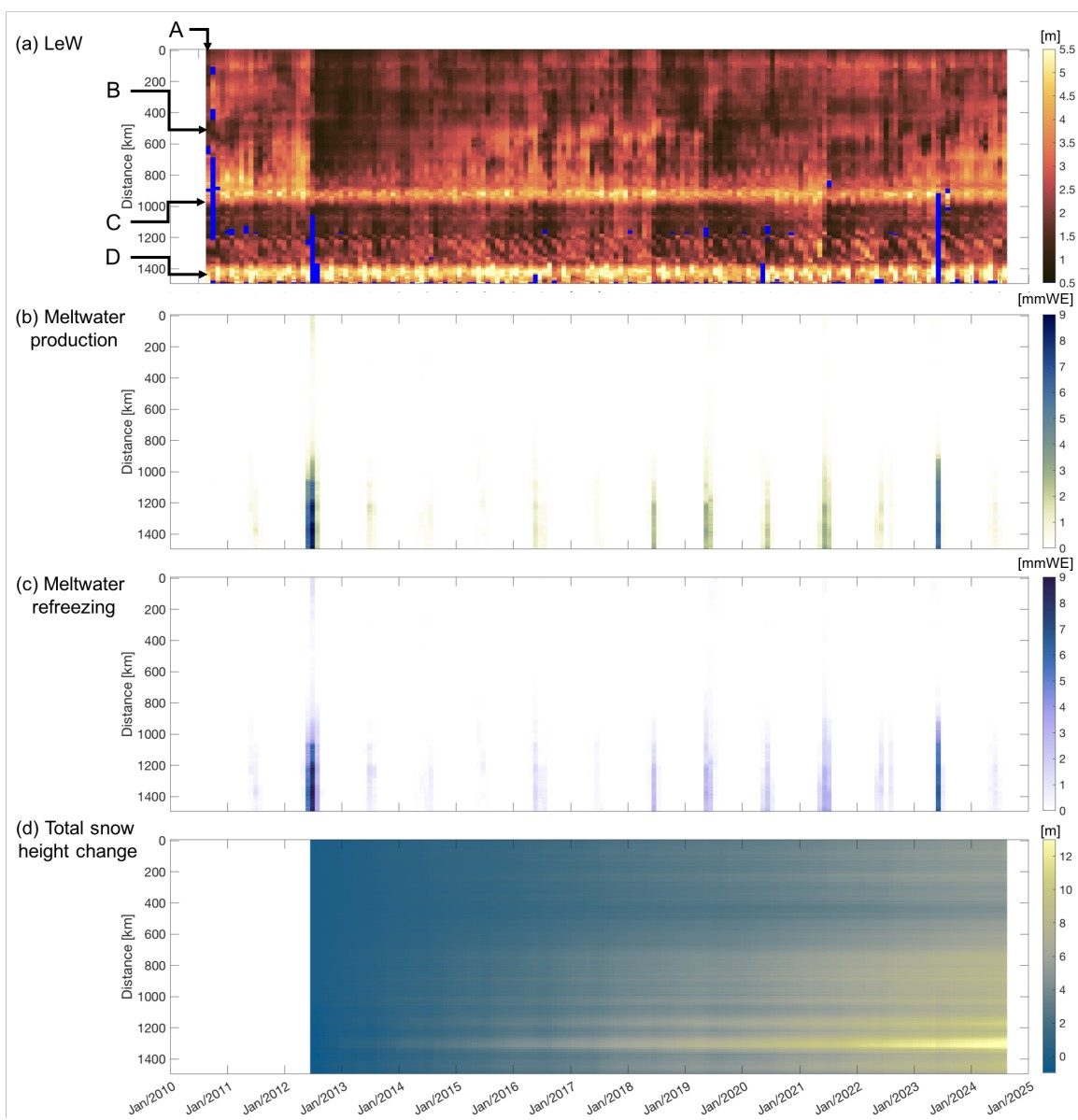

**Figure A2.** Monthly mean LeW, MAR surface meltwater production, MAR meltwater refreezing, and monthly accumulated total snow height change (i.e. snow accumulation) from 12 July 2012 time series per pixel along the transect visualised in Fig. 2. The y-axes refer to the distance from the northernmost pixel. White colour indicates the values that are not available. Arrows indicate the inspected pixels A—D.

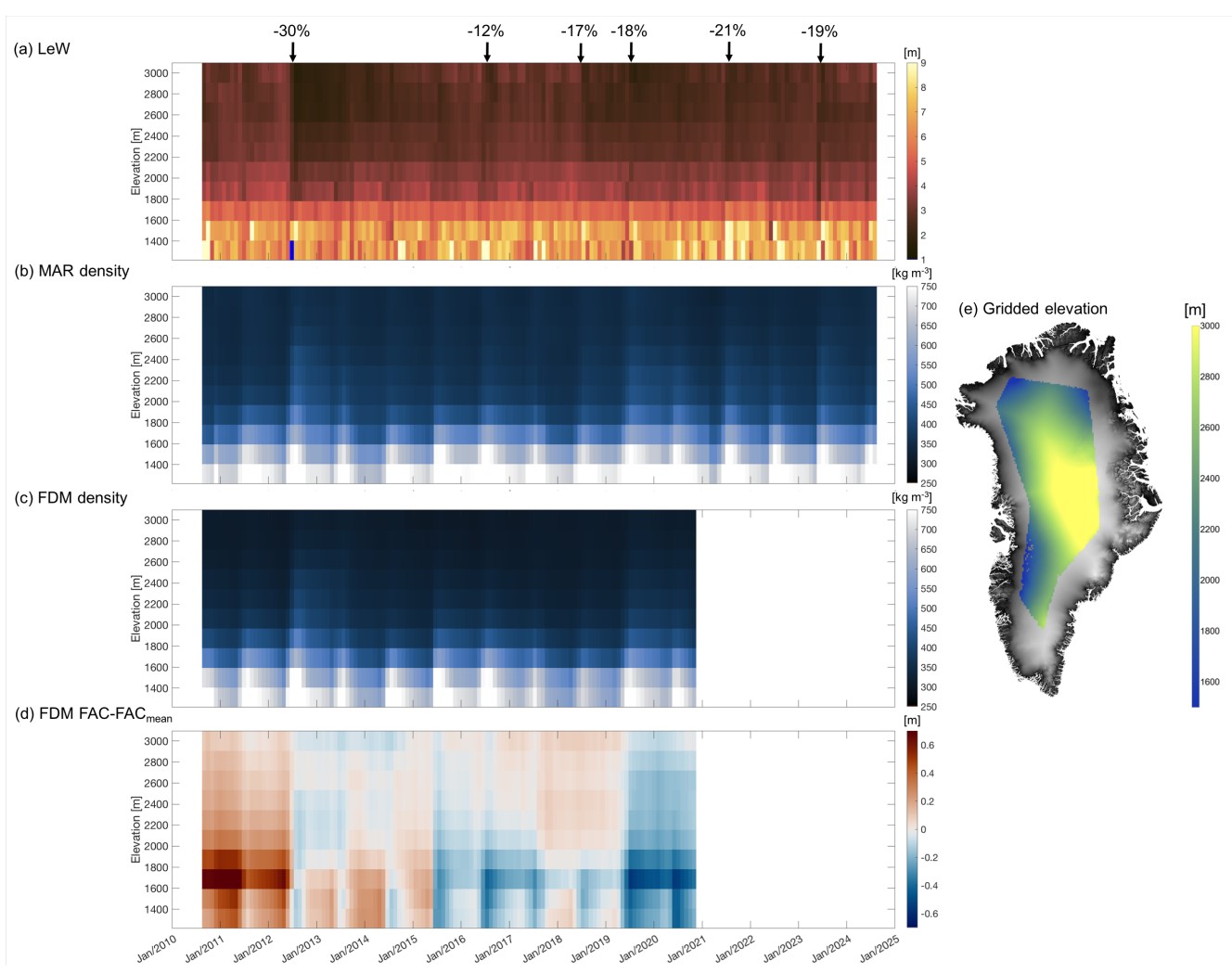

**Figure A3.** Time series of monthly mean LeW, density from MAR, density from IMAU-FDM, and normalised FAC from IMAU-FDM, grouped by a down-sampled (gridded) DEM. A map of the gridded DEM is provided on the right (e), with the original $1\ \text{km} \times 1\ \text{km}$ ArcticDEM as background.

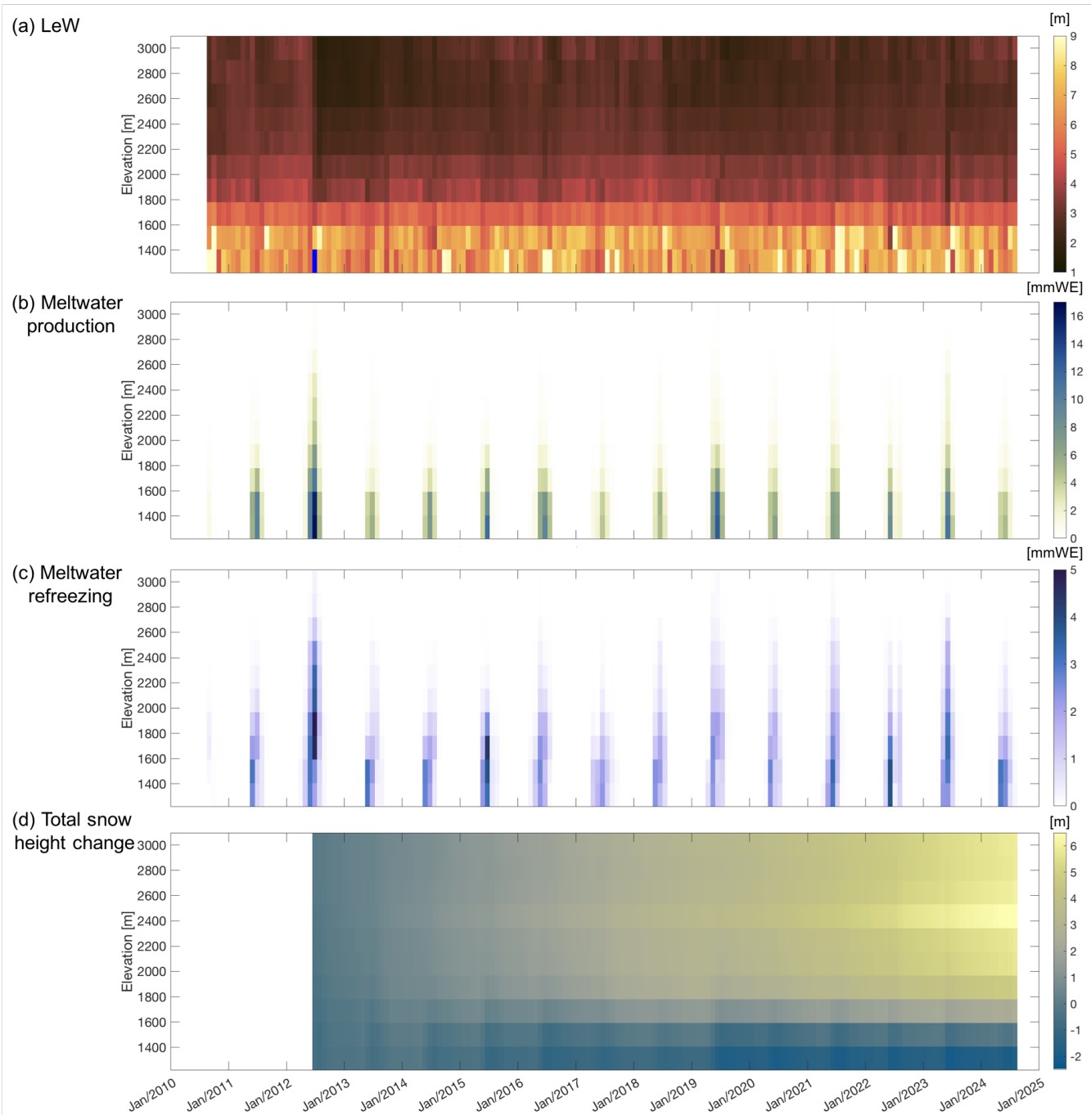

**Figure A4.** Monthly mean LeW, MAR surface meltwater production, meltwater refreezing, and monthly accumulated total snow height change (i.e. snow accumulation) from July 2012 time series grouped by a down-sampled (gridded) DEM. White colour indicates that the data are not available.

*Competing interests.* Stef Lhermitte, Bert Wouters and Xavier Fettweis are members of the editorial board of The Cryosphere.

*Acknowledgements.* The research is supported by the Dutch Research Council (NWO) through the ALWGO.2017.033 project. ArcticDEM is provided by the Polar Geospatial Center under NSF-OPP awards 1043681, 1559691, and 1542736. Some of the colour maps are acquired from Crameri (2023) under an MIT License. The authors would like to thank Horst Machguth for reviewing and editing this paper. We would
also like to thank the referees for reviewing and providing recommendations to improve this paper.

ChatGPT is used for language checks in parts of the manuscript.

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
