# Peer review of "Assessing spatio-temporal variability of melt—refreeze patterns in firn over Greenland with CryoSat-2"

_EGUsphere, 2024_

## Author Comment (AC1)

**Response to Referee 1 on egusphere-2024-3251**

First, we would like to thank the Referee for reviewing and commenting on the manuscript, which will improve the quality of the manuscript. Please find the item-by-item reply below, with the responses in blue. All the suggested changes will be implemented in the revised text that will be uploaded.

**General Comments**

This manuscript presents an interesting study into using the leading edge width (LeW) of CryoSat-2 measurements between 2011 and 2021 to investigate the long-term characteristics of Greenland firn conditions. The authors begin their manuscript with an introduction to the importance of understanding melt events, their effects across the Greenland Ice Sheet (GrIS) and how they intend to approach the problem using remote sensing data supported by in situ measurements and numerical climate model outputs. In Section 2, the authors present the various datasets considered in their study. Section 3 then outlines how the data are used, combined and the types of analyses the authors perform; the results of which are presented in Section 4. Finally, in Section 5, the authors reflect on the implications of the results and how future studies could expand on them, before the main conclusions are outlined in Section 6.

Overall, the authors have analyzed and presented a substantial amount of data. They take a very thorough approach to assessing the long-term spatial patterns in CryoSat-2 LeW by incorporating another satellite altimeter (i.e., ICESat-2), other derived satellite and airborne datasets (i.e., roughness, radar-laser offsets, topography), in-situ measurements (i.e., densities), and model results (i.e., densities, meltwater content, and firn air content). The scope of bringing all the data together is impressive and I think very exciting direction of work. Relatedly however, presenting so much data requires a very clear narrative and defined structure to help a general and non-specialized reader avoid being lost in all the details. My main comment on the manuscript is that I found this aspect of it to be underdeveloped, which could hamper the impact of the results.

While reading through study and going through all the results, I found it hard at times to get a sense of how all the different pieces fit together. I think the manuscript would benefit from a clearer statement of the central hypothesis and the physical reasoning behind it that would then frame the overall study. In the Introduction, the authors state the importance of melt and refreeze events and postulate that LeW could be used to study long-term patterns. What I think is missing though is how LeW is affected by melting/refreezing events. What is changing in the firn and how does that affect the radar signals? Concepts of surface and volume scattering as well as refrozen layers appear repeatedly later on in the manuscript, but I think explicit descriptions of what the authors mean by these, their linkage to the physical state of the firn and what that means for the CryoSat- 2 would substantially help fortify the overall narrative. This bridging between radar theory and more classical glaciological concepts will also strengthen the impact the manuscript will have by really outlining how all these pieces fit together and what the results mean. Some of the specific comments below will also be in this direction.

We appreciate the general comments. According to the Greenland snow and firn facies defined in Benson (1960), the dry-snow zone is characterised by snow and firn with low

density, small grain size and uniform crystals. In lower altitudes, the percolation zone is characterised by ice pipes or ice lenses with large grain size, due to the vertical or horizontal percolation of meltwater and its subsequent refreezing. In the lowest altitudes, the wet-snow zone also has ice pipes or ice lenses, but the densities are higher than those in the percolation zone due to the compaction from a higher temperature. Therefore, without melting, the microwave scattering within Greenland dry-snow zone should be stable (Tran et al., 2008). However, the strong melt events, e.g. the 2012 melt, can cause the formation of ice lenses within the dry-snow zone (Nilsson et al., 2015). These high-density ice lenses reduce the Ku-band radar penetration by approximately 50% (Otosaka et al., 2020). Moreover, the radar altimetry leading edge width (LeW) can be influenced by surface roughness, topography and surface penetration effects of the radar signal. Since the topography of the dry-snow zone is typically flat, we can observe the reduced radar penetration within dry-snow zones by observing the drops in LeW time series. As new snow deposits, the radar penetration hence LeW is expected to recover (rise) again (Nilsson et al., 2015).

However, within the CryoSat-2 LRM coverage, the percolation zone is also present, where the high-density firn may limit radar penetration, but the large firn grain size can increase the roughness, hence can increase LeW (Nilsson et al., 2015). This phenomenon causes the limitation of detecting refrozen layers within Greenland firn using LeW time series.

We will better present this concept in the revised manuscript.

I hope the authors find the following comments constructive as they work towards revising their manuscript.

**Specific Comments**
1) As the manuscript primarily centers on CryoSat-2 LeW results, I would suggest the authors consider revising the title to "Assessing spatio-temporal variability of firn scattering over Greenland with CryoSat-2". I understand that the inclusion of ICESat-2 data makes a case for multiple altimeters, but my impression is that the ICESat-2 data are more complementary to the main CryoSat-2 dataset. In a similar way to the MAR and IMAU climate model data, ICESat-2 data appear to be used more to help explain trends in the CryoSAt-2 data, not necessarily as the primary data source themselves.
This will be implemented in the revised manuscript.

2) Line 57 "The LeW is adopted as it is sensitive to volume scattering ..." Line 67 "... we have to understand both volume scattering and surface scattering ..." These are two instances where a more explicit statement of what the authors mean by volume and surface scattering could help improve the overall framing of the study. How/why LeW is sensitive to these two concepts and what on the surface and in the firn contributes to them? I think it would broaden the reach of the manuscript by removing the hurdle of needing to be familiar with nuanced radar theory concepts and motivate exactly why the specific model outputs are chosen for comparison with the LeW results in the latter stages of the manuscript.
We agree and, in the revised manuscript, we will clarify this by referring to the modelling results from Lacroix et al. (2008).

3) The authors dedicate Lines 35-51 motivating CryoSat-2 and LeW as a metric for studying firn. I recommend the authors consider expanding more clearly on the motivations for using the other datasets (e.g., ICESat-2, in-situ densities, dz, roughness, topography, and model results) to help explain the LeW results. What aspect of the LeW signal are these datasets being used to interpret? I found Lines 52-69 to be confusing as it was not always clear how these different datasets all supported the LeW analysis.

The use of in situ densities and modelled densities are to demonstrate the formation of high-density firn following melt-refreezing events. Specifically, the in situ data are used to show that these high-density layers are indeed gradually buried due to the new-snow deposition, hence LeW rises. Roughness and topography data are used to delineate the regions where LeW variations may be dominated by surface scattering rather than volume scattering, therefore we cannot effectively interpret the LeW time series in certain areas. The dz dataset was a comparison to show the strengths of our approach: compared to the OIB measurements, the CryoSat-2 LeW not only shows the melt event, but also the firn recovery. It also has much better spatial and temporal continuity. Finally, the use of modelled FAC provides insights into the overall condition of firn, as we may wonder whether the recurrent melt-refreezing patterns are causing a permanent change in firn's health.

This will be explained in better detail in the revised manuscript.

4) I recommend the authors consider reducing the number of adverbs (e.g., furthermore, finally, in addition, therefore, additionally, etc.) used to start sentences to make them more direct and impactful.

This will be improved in the revised manuscript.

5) To be more specific on the types of GrIS changes of interest in this study, I recommend the authors re-phrase Line 70 from "… assess long-term changes over the …" to "… assess long-term surface changes over the …"

We appreciate the suggestion of the referee, however surface changes may still be a bit shallow compared to the 1.5m snowpack in our study. Therefore, to make it more specific, we will specify that we are assessing sub-surface changes.

6) In Section 2.1, I recommend the authors include more detail on the nature of the CryoSat-2 LRM data and what differentiate them from other CryoSat-2 data products (e.g., what is unique/different in their acquisition/data processing?).

The CryoSat-2 LRM data are available within the interior of Greenland, SARIn mode is available over the coastal areas, and SAR mode operates over the sea. This will be elaborated in the revised manuscript. However, for the data processing, since SAR and SARIn modes are not available in our region of interest (Greenland interior), it may not be relevant to add everything in detail.

7) Line 89, please include the range resolution of CryoSat-2.

It should be made more explicit that $S_r$ should be $\frac{1}{2}c \cdot dt$, where $c$ is the speed of light and $dt$ is the waveform sampling interval (3.125ns) This will be improved in the revised manuscript.

8) In Figure 1, I'd ask the authors to consider including the b0.99 and b0.01 values for each waveform as well as map (perhaps as an insert) of where these two locations are in Greenland are.

The indications will be added in the revised manuscript.

9) Line 94, what high-resolution DEM model is used?

It should be the 100m resolution ArcticDEM mentioned in Section 2.3. Following the recommendation, we will mention the ArcticDEM at the beginning of the Data section.

10) In Line 97, the authors state that they used July measurements as indicative of post-melt conditions but there is no way for the reader to assess if melting has ceased at these locations by the time the data were acquired; especially knowing how extreme the melt extents observed in the summer of 2012 were. I would recommend the authors provide further support for this statement or consider using data from later in the year.

This will be changed into the September data in the revised manuscript.

11) I am not sure I fully understand the context for why two different grid resolutions (50x50 and 25x25) are used. I suggest the authors clarify this point.

Originally, we aimed to use the finer resolution (25km x 25km) to calculate long-term (decadal) statistics and the lower resolution (50km x 50km) 50 calculate monthly statistics. However, following the recommendation of Referee 2, we noticed that using 10km x 10km resolution can still ensure sufficient (more than 10) data points per pixel per month, while being consistent with the spatial resolution of the firn models. Therefore, we adopt the recommendations of both referees and will adopt the 10km x 10km resolution throughout the revised manuscript.

12) Line 115. Do all CryoSat-2 measurements in a given month have a corresponding ICESat-2 measurement within 50 m or are their spatial gaps? I'd also recommend the authors provide their reasoning for choosing 50m when the footprint of CryoSat-2 LRM data is much larger.

It is true that not all CryoSat-2 measurements have a corresponding ICESat-2 measurement within the 50 m radius. As shown in Table 1 of Li et al. (2022), such a criterion results in approximately 30 times fewer measurements in year 2019.

We agree that the footprint of CryoSat-2 LRM mode is much larger. The motivation of using a smaller search criterion was that over the undulating terrain, the true footprint of CryoSat-2 LRM should be smaller than the theoretical one, therefore we would choose a corresponding ICESat-2 point as close as possible, yet not largely reducing the number of valid points. However, the selection of 50 m is rather arbitrary, therefore we have conducted a sensitivity analysis, shown in Fig. R1. It can be observed that as the search radius increases, the number of valid dh increases, while the correlation between dh and LeW decreases. Especially, when using 800 m as the search range, which is comparable to the theoretical pulse-limited footprint of CryoSat-2 LRM, the correlation coefficients are overall below 0.5. Using 100 m and using 50 m do not demonstrate distinct differences. Therefore, we prefer to choose a search range as small as possible, which is also similar to the crossover principle proposed by Michel et al. (2014).

[Figure]

Figure R1. Comparison of correlation coefficients between dh and LeW when using different search range for the corresponding ICESat-2 point for each CryoSat-2 point.

13) With how Sections 2, 3 and 4 are structured, the CryoSat-2/ICESat-2 results from Figure 2 and Lines 122-128 seem to be more suited to Section 4 than Section 2. I understand they are used again in Section 2.4, but could Section 2.4 be treated more abstractly by referring to a subsurface depth extent to be determined later? The current placement seems to interrupt the flow of describing all the individual datasets considered.
We appreciate the suggestion of the referee and this will be re-structured in the revised manuscript.

14) Lines 134-136. I recommend the authors elaborate a bit more on how "computational efficiency" necessitates using both the 100m and 1km ArcticDEMs in these two instances. What about these specific applications makes the use of two different DEMs more efficient?
The use of the 100m ArcticDEM is inherited from Li et al. (2022): in principle, ArcticDEM can be available with the resolution of 2m (Porter et al., 2018). However, since our computation is performed in MATLAB, it is difficult to load the large ArcticDEM Geotiff files. Therefore, the finest resolution considered in our studies is 100m.

We admit that the "computationtal efficiency" that comes with the 1km ArcticDEM can be confusing: originally, we divide the interior of Greenland into a 50km-by-50km grid, so that the topography for each pixel is represented by the standard deviation of the 50-by-50 ArcticDEM height values within the pixel. However, following both referees' comments, we will reconsider the resolution of the grid over the interior Greenland, and use the 100m ArcticDEM to compute the topography in the revised manuscript.

15) Figures 2, 3, 4, 6, 7, and 9. I recommend the authors elaborate why they used a DEM from Helm et al. (2014a, b) as their basemap instead of one of the ArcticDEMs they use in their analysis. Also, I'd recommend including a colorbar for the elevations the first time it is used.
The only reason is that the Helm et al. (2014) DEM focuses on Greenland, while ArcticDEM covers the entire Arctic including the ocean surrounding Greenland and is difficult to crop. We will edit the ArcticDEM in the revised manuscript for better consistency. The colorbar will also be added.

16) Line 146. I recommend the authors clarify how the weights are determined in their weighted average densities.

The weights are defined as the thickness of each layer. This will be elaborated in the revised manuscript.

17) I recommend the authors consider better motivating the inclusion of the IMAU FAC. FAC is a column-integrated measurement (Line 168) whereas LeW derived from CryoSat-2 is seemingly only sensitive to the upper few meters (Figure 2). Why would these two datasets derived over different depth ranges be considered comparable?

We agree with the concern of the referee. In general, we aim to use the density dataset at upper 1.5m to prove that CryoSat-2 is indeed sensitive to the changes that happen within the 1.5m firn layer. On the other hand, as we also try to learn about the overall condition of the firn (beyond this 1.5m threshold), the FAC over the entire snowpack is used as additional information to indicate whether Greenland firn experiences a continuous decrease in pore spaces. We will better motivate it in the revised manuscript.

18) Line 168. The "but" in "... 1.5 m but the FAC ..." can be removed.

This will be changed from

*"While the density is calculated over the first 1.5 m but the FAC is calculated for the entire firn column."*

to

*"While the density is calculated over the first 1.5 m, the FAC is calculated for the entire firn column."*

in the revised manuscript.

19) I recommend the authors expand on why these particular in-situ firn density measurements are used instead of the more comprehensive SUMup dataset (i.e., Vandecrux et al. 2023)? Furthermore, why is it necessary for firn density profiles to contain the 2012 melt year (Line 185)?

*Vandecrux, B. et al. The SUMup collaborative database: Surface mass balance, subsurface temperature and density measurements from the Greenland and Antarctic ice sheets (1912-2023). Arctic Data Center https://doi.org/10.18739/A2M61BR5M (2023).*

We appreciate the referee for the suggestion and will include this dataset in the revised manuscript.

The main reason to contain the 2012 melt year is to provide more sound evidence that the 2012 melt results in a visible density increase, which can also be observed in the modelled firn densities. This high-density layer is buried in the subsequent years, therefore the recovery in LeW can be eventually observed.

20) Line 212. These 10 DEM elevation groups have not been mentioned yet, so I do not follow how they can be "aforementioned". I recommend the authors clarify this statement.

It should be 8 groups equally divided between 1500m and 3000m, 1 group below 1500m and 1 group above 3000m. This will be improved in the revised manuscript.

21) Line 230-231. These seem to be the elevation bands mentioned in Line 212. I recommend the authors clarify why they include elevation bins down to 100 m elevation. It is my understanding that the study only considers CryoSat-2 LRM data which cover the high-elevation interior portion of the GrIS.

We made a wrong estimation of the lowest elevation within the CryoSat-2 LRM data coverage. This will be improved (as mentioned above) in the revised manuscript.

22) Line 243. I recommend the authors clarify the "Following ..." used to start this section. The previous two analyses described in Section 3.1 and 3.2 use to 25x25 km grid. The adoption of the 50x50 km grid here seems to be a marked departure from what has occurred previously as opposed to following/continuing.

The analyses before were performed to understand which regions are dominated by surface scattering and which regions by volume scattering, so that the following time series can be better interpreted. However, we agree that the logic of this sentence is weak. This will be removed in the revised manuscript.

23) Section 3.3. I recommend the authors clarify which months are included in their analysis of long-term variations. As it reads, it seems as though June-December LeW data are not represented (average is derived between January and May, Lines 243-244). What motivates this choice and why are Fall/early winter data not considered? If the goal is to avoid melt being present in the snow, would focusing on the full non-melt season (e.g., Oct.-Apr.) be more appropriate as opposed to following calendar years?

The goal is indeed to avoid melt being present in the snow, therefore the analysis will be changed to the full non-melt season in the revised manuscript.

24) Lines 261-263. I recommend the authors be more specific on where on the GrIS they are referring to. Are the number they state representative of the ice sheet as a whole or only a portion of it?

It is true that the observation represents only a portion of the GrIS. This will be better clarified in the revised manuscript.

25) Figure 4. I suggest the authors be specific with the LeW time periods behind the data presented here. Do they match the time periods shown at the top of the plot or are they those outlined in Section 3.3?

They match the time periods at the top of each column. This will be clarified in the revised manuscript.

26) Line 266. I have a hard time following the logic behind this statement because there isn't a really clear statement of how/why LeW is sensitive to volume scattering. Is increased volume scattering expected to increase or decrease LeW? Figure 1 would imply a positive correlation but, to me, here it seems to imply the opposite (reduced scattering (implying reduced LeW) due to subsurface high-density layers).

Following Nilsson et al. (2016) and Fig. 1, the melt events result in the formation of subsurface ice lenses, which reduces the radar penetration hence volume scattering; the LeW in turn reduces (e.g. the LeW at Pixel A reduces from 6.05 m to 3.21 m, according to Fig. 1). Therefore, it is correct that the reduced volume scattering implies a reduced LeW. This will be better clarified in the revised manuscript.

27) Figure 6. I recommend the authors consider including select representative 2D histograms directly comparing dh and LeW in addition to the correlation coefficient maps. I think this would give a sense on if the data are clustered or the range over which they co-vary against one another.

The following figure will be added in the revised manuscript, where the point density distribution is estimated using Gaussian kernel estimation (Węglarczyk, 2018).

[Figure]

Figure R2. Scatterplot between LeW and $\Delta h$. The point density distribution is shown in colours.

28) Line 291. Could the authors expand on this point and elaborate on how surface scattering effects the LeW/dh correlation? Is it because the OCOG retracker becomes less sensitive in rough areas?

As Referee 2 also pointed out, our original method to compute LeW was not sufficiently robust, as it directly searches for the peak of the normalised waveforms. We have improved the method to use the OCOG amplitude as the maximum amplitude, and define the bins between 0.05 and 0.95 thresholds as the LeW. By improving the method, the overall correlation increased from on average 0.3 to on average 0.7.

Regarding the specific regions where the correlation coefficients are generally lower than 0.5, they are typically characterised by more undulating topography close to the coastal line of Greenland or the southern regions with more recurrent melting. We present an example of the time series of LeW versus dh in Fig. R2. Two pixels in the 10km x 10km resolution are chosen for the visualisation. The pixel in the north shows a matching trend between LeW and dh, while the pixel in the south only shows a match partly, with a large standard deviation of both LeW and dh values.

[Figure]

Figure R3. Example time series of LeW (blue) and dh (red) for two locations. Shaded areas show the standard deviation of the inspected parameters within the pixel.

29) Line 293. I'd recommend the authors be very careful with the statement that penetration depth increases because LeW increases. LeW is an interpretation of an observed signal. If there was no volume scattering in the subsurface, the signal would penetrate as just deeply but no reflected power would exist at that point in the waveform, so LeW would only be a function of surface roughness. The depth to which it is possible for a radar to say something about the subsurface is a function of both how radar is designed and operated (e.g., transmit power, noise levels, data processing) as well as the structure and makeup of the surface and subsurface. All of these would affect at what point the SNR of a reflection from the subsurface would reach 0 dB. In light of this, would it be a more appropriate/accurate option to use "radar-laser height offsets" as opposed to "penetration depths"?

We agree. The concept of "penetration depths" will be replaced with "radar-laser height offsets" in the revised manuscript.

30) Line 314. I am confused by the statement here of a notable recovery in firn conditions and what is on Line 266 where the authors state firn recovery is not reflected. I recommend the authors clarify the distinction/difference between these two seemingly conflicting results.

Line 266 was an imprecise phrasing. We meant to say that dz was showing an abrupt recovery between 2013 and 2014, while the LeW recovery was more gradual. This will be improved in the revised manuscript.

31) A general comment on the Figures, but I'd ask the authors to consider using different colormaps for different variables. The same red-to-blue colormap is used in Figures 4, 6, 7, 8, 9, and 10 even though the variable being plotted changes; sometimes an absolute value is shown and sometimes a difference. I would also recommend that when presenting data on a map, the authors label their colorbars to make it explicit what variable is being shown.

The colormaps will be differed and the colorbars will be labelled in the revised manuscript.

32) Lines 338-339. I recommend the authors provide more explanation regarding why regular, annual melt-refreeze cycles are less impactful on volume scattering compared to more intermittent events.

Lines 338-339 particularly discusses the LeW variations in the southern part of Greenland. Here, the snow deposition rate is higher than the other regions of Greenland, as shown in Fig. A1 of the original manuscript. We will better explain this in the revised manuscript.

33) In the Discussion section, the authors devote the first paragraph to contrasting their results against those of Rutishauser et al. (2024). The authors compare the results in terms of their spatial patterns, but I would also suggest the authors consider the nature of the underlying radar measurements as well. The OIB MCoRDS radar operates in a much different frequency range compared to the CryoSat-2 SIRAL altimeter. What affect will that have on the resulting data and interpretations that could be assumed to be responding to more or less the same near-surface stratigraphy?

The CryoSat-2 SIRAL altimeter operates in a different manner from the OIB MCoRDS radar. While we used dh and LeW to indicate the terrain and part of the firn layer that have an impact on radar altimeter's waveforms, the Rutishauser et al. (2024) study tracks the peak of the the reflected radar signal. Therefore, in the Rutishauser et al. (2024) study, a perfectly dry-snow condition results in dz=0, indicating the radar reflection from the air-firn interface, while in our study, dry snow results in dh>0, indicating the height offset between laser and radar due to radar penetration. With the formation of an ice lens, dz from the Rutishauser et al. (2024) study increases, as another strong sub-surface reflector is detected, while in our study, dh and LeW immediately drop due to the reduction of Ku-band penetration ability. We will elaborate it better in the revised manuscript.

34) Also in the Discussion section, I would also suggest the authors be more specific with what they expect can be gained from integrating radar measurements at other frequencies (Lines 413- 415)? MCoRDS data are substantially different from CryoSat-2 but, as outlined in the previous comment, frequency-dependent impacts are not discussed. How can improved results in complex surface and volume scattering areas be improved by adopting more frequencies? At the same time, I'd ask the authors to consider what this means for future dual-frequency radar altimeters such as ESA CRISTAL which will operate Ku- and Ka-band altimeters simultaneously.

We appreciate the detailed recommendations of the referee, and will elaborate on this point in the revised manuscript.

35) Figure 10. I recommend the authors elaborate more on the specific elevation intervals presented in 10b, 10d, and 10e. Why are these specific intervals chosen and what additional information do they add? Thes subplots and the specific elevation intervals, not only deviate

from every plot that has been shown so far but they are not even mentioned in the main text. What overall purpose do they serve?

They were originally randomly sampled among the 10 elevation groups with the aim to show separate time series in terms of curves with corresponding standard deviations. However, we understand that this can cause confusion, and will improve the illustration in the revised manuscript.

36) Line 433. Here "sub-surface" appears with a hyphen, while through the rest of the manuscript it is written as "subsurface".

This will be corrected in the revised manuscript as "subsurface".

36) Line 441. "stratigraphy" is misspelt in the Code and data availability section.

This will be corrected in the revised manuscript.

**Reference**

Benson, C. S.: Stratigraphic Studies in the Snow and Firn of the Greenland Ice Sheet. Dissertation (Ph.D.), California Institute of Technology. doi:10.7907/G7V2-0T57. https://resolver.caltech.edu/CaltechETD:etd-03232006-104828, 1960.

Lacroix, P., Dechambre, M., Legrésy, B., Blarel, F., and Rémy, F.: On the use of the dual-frequency ENVISAT altimeter to determine snowpack properties of the Antarctic ice sheet, Remote Sensing of Environment, 112, 1712–1729, https://doi.org/10.1016/j.rse.2007.08.022, 2008.

Li, W., Slobbe, C., and Lhermitte, S.: A leading-edge-based method for correction of slope-induced errors in ice-sheet heights derived from radar altimetry, The Cryosphere, 16, 2225–2243, https://doi.org/10.5194/tc-16-2225-2022, 2022.

Michel, A., Flament, T., and Rémy, F.: Study of the Penetration Bias of ENVISAT Altimeter Observations over Antarctica in Comparison to ICESat Observations, Remote Sensing, 6, 9412–9434, https://doi.org/10.3390/rs6109412, 2014.

Nilsson, J., Vallelonga, P., Simonsen, S. B., Sørensen, L. S., Forsberg, R., Dahl-Jensen, D., Hirabayashi, M., Goto-Azuma, K., Hvidberg, C. S., Kjaer, H. A., and Satow, K.: Greenland 2012 melt event effects on CryoSat-2 radar altimetry, Geophysical Research Letters, 42, 3919–3926, https://doi.org/10.1002/2015gl063296, 2015.

Otosaka, I. N., Shepherd, A., Casal, T. G. D., Coccia, A., Davidson, M., Di Bella, A., Fettweis, X., Forsberg, R., Helm, V., Hogg, A. E., Hvidegaard, S. M., Lemos, A., Macedo, K., Kuipers Munneke, P., Parrinello, T., Simonsen, S. B., Skourup, H., and Sørensen, L. S.: Surface Melting Drives Fluctuations in Airborne Radar Penetration in West Central Greenland, Geophysical Research Letters, 47, https://doi.org/10.1029/2020gl088293, 2020.

Porter, C., Morin, P., Howat, I., Noh, M.-J., Bates, B., Peterman, K., Keesey, S., Schlenk, M., Gardiner, J., Tomko, K., Willis, M., Kelleher, C., Cloutier, M., Husby, E., Foga, S., Nakamura, H., Platson, M., Wethington Jr., M., Williamson, C., Bauer, G., Enos, J., Arnold, G., Kramer, W.,

Becker, P., Doshi, A., D'Souza, C., Cummens, P., Laurier, F., and Bojesen, M.: ArcticDEM, Harvard Dataverse, https://doi.org/10.7910/DVN/OHHUKH, 2018.

Rutishauser, A., Scanlan, K. M., Vandecrux, B., Karlsson, N. B., Jullien, N., Ahlstrøm, A. P., Fausto, R. S., and How, P.: Mapping the vertical heterogeneity of Greenland's firn from 2011–2019 using airborne radar and laser altimetry, The Cryosphere, 18, 2455–2472, https://doi.org/10.5194/tc-18-2455-2024, 2024.

Tran, N., Remy, F., Feng, H., and Femenias, P.: Snow Facies Over Ice Sheets Derived From Envisat Active and Passive Observations, IEEE Transactions on Geoscience and Remote Sensing, 46, 3694–3708, https://doi.org/10.1109/tgrs.2008.2000818, 2008.

Węglarczyk, S.: Kernel density estimation and its application, ITM Web of Conferences, 23, 00037, https://doi.org/10.1051/itmconf/20182300037, 2018.

---

## Author Response (AR1)

**Response to Referees on egusphere-2024-3251**
We appreciate both the referees and the editor for the support for improving out manuscript. Please find the response to Referee 1 from page 1 to page 16, and the response to Referee 2 from page 17 to page 25.

**Response to Referee 1 on egusphere-2024-3251**
First, we would like to thank the Referee for reviewing and commenting on the manuscript, which will improve the quality of the manuscript. Please find the item-by-item reply below, with the responses in blue. All the suggested changes will be implemented in the revised text that will be uploaded.

**General Comments**
This manuscript presents an interesting study into using the leading edge width (LeW) of CryoSat-2 measurements between 2011 and 2021 to investigate the long-term characteristics of Greenland firn conditions. The authors begin their manuscript with an introduction to the importance of understanding melt events, their effects across the Greenland Ice Sheet (GrIS) and how they intend to approach the problem using remote sensing data supported by in situ measurements and numerical climate model outputs. In Section 2, the authors present the various datasets considered in their study. Section 3 then outlines how the data are used, combined and the types of analyses the authors perform; the results of which are presented in Section 4. Finally, in Section 5, the authors reflect on the implications of the results and how future studies could expand on them, before the main conclusions are outlined in Section 6.

Overall, the authors have analyzed and presented a substantial amount of data. They take a very thorough approach to assessing the long-term spatial patterns in CryoSat-2 LeW by incorporating another satellite altimeter (i.e., ICESat-2), other derived satellite and airborne datasets (i.e., roughness, radar-laser offsets, topography), in-situ measurements (i.e., densities), and model results (i.e., densities, meltwater content, and firn air content). The scope of bringing all the data together is impressive and I think very exciting direction of work. Relatedly however, presenting so much data requires a very clear narrative and defined structure to help a general and non-specialized reader avoid being lost in all the details. My main comment on the manuscript is that I found this aspect of it to be underdeveloped, which could hamper the impact of the results.

While reading through study and going through all the results, I found it hard at times to get a sense of how all the different pieces fit together. I think the manuscript would benefit from a clearer statement of the central hypothesis and the physical reasoning behind it that would then frame the overall study. In the Introduction, the authors state the importance of melt and refreeze events and postulate that LeW could be used to study long-term patterns. What I think is missing though is how LeW is affected by melting/refreezing events. What is changing in the firn and how does that affect the radar signals? Concepts of surface and volume scattering as well as refrozen layers appear repeatedly later on in the manuscript, but I think explicit descriptions of what the authors mean by these, their linkage to the physical state of the firn and what that means for the CryoSat- 2

would substantially help fortify the overall narrative. This bridging between radar theory and more classical glaciological concepts will also strengthen the impact the manuscript will have by really outlining how all these pieces fit together and what the results mean. Some of the specific comments below will also be in this direction.

We appreciate the general comments. We have added the following theoretical descriptions from Line 31 onwards in the revised manuscript:

*"The underlying principle is based on the fact that over firn-covered regions of the ice sheet—primarily at higher elevations—radar pulses at frequencies commonly used in satellite altimetry penetrate into the firn (e.g., Ridley and Partington, 1988). According to Ridley and Partington (1988) and Davis and Zwally (1993), the penetration depth may range from a few centimetres to several metres. Consequently, recorded waveforms contain signals from both surface scattering and volume (or sub-surface) scattering caused by inhomogeneities within the underlying firn layers. Surface scattering dominates the start of the waveform, while volume scattering becomes predominant beyond the inflection point, where the illuminated surface area becomes constant. The rise of the backscattered power from volume scattering depends on firn parameters, including firn density, firn air content (FAC), and grain size (Ridley and Partington, 1988; Vandecrux et al., 2019; Brils et al., 2022). For example, larger grain radii and higher firn densities lead to a faster increase in backscattered power (i.e., a steeper leading edge) (Ridley and Partington, 1988, Figs. 9, 10). Melt and refreezing events alter firn parameters and form refrozen layers, modifying scattering behaviour and waveform shape. Depending on thickness and density, refrozen layers can substantially reduce radar penetration, diminishing volume scattering. While these changes in waveform shape are observable, attributing them to variations in volume scattering—and thus to changes in firn properties—requires distinguishing them from variations in surface scattering, particularly those driven by surface roughness. Indeed, a decrease in surface roughness also results in steeper leading edges (Ridley and Partington, 1988, Fig. 8)."*

We also noticed that, directly aiming at "volume scattering" in the title is confusing, as we have not been able to quantitatively distinguish which part of the waveform signal is actually affected by volume scattering and which part by surface scattering. Therefore, we have also changed the title into "Assessing spatio-temporal variability of melt–refreeze patterns in firn over Greenland with CryoSat-2", hoping to be more specific.

To improve readability of the manuscript, we also shifted the Methods and Results sections, to focus on:
- The yearly spatio-temporal variations in LeW, visualised in maps;
- The monthly spatio-temporal variations in LeW, interpreted with the surface roughness datasets;
- The correlation between LeW and laser-radar elevation biases, aiming to link temporal variations in LeW with temporal variations in volume scattering;
- The monthly temporal variations in LeW, interpreted with the help of modelled densities and firn air content (FAC).

I hope the authors find the following comments constructive as they work towards revising their manuscript.

**Specific Comments**

1) As the manuscript primarily centers on CryoSat-2 LeW results, I would suggest the authors consider revising the title to "Assessing spatio-temporal variability of firn scattering over Greenland with CryoSat-2". I understand that the inclusion of ICESat-2 data makes a case for multiple altimeters, but my impression is that the ICESat-2 data are more complementary to the main CryoSat-2 dataset. In a similar way to the MAR and IMAU climate model data, ICESat-2 data appear to be used more to help explain trends in the CryoSAt-2 data, not necessarily as the primary data source themselves.

This has been implemented in the revised manuscript.

2) Line 57 "The LeW is adopted as it is sensitive to volume scattering ..." Line 67 "... we have to understand both volume scattering and surface scattering ..." These are two instances where a more explicit statement of what the authors mean by volume and surface scattering could help improve the overall framing of the study. How/why LeW is sensitive to these two concepts and what on the surface and in the firn contributes to them? I think it would broaden the reach of the manuscript by removing the hurdle of needing to be familiar with nuanced radar theory concepts and motivate exactly why the specific model outputs are chosen for comparison with the LeW results in the latter stages of the manuscript.

We agree and have re-organised the paragraph (and the previous paragraphs) as follows (now it becomes Line 47 onwards):

*"Several studies have employed waveform shape parameters to gain insight into the impact of melt and refreezing on Greenland's firn layer, as well as to estimate the bias in radar altimeter-derived elevations caused by radar penetration into the firn. Nilsson et al. (2015), for example, used Ku-band (13.575 GHz) altimeter data acquired by CryoSat-2 to track the formation of ice lenses following melt events. Simonsen and Sørensen (2017) explored the same data to investigate the impact of volume scattering properties on elevation estimates. Both Nilsson et al. (2015) and Simonsen and Sørensen (2017) showed that a large leading edge width (LeW) is indicative of volume scattering of the signal in the upper parts of the firn, while Nilsson et al. (2015) in particular observed the impact of the 2012 Greenland melt event and its subsequent refreezing on waveform-derived parameters, including LeW, trailing edge slope (TeS) and peakiness, backscatter intensity, and height. The Simonsen and Sørensen (2017) study indicated that within the region of the Greenland Ice Sheet covered by Low Resolution Mode (LRM) data (i.e., the LRM zone), LeW could be effectively used to correct for elevation biases caused by volume scattering. In addition to waveform shape parameters, other radar altimeter-derived variables have also been utilised to infer firn properties. For instance, Scanlan et al. (2023) leveraged surface echo powers in Ku-band CryoSat-2 and Ka-band SARAL radar waveforms to derive monthly maps of Greenland Ice Sheet's wavelength-scale surface roughness and density between January 2013 and December 2018. Furthermore, several studies have estimated radar penetration depths by combining radar and laser altimetry data. Michel et al. (2014), for example,*

*analysed the differences between radar (ENVISAT) and laser (ICESat) altimeter heights over Antarctica to derive Ku-band radar penetration biases into firn and compared these height differences with LeWs. The study provides insights into opportunities for similar approaches to study Greenland's firn.*

*Despite these advances in using radar altimetry to monitor Greenland's firn properties—particularly in assessing the effects of melt and refreezing—existing studies have largely been confined to a period without extensive melt (e.g., January 2013 to December 2018; Scanlan et al., 2023), or to the short timeframe immediately following the 2012 melt event (e.g., up to 2014; Nilsson et al., 2015). The impact on the long term, especially following the 2019 melt (Tedesco and Fettweis, 2020), remain insufficiently monitored. The availability of more than a decade of CryoSat-2 data presents a valuable opportunity to address this gap."*

With this revision, we also hope to show that LeW is the main focus of this study, and the ICESat-2 data are more complementary, following the suggestion of the Referee's comment 1.

3) The authors dedicate Lines 35-51 motivating CryoSat-2 and LeW as a metric for studying firn. I recommend the authors consider expanding more clearly on the motivations for using the other datasets (e.g., ICESat-2, in-situ densities, dz, roughness, topography, and model results) to help explain the LeW results. What aspect of the LeW signal are these datasets being used to interpret? I found Lines 52-69 to be confusing as it was not always clear how these different datasets all supported the LeW analysis.

We understand and appreciate the recommendation. Following the revision of the previous comment, which already indicated the use of roughness dataset, topography and ICESat-2 data, we re-wrote the paragraph 52—69 (now it becomes Line 70 onwards) as:

*"The main objective of this paper is to assess the impact of melt and refreezing events on the properties of Greenland's upper firn layer, using LeWs derived from CryoSat-2 radar waveforms acquired between 2010 and 2024. To support the interpretation of the results, we complement the assessment by comparing the LeWs with: (i) the surface roughness dataset derived by Scanlan et al. (2023) to analyse under which circumstances the LeW variation is dominated by surface scattering; (ii) the ArcticDEM standard deviation to assess the impact of macro-scale roughness due to topographic variation on LeW; (iii) penetration depths obtained by differencing ICESat-2 laser altimeter elevations and CryoSat-2 radar altimeter elevations to gain further insights into volume scattering; and (iv) firn densities and FAC from the Modèle Atmosphérique Régionale (MAR) (Fettweis et al., 2011, 2017; Lambin et al., 2022) and the Firn Densification Model from the Institute for Marine and Atmospheric research Utrecht (IMAU-FDM v1.2G; Brils et al., 2022) to analyse how spatial and temporal variations in firn properties affect the LeW and to improve the interpretation of radar altimeter scattering properties for future research."*

4) I recommend the authors consider reducing the number of adverbs (e.g., furthermore, finally, in addition, therefore, additionally, etc.) used to start sentences to make them more direct and impactful.

We appreciate the recommendation. Many of the adverbs have been removed in the revised manuscript.

5) To be more specific on the types of GrIS changes of interest in this study, I recommend the authors re-phrase Line 70 from "... assess long-term changes over the ..." to "... assess long-term surface changes over the ..."

We appreciate the suggestion of the referee. As mentioned in previous discussions, to also make it more specific, we have revised it into *"The main objective of this paper is to assess the impact of melt and refreezing events on the properties of Greenland's upper firn layer..."*.

6) In Section 2.1, I recommend the authors include more detail on the nature of the CryoSat-2 LRM data and what differentiate them from other CryoSat-2 data products (e.g., what is unique/different in their acquisition/data processing?).

The CryoSat-2 LRM data are available within the interior of Greenland, SARIn mode is available over the coastal areas, and SAR mode operates over the sea. This will be elaborated in the revised manuscript. However, for the data processing, since SAR and SARIn modes are not available in our region of interest (Greenland interior), it may not be relevant to add everything in detail. We have briefly added from Line 99 onwards:

*"CryoSat-2's primary payload, the SAR Interferometric Radar Altimeter (SIRAL), operates in three measurement modes (Wingham et al., 2006):*
*1. Low Resolution Mode (LRM): analogous to pulse-limited radar altimetry, this mode is used over relatively flat ice sheet regions and the open ocean.*
*2. Synthetic Aperture Radar (SAR) Mode: utilising coherent echo processing, this mode provides higher-resolution along-track data from sea ice and sea-ice-contaminated ocean surfaces.*
*3. SAR Interferometric (SARIn) Mode: combining coherent echo processing with interferometry, SARIn delivers high-resolution along-track data along with across-track echo directions, primarily for ice sheet margins."*

7) Line 89, please include the range resolution of CryoSat-2.

(Now Line 114) It should have been made more explicit that the "range resolution", $S_r$, should be $\frac{1}{2}c \bullet dt$, where $c$ is the speed of light and $dt$ is the waveform sampling interval (3.125ns) This has been improved in the revised manuscript (Eq. 2).

8) In Figure 1, I'd ask the authors to consider including the b0.99 and b0.01 values for each waveform as well as map (perhaps as an insert) of where these two locations are in Greenland are.

We have realised that the original Fig. 1 has little added value to the overall readability, hence have removed it from the revised manuscript.

9) Line 94, what high-resolution DEM model is used?

(Now Line 85) It should be the 100m resolution ArcticDEM mentioned in Section 2.3. Following the recommendation also from Referee 2, we have introduced the ArcticDEM at the beginning of the Data section (now Section 2.1).

10) In Line 97, the authors state that they used July measurements as indicative of post-melt conditions but there is no way for the reader to assess if melting has ceased at these locations by the time the data were acquired; especially knowing how extreme the melt extents observed in the summer of 2012 were. I would recommend the authors provide further support for this statement or consider using data from later in the year.

Following comment 8, the original Fig. 1 and the related description have been removed from the revised manuscript.

11) I am not sure I fully understand the context for why two different grid resolutions (50x50 and 25x25) are used. I suggest the authors clarify this point.

Originally, we aimed to use the finer resolution (25km x 25km) to calculate long-term (decadal) statistics and the lower resolution (50km x 50km) to calculate monthly statistics. However, following the recommendation of Referee 2, we noticed that using 10km x 10km resolution can still ensure sufficient (more than 10) data points per pixel per month, while being consistent with the spatial resolution of the firn models. Therefore, we adopt the recommendations of both referees and have used the 10km x 10km resolution throughout the revised manuscript.

12) Line 115. Do all CryoSat-2 measurements in a given month have a corresponding ICESat-2 measurement within 50 m or are their spatial gaps? I'd also recommend the authors provide their reasoning for choosing 50m when the footprint of CryoSat-2 LRM data is much larger.

(Now Line 140) It is true that not all CryoSat-2 measurements have a corresponding ICESat-2 measurement within the 50 m radius. As shown in Table 1 of Li et al. (2022), such a criterion results in approximately 30 times fewer measurements in year 2019.

We agree that the footprint of CryoSat-2 LRM mode is much larger. The motivation of using a smaller search criterion was that over the undulating terrain, the true footprint of CryoSat-2 LRM should be smaller than the theoretical one, therefore we would choose a corresponding ICESat-2 point as close as possible, yet not largely reducing the number of valid points. However, the selection of 50 m is rather arbitrary, therefore we have conducted a sensitivity analysis, shown in Fig. R1. It can be observed that as the search radius increases, the number of valid dh increases, while the correlation between dh and LeW decreases. Especially, when using 800 m as the search range, which is comparable to the theoretical pulse-limited footprint of CryoSat-2 LRM, the correlation coefficients are overall below 0.5. Using 100 m and using 50 m do not demonstrate distinct differences. Therefore, we prefer to choose a search range as small as possible, which is also similar to the crossover principle proposed by Michel et al. (2014).

[Figure]

Figure R1. Comparison of correlation coefficients between dh and LeW when using different search range for the corresponding ICESat-2 point for each CryoSat-2 point.

13) With how Sections 2, 3 and 4 are structured, the CryoSat-2/ICESat-2 results from Figure 2 and Lines 122-128 seem to be more suited to Section 4 than Section 2. I understand they are used again in Section 2.4, but could Section 2.4 be treated more abstractly by referring to a subsurface depth extent to be determined later? The current placement seems to interrupt the flow of describing all the individual datasets considered.

We appreciate the suggestion of the referee, and this has been re-structured in the revised manuscript (Section 3.3).

14) Lines 134-136. I recommend the authors elaborate a bit more on how "computational efficiency" necessitates using both the 100m and 1km ArcticDEMs in these two instances. What about these specific applications makes the use of two different DEMs more efficient?

The use of the 100m ArcticDEM is inherited from Li et al. (2022): in principle, ArcticDEM can be available with the resolution of 2m (Porter et al., 2018). However, since our computation is performed in MATLAB, it is difficult to load the large ArcticDEM Geotiff files. Therefore, the finest resolution considered in our studies is 100m. Now we have clarified this in Line 91:

*"The model is available at various resolutions, ranging from 2 m to 1 km (Porter et al., 2023). Consistent with Li et al. (2022), we employ the 100 m resolution ArcticDEM for slope-induced error correction in CryoSat-2 elevation estimates, balancing accuracy and computational efficiency compared to the higher-resolution 2 m dataset."*

We admit that the "computationtal efficiency" that comes with the 1km ArcticDEM can be confusing. Following both referees' comments, we have used a 10km-by-10km grid over the interior Greenland in the revised manuscript. When we used the 100m-resolution ArcticDEM to compute the standard deviation of each grid cell, it

again resulted in an overload of computational power, therefore we had to use the 1km-resolution ArcticDEM.

15) Figures 2, 3, 4, 6, 7, and 9. I recommend the authors elaborate why they used a DEM from Helm et al. (2014a, b) as their basemap instead of one of the ArcticDEMs they use in their analysis. Also, I'd recommend including a colorbar for the elevations the first time it is used.
The only reason is that the Helm et al. (2014) DEM focuses on Greenland, while ArcticDEM covers the entire Arctic including the ocean surrounding Greenland and is difficult to crop. We have edited the ArcticDEM in the revised manuscript for better consistency. The colorbar is also shown in Figs. 1 and 2.

16) Line 146. I recommend the authors clarify how the weights are determined in their weighted average densities.
(Now Line 56) The weights are defined as the thickness of each layer. This has been elaborated in the revised manuscript as:

*"We use the time series of modelled firn density profiles to compute the weighted average density of the upper firn column, from the surface to the max Ku-band radar penetration depth. Thickness is used as a weighting factor to account for the model's uneven vertical resolution."*

17) I recommend the authors consider better motivating the inclusion of the IMAU FAC. FAC is a column-integrated measurement (Line 168) whereas LeW derived from CryoSat-2 is seemingly only sensitive to the upper few meters (Figure 2). Why would these two datasets derived over different depth ranges be considered comparable?
We agree with the concern of the referee. In general, we aim to use the density dataset at upper 1.5m to prove that CryoSat-2 is indeed sensitive to the changes that happen within the 1.5m firn layer. On the other hand, as we also try to learn about the overall condition of the firn (beyond this 1.5m threshold), the FAC over the entire snowpack is used as additional information to indicate whether Greenland firn experiences a continuous decrease in pore spaces. We have added in Line 181:

*"The FAC represents the vertically integrated porosity of the firn (Kuipers Munneke et al., 2015), expressed in metres. It is computed over the entire firn column and serves as a measure of total firn porosity, indicating the firn's capacity to retain meltwater (Vandecrux et al., 2019). Although CryoSat-2 signals primarily penetrate the upper firn layers, we leverage the modelled FAC time series to assess whether the observed melt–refreeze patterns notably influence broader firn conditions."*

However, from our experience, variability in FAC is mainly a result of changes in the upper meters of firn.

18) Line 168. The "but" in "... 1.5 m but the FAC ..." can be removed.
(Now Line 181) This has been rewritten as above in the revised manuscript.

19) I recommend the authors expand on why these particular in-situ firn density measurements are used instead of the more comprehensive SUMup dataset (i.e., Vandecrux et al. 2023)? Furthermore, why is it necessary for firn density profiles to contain the 2012 melt year (Line 185)?

*Vandecrux, B. et al. The SUMup collaborative database: Surface mass balance, subsurface temperature and density measurements from the Greenland and Antarctic ice sheets (1912- 2023). Arctic Data Center https://doi.org/10.18739/A2M61BR5M (2023).*

We appreciate the referee for the suggestion and have included this dataset in the revised manuscript (the Schaller et al., 2016 and Otosaka et al., 2020 datasets are already included in the SUMup database).

The main reason to contain the 2012 melt year is to provide more sound evidence that the 2012 melt results in a visible density increase, which can also be observed in the modelled firn densities. This high-density layer is buried in the subsequent years, therefore the recovery in LeW can be eventually observed. This motivation has been added to Line 196 of the revised manuscript.

20) Line 212. These 10 DEM elevation groups have not been mentioned yet, so I do not follow how they can be "aforementioned". I recommend the authors clarify this statement.

It should be 8 groups equally divided between 1500m and 3000m, 1 group below 1500m and 1 group above 3000m. This has been moved to Line 87, Section 2.1 (together with the introduction to ArcticDEM) of the revised manuscript.

21) Line 230-231. These seem to be the elevation bands mentioned in Line 212. I recommend the authors clarify why they include elevation bins down to 100 m elevation. It is my understanding that the study only considers CryoSat-2 LRM data which cover the high-elevation interior portion of the GrIS.

(Now Line 227) We made a wrong estimation of the lowest elevation within the CryoSat-2 LRM data coverage. This have been improved (as mentioned above) in the revised manuscript.

22) Line 243. I recommend the authors clarify the "Following ..." used to start this section. The previous two analyses described in Section 3.1 and 3.2 use to 25x25 km grid. The adoption of the 50x50 km grid here seems to be a marked departure from what has occurred previously as opposed to following/continuing.

The analyses before were performed to understand which regions are dominated by surface scattering and which regions by volume scattering, so that the following time series can be better interpreted. However, we agree that the logic of this sentence is weak. This sentence has been removed in the revised manuscript (and now we use the 10x10 km grid everywhere).

23) Section 3.3. I recommend the authors clarify which months are included in their analysis of long-term variations. As it reads, it seems as though June-December LeW data are not represented (average is derived between January and May, Lines 243-244). What motivates this choice and why are Fall/early winter data not considered? If the goal is to avoid melt being present in the snow, would focusing

on the full non-melt season (e.g., Oct.-Apr.) be more appropriate as opposed to following calendar years?

The goal is indeed to avoid melt being present in the snow, therefore the analysis has been changed to the full non-melt season (Oct.-Apr.) in the revised manuscript.

24) Lines 261-263. I recommend the authors be more specific on where on the GrIS they are referring to. Are the number they state representative of the ice sheet as a whole or only a portion of it?

It is true that the observation represents only a portion of the GrIS. We have added *"Within the coverage of CryoSat-2 LRM data"* in contents such as Line 251 of the revised manuscript.

25) Figure 4. I suggest the authors be specific with the LeW time periods behind the data presented here. Do they match the time periods shown at the top of the plot or are they those outlined in Section 3.3?

Figure 4 has been removed from the revised manuscript.

26) Line 266. I have a hard time following the logic behind this statement because there isn't a really clear statement of how/why LeW is sensitive to volume scattering. Is increased volume scattering expected to increase or decrease LeW? Figure 1 would imply a positive correlation but, to me, here it seems to imply the opposite (reduced scattering (implying reduced LeW) due to subsurface high-density layers).

Following Nilsson et al. (2016) and Ronan et al. (2024), the melt events result in the formation of subsurface ice lenses, which reduces the radar penetration hence volume scattering; the LeW in turn reduces. Therefore, it is correct that the reduced volume scattering implies a reduced LeW. We agree that the comparison against the Rutishauser et al. (2024) study causes confusion and have removed it from the revised manuscript.

27) Figure 6. I recommend the authors consider including select representative 2D histograms directly comparing dh and LeW in addition to the correlation coefficient maps. I think this would give a sense on if the data are clustered or the range over which they co-vary against one another.

The following figure has been added in the revised manuscript (Fig. 5c), where the point density distribution is estimated using Gaussian kernel estimation (Węglarczyk, 2018).

[Figure]

Figure R2. Scatterplot between LeW and $\Delta h$. The point density distribution is shown in colours.

28) Line 291. Could the authors expand on this point and elaborate on how surface scattering effects the LeW/dh correlation? Is it because the OCOG retracker becomes less sensitive in rough areas?
As Referee 2 also pointed out, our original method to compute LeW was not sufficiently robust, as it directly searches for the peak of the normalised waveforms. We have improved the method to use the OCOG amplitude as the maximum amplitude, and define the bins between 0.05 and 0.95 thresholds as the LeW. By improving the method, the overall correlation increased from on average 0.3 to on average 0.6.

Regarding the specific regions where the correlation coefficients are generally lower than 0.5, they are typically characterised by more undulating topography close to the coastal line of Greenland or the southern regions with more recurrent melting. We present an example of the time series of LeW versus dh in Fig. R2. Two pixels in the 10km x 10km resolution are chosen for the visualisation. The pixel in the north shows a matching trend between LeW and dh, while the pixel in the south only shows a match partly, with a large standard deviation of both LeW and dh values.

[Figure]

Figure R3. Example time series of LeW (blue) and dh (red) for two locations. Shaded areas show the standard deviation of the inspected parameters within the pixel.

The figure shows that towards the coast of Greenland, the distribution of LeW and dh is not as uniform as in the interior, exhibiting large uncertainties. This can be due to the uncertainties in LeW, in ICESat-2 height measurements, as well as in CryoSat-2 height estimates. We have added the following explanation in Line 330:

*"The lower correlation and significance towards the margins can likely be attributed to the compromised performance of ICESat-2 elevation measurements due to the large slopes, rough surfaces (Smith et al., 2023c) and scattering biases in the low-elevation regions (Smith et al., 2018). These biases can be propagated to the derived Δh, which may not properly indicate volume scattering variations."*

29) Line 293. I'd recommend the authors be very careful with the statement that penetration depth increases because LeW increases. LeW is an interpretation of an observed signal. If there was no volume scattering in the subsurface, the signal would penetrate as just deeply but no reflected power would exist at that point in the waveform, so LeW would only be a function of surface roughness. The depth to which it is possible for a radar to say something about the subsurface is a function of both how radar is designed and operated (e.g., transmit power, noise levels, data processing) as well as the structure and makeup of the surface and subsurface. All of these would affect at what point the SNR of a reflection from the subsurface

would reach 0 dB. In light of this, would it be a more appropriate/accurate option to use "radar-laser height offsets" as opposed to "penetration depths"?

(Now Line 334) We agree. The concept of "penetration depths" has been replaced with "laser-radar height offsets" (because laser height measurements are higher) in or "a proxy for Ku-band radar penetration depth" the revised manuscript.

30) Line 314. I am confused by the statement here of a notable recovery in firn conditions and what is on Line 266 where the authors state firn recovery is not reflected. I recommend the authors clarify the distinction/difference between these two seemingly conflicting results.

Line 266 was an imprecise phrasing. We meant to say that dz was showing an abrupt recovery between 2013 and 2014, while the LeW recovery was more gradual. This has been removed from the revised manuscript (shown in the response to comment 26).

31) A general comment on the Figures, but I'd ask the authors to consider using different colormaps for different variables. The same red-to-blue colormap is used in Figures 4, 6, 7, 8, 9, and 10 even though the variable being plotted changes; sometimes an absolute value is shown and sometimes a difference. I would also recommend that when presenting data on a map, the authors label their colorbars to make it explicit what variable is being shown.

The colormaps have been differed and the colorbars are labelled in the maps in the revised manuscript except for Figs. 5a and 5b, where the names of the variables are too long and are therefore added to the titles.

32) Lines 338-339. I recommend the authors provide more explanation regarding why regular, annual melt-refreeze cycles are less impactful on volume scattering compared to more intermittent events.

Lines 338-339 (now Lines 347-351) particularly discusses the LeW variations in the southern part of Greenland. Here, the snow deposition rate is higher than the other regions of Greenland, as shown in Fig. A1 of the original manuscript. We have added this explanation in the revised manuscript.

33) In the Discussion section, the authors devote the first paragraph to contrasting their results against those of Rutishauser et al. (2024). The authors compare the results in terms of their spatial patterns, but I would also suggest the authors consider the nature of the underlying radar measurements as well. The OIB MCoRDS radar operates in a much different frequency range compared to the CryoSat-2 SIRAL altimeter. What affect will that have on the resulting data and interpretations that could be assumed to be responding to more or less the same near-surface stratigraphy?

OIB MCoRDS does indeed operate in a much lower frequency, but for our study, the main difference lies primarily in how the radar waveforms are analysed. While we used dh and LeW to indicate the terrain and part of the firn layer that have an impact on radar altimeter's waveforms, the Rutishauser et al. (2024) study tracks the peak of the the reflected radar signal. Therefore, in the Rutishauser et al. (2024) study, a perfectly dry-snow condition results in dz=0, indicating the radar reflection from the air-firn interface, while in our study, dry snow results in dh>0, indicating the

height offset between laser and radar due to radar penetration. With the formation of an ice lens, dz from the Rutishauser et al. (2024) study increases, as another strong sub-surface reflector is detected, while in our study, dh and LeW immediately drop due to the reduction of Ku-band penetration ability.

Now we realised that this comparison can really cause confusion, especially because the main conclusion from this comparison is that "CryoSat-2 has a better spatial and temporal continuity than OIB". Therefore, all the comparison with dz has been removed from the revised manuscript.

34) Also in the Discussion section, I would also suggest the authors be more specific with what they expect can be gained from integrating radar measurements at other frequencies (Lines 413- 415)? MCoRDS data are substantially different from CryoSat-2 but, as outlined in the previous comment, frequency-dependent impacts are not discussed. How can improved results in complex surface and volume scattering areas be improved by adopting more frequencies? At the same time, I'd ask the authors to consider what this means for future dual-frequency radar altimeters such as ESA CRISTAL which will operate Ku- and Ka-band altimeters simultaneously.
We appreciate the detailed recommendations of the referee, and have elaborated on this point in the revised manuscript (Line 429 onwards):

*"According to Lacroix et al. (2008) who compared waveform parameters from S-band and Ku-band radar altimeters, the impact of surface scattering as well as from snow grain size decreases with an increasing radar frequency. According to Scanlan et al. (2023) who derived firn properties using both Ku-band and Ka-band radar altimeters, radar altimeters operating in a lower frequency are sensitive to firn densities at a larger depth. For future dual-frequency radar altimeters, e.g. the Copernicus Polar Ice and Snow Topography Altimeter (CRISTAL) mission which operates in both Ku- and Ka-bands, the different penetration abilities and sensitivities to firn properties offer the potential of a multi-layered analysis approach. For a higher frequency such as Ka-band, the penetration depth is smaller, hence we expect a quicker recovery of LeW after a melt event than that of Ku-band. This different recovery rate can help future studies to locate the subsurface refrozen layers and derive accumulation rate."*

35) Figure 10. I recommend the authors elaborate more on the specific elevation intervals presented in 10b, 10d, and 10e. Why are these specific intervals chosen and what additional information do they add? Thes subplots and the specific elevation intervals, not only deviate from every plot that has been shown so far but they are not even mentioned in the main text. What overall purpose do they serve?
(Now Fig. 8) They were originally randomly sampled among the 10 elevation groups with the aim to show separate time series in terms of curves with corresponding standard deviations. However, we understand that this can cause confusion, and have removed the curves from the revised manuscript.

36) Line 433. Here "sub-surface" appears with a hyphen, while through the rest of the manuscript it is written as "subsurface".

(Now Line 461) This has been corrected in the revised manuscript as "subsurface".

36) Line 441. "stratigraphy" is misspelt in the Code and data availability section.
This comparison has been removed from the revised manuscript.

**Reference**

Benson, C. S.: Stratigraphic Studies in the Snow and Firn of the Greenland Ice Sheet. Dissertation (Ph.D.), California Institute of Technology. doi:10.7907/G7V2-0T57. https://resolver.caltech.edu/CaltechETD:etd-03232006-104828, 1960.

Brils, M., Kuipers Munneke, P., van de Berg, W. J., and van den Broeke, M.: Improved representation of the contemporary Greenland ice sheet firn layer by IMAU-FDM v1.2G, Geoscientific Model Development, 15, 7121–7138, https://doi.org/10.5194/gmd-15-7121-2022, 2022.

Davis, C. H. and Zwally, H. J.: Geographic and seasonal variations in the surface properties of the ice sheets by satellite-radar altimetry, Journal of Glaciology, 39, 687–697, https://doi.org/10.3189/S0022143000016580, 1993.

Lacroix, P., Dechambre, M., Legrésy, B., Blarel, F., and Rémy, F.: On the use of the dual-frequency ENVISAT altimeter to determine snowpack properties of the Antarctic ice sheet, Remote Sensing of Environment, 112, 1712–1729, https://doi.org/10.1016/j.rse.2007.08.022, 2008.

Li, W., Slobbe, C., and Lhermitte, S.: A leading-edge-based method for correction of slope-induced errors in ice-sheet heights derived from radar altimetry, The Cryosphere, 16, 2225–2243, https://doi.org/10.5194/tc-16-2225-2022, 2022.

Michel, A., Flament, T., and Rémy, F.: Study of the Penetration Bias of ENVISAT Altimeter Observations over Antarctica in Comparison to ICESat Observations, Remote Sensing, 6, 9412–9434, https://doi.org/10.3390/rs6109412, 2014.

Nilsson, J., Vallelonga, P., Simonsen, S. B., Sørensen, L. S., Forsberg, R., Dahl-Jensen, D., Hirabayashi, M., Goto-Azuma, K., Hvidberg, C. S., Kjaer, H. A., and Satow, K.: Greenland 2012 melt event effects on CryoSat-2 radar altimetry, Geophysical Research Letters, 42, 3919–3926, https://doi.org/10.1002/2015gl063296, 2015.

Otosaka, I. N., Shepherd, A., Casal, T. G. D., Coccia, A., Davidson, M., Di Bella, A., Fettweis, X., Forsberg, R., Helm, V., Hogg, A. E., Hvidegaard, S. M., Lemos, A., Macedo, K., Kuipers Munneke, P., Parrinello, T., Simonsen, S. B., Skourup, H., and Sørensen, L. S.: Surface Melting Drives Fluctuations in Airborne Radar Penetration in West Central Greenland, Geophysical Research Letters, 47, https://doi.org/10.1029/2020gl088293, 2020.

Porter, C., Morin, P., Howat, I., Noh, M.-J., Bates, B., Peterman, K., Keesey, S., Schlenk, M., Gardiner, J., Tomko, K., Willis, M., Kelleher, C., Cloutier, M., Husby, E., Foga, S., Nakamura, H., Platson, M., Wethington Jr., M., Williamson, C., Bauer, G.,

Enos, J., Arnold, G., Kramer, W., Becker, P., Doshi, A., D'Souza, C., Cummens, P., Laurier, F., and Bojesen, M.: ArcticDEM, Harvard Dataverse, https://doi.org/10.7910/DVN/OHHUKH, 2018.

Ridley, J. K. and Partington, K. C.: A model of satellite radar altimeter return from ice sheets, International Journal of Remote Sensing, 9, 601–624, https://doi.org/10.1080/01431168808954881, 1988.

Ronan, A. C., Hawley, R. L., and Chipman, J. W.: Impacts of differing melt regimes on satellite radar waveforms and elevation retrievals, The Cryosphere, 18, 5673–5683, https://doi.org/10.5194/tc-18-5673-2024, 2024.

Rutishauser, A., Scanlan, K. M., Vandecrux, B., Karlsson, N. B., Jullien, N., Ahlstrøm, A. P., Fausto, R. S., and How, P.: Mapping the vertical heterogeneity of Greenland's firn from 2011–2019 using airborne radar and laser altimetry, The Cryosphere, 18, 2455–2472, https://doi.org/10.5194/tc-18-2455-2024, 2024.

Tran, N., Remy, F., Feng, H., and Femenias, P.: Snow Facies Over Ice Sheets Derived From Envisat Active and Passive Observations, IEEE Transactions on Geoscience and Remote Sensing, 46, 3694–3708, https://doi.org/10.1109/tgrs.2008.2000818, 2008.

Vandecrux, B., MacFerrin, M., Machguth, H., Colgan, W. T., van As, D., Heilig, A., Stevens, C. M., Charalampidis, C., Fausto, R. S., Morris, E. M., Mosley-Thompson, E., Koenig, L., Montgomery, L. N., Miège, C., Simonsen, S. B., Ingeman-Nielsen, T., and Box, J. E.: Firn data compilation reveals widespread decrease of firn air content in western Greenland, The Cryosphere, 13, 845–859, https://doi.org/10.5194/tc-13-845-2019, 2019.

Węglarczyk, S.: Kernel density estimation and its application, ITM Web of Conferences, 23, 00037, https://doi.org/10.1051/itmconf/20182300037, 2018.

**Response to Referee 2 on egusphere-2024-3251**

First, we would like to thank the Referee for reviewing and commenting on the manuscript, which will improve the quality of the manuscript. Please find the item-by-item reply below, with the responses in blue. All the suggested changes will be implemented in the revised text that will be uploaded.

**Referee comments**

The study provides valuable insights into firn properties using altimetry data from CryoSat-2 (CS2) and ICESat-2 (IS2), but there are several areas that need clarification and refinement to validate the conclusions. Before any major insight or conclusion can be drawn I find that there are several aspect of the methodology that needs more validation or attention to ensure the accuracy of the results. Except that I find that its a very interesting approach that can yield some good scientific insight into this area.

Below are detailed comments and suggestions to help improve the study with a focus on the main methodology for the altimetry components and firn models.

**General Comments**

**LeW Computation:**

In Figure 1 (related to L87), it is clear that using thresholds at 0.01 and 0.99 may result in unrealistic LeW values unrelated to the volume/surface scattering ratio. How exactly is LeW computed? Is a peak finder algorithm employed? I strongly suggest either smoothing the waveform for better LeW extraction or using the Offset Center of Gravity (OCOG) method to compute the width after identifying the leading edge. Alternatively, the overall OCOG amplitude could serve as the max. The critical objective is to minimize jitter in the LeW estimation. A specific example is pixel C, where the algorithm identifies a maximum beyond the true leading edge, likely near bin 40–45, which aligns with observations for pixel A.

We directly normalised each waveform using the maximum power and searched for the first bin (except for the initial noisy bins) that exceeds 0.01 and the last bin that exceeds 0.99 of the normalised waveform. We agree with the recommendations of the referee, and have improved the method in the revised manuscript. From here onwards, we compute the OCOG amplitude to serve as the maximum power, and following another one of the referee's comments below, we define the beginning and the end of the leading edge using thresholds at 0.05 and 0.95.

In addition, in response to one of the comments below, we adopted Baseline E data, and the results shown in this document will be based on Baseline E instead of Baseline D. Accordingly, the time series of our study have been extended from 2011—2021 to 2010—2024. Everything has been implemented in the revised manuscript.

**DEM (Section 2.3)**

The REMA description should be moved to the beginning of the data description section. Introducing it first provides essential context, as the DEM is referenced throughout both the CS2 and IS2 sections.
This has been implemented in the revised manuscript (now becomes Section 2.1).

**FDM (Section 2.4)**

Given the availability of multiple firn models such as GSFC and GEMB, have you compared their results against the IMAU-FDM model? Previous analyses have shown substantial spatial and temporal differences among these models, which I think is crucial when evaluating penetration depth from laser and radar measurements. At a minimum, a discussion on the potential impact of model differences is necessary to gauge the validity of the results.

Furthermore, models like GSFC and GEMB have been updated to include data through the end of 2024, which presents a valuable opportunity to extend your CS2 and IS2 time series analysis. Incorporating these more recent datasets will enhance the robustness of your study and help provide more insight into how melt events affect the LeW and elevation relationships.
While we agree with the reviewer that comparing the altimetry results with more firn models would strengthen our analysis, we do not do this for the following reasons:

1) We believe that adding more firn models to our analysis has little added benefit. We already make use of two different models, driven by two different RCMs (IMAU-FDM and MAR's firn module). While the models differ in their exact density, temperature and water content, they qualitatively agree in their response to 2012's melt season, the subsequent drier years, and increased melt after 2018. The focus in this paper is on the latter and different firn models would show similar trends. This makes sense, as these trends are largely driven by the climate and not the firn physics. We explicitly refrain from making quantitative statements that would not be supported by a single firn model.

2) A comparison of the performance of different firn models is out of the scope of this work. IMAU-FDM's capabilities have already been compared to other models under idealised non-melt conditions (Lundin et al., 2017), melting conditions (Vandecrux et al., 2020) and runoff capabilities (Machguth et al., under review).

3) Finally, there is a simple practical reason for not including GEMB and GSFC in our analysis: the mean density of the uppermost 1.5 m of firn, which is used extensively in our analysis, is not publicly available online for either model, whereas IMAU-FDM's and MAR's data was already available to the authors. While we could have asked the developers of GEMB and GSFC to also provide us with these data sets, we decided against this given the reasons above and include the potential of GEMB and GSFC in Discussion.

In the revised manuscript (Line 443), we have added:

*"Although not presented in this study, the up-to-date Goddard Space Flight Center (GSFC) firn model (Medley et al., 2022) and Glacier Energy and Mass Balance (GEMB) firn model (Gardner et al., 2023) can also be incorporated in the satellite*

*time series analysis by both qualitatively indicating the presence of subsurface ice lenses and by quantitatively deriving firn properties with radiative transfer models."*

**Resolution (Sections 3.1, 3.2, and 3.3):**

The current 50x50 km binning resolution seems excessively coarse and likely introduces decorrelation, especially for the "dz" variable but also to elevation as you are mixing a lot of different elevations regions. Increasing the spatial resolution would likely improve both spatial and temporal patterns and correlations.
The increased spatial resolution of 10kmx10km has been implemented in the revised manuscript.

**Correlation (Section 4.3):**

The low correlation values (~0.3) are surprising, especially when using a 50% threshold, which should generally yield higher correlations due to its more sensitive to volume change. I remember seeing much larger correlations in both Antarctica and Greenland using the same methodology you have provided. A few factors may contribute to this, including the coarse resolution and the LEPTA slope correction method. LEPTA may inadvertently remove signal by basing its correction on leading-edge range information that varies over time. Testing a more traditional slope correction method, as suggested you explained in Li et al. (2022), would help to better understand this. Additionally, localized analyses are likely to reveal higher correlations, as elevation usually de-correlates a lot more over larger distance while LeW might have larger spatial cohesion.
We appreciate the referee for pointing out the problem. First of all, we realised that the problem lies indeed in our original intuitive LeW estimation, where we directly used 0.01 and 0.99 thresholds to cut the normalised waveform. After following the referee's suggestions to use the OCOG amplitude as the maximum amplitude and using 0.05 and 0.95 thresholds, the correlation between LeW and dh improved to approximately 0.6 (Fig. R4 in blue).

[Figure]

Figure R4. (a)—(b) Probability distribution histogram and, (c)—(d) cumulative distribution function of correlation coefficients when using the improved LeW estimation and different grid resolutions. (a) and (c) include all correlation coefficients. (b) and (d) only use the correlation coefficients with $p$-values not higher than 0.05.

To inspect the effect of different resolutions, we used a 10km x 10km grid to derive the correlation coefficients and analysed the results. In order to provide a more straightforward assessment, we compare the histogram of correlation coefficient using different resolutions, as shown in Fig. R4. It is true that according to the probability distribution, the number of pixels with correlation coefficients higher than 0.75 increased, compared to the 25km x 25km case. In addition, both resolutions result in insignificant ($p$-value > 0.05) correlation coefficients towards the Greenland coastal regions. Therefore, we remove the insignificant values and evaluate the probability and cumulative distribution function (CDF), as shown in Fig. R4b and d. After removing the insignificant values, it is more apparent that the correlation coefficients derived using the 10km x 10km resolution is more concentrated above 0.7, while those derived using the 25km x 25km resolution is only concentrated between 0.5 and 0.7.

Finally, we also inspect whether using the traditional slope method (Bamber, 1994) and the point-based method (Roemer et al., 2007) can further improve the correlation. For this comparison, we show the map of correlation coefficients in Fig. R5, using the 10km x 10km grid determined above. The figure shows that the slope method only results in high correlation between LeW and dh in the Greenland interior with little topography, and the point-based method results in slightly lower (~0.5) correlation coefficients than LEPTA. Our explanation is as follows. The elevation estimates from LEPTA method best represents the elevation of the subsurface, which in turn indicates the volume scattering effects. This can be also

reflected by LeW, which varies due to the variation in volume scattering. The slope method, on the contrary, does include the topography signal that theoretically also has an impact on LeW. However, its implication of the laser-radar height offsets can be compromised, as it introduces the uncertainties caused by the simple assumption that the topography within the radar pulse-limit footprint can be represented by a slope. Similarly, the point-based method may also suffer from the simplication of a fixed footprint size, resulting a slightly less ideal derived dh, as shown in Li et al. (2022).

[Figure]

Figure R5. Comparison of correlation coefficients between LeW and dh derived using (a) LEPTA, (b) slope method (Bamber, 1994; Li et al., 2022), and point-based method (Roemer et al., 2007; Li et al., 2022).

Therefore, we believe that 0.6 is a sufficiently high correlation coefficient between dh and LeW, as it on the one hand indicates that LeW increases simultaneously with the laser-radar height offset, indicating an increased volume scattering. On the other hand, it is also consistent with the observation of Nilsson et al. (2015), where the extreme melt event has a more prolonged effect on LeW than other parameters derived by a satellite radar altimeter.

**Specific Comments**

L54: The Nilsson et al. (2015) study was not limited to NEEM; it covered the entire LRM region, although the time series presented was from NEEM.
(Now L64) This has been changed from
*"Despite the advances in using altimetry to monitor Greenland's firn, the evaluation of firn properties has been limited to either to periods without extensive melt (e.g. January 2013 to January 2019; Scanlan et al., 2023) or small regions (e.g., NEEM site; Nilsson et al., 2015)."*
to
*"Despite these advances in using radar altimetry to monitor Greenland's firn properties—particularly in assessing the effects of melt and refreezing—existing*

*studies have largely been confined to a period without extensive melt (e.g., January 2013 to December 2018; Scanlan et al., 2023), or to the short timeframe immediately following the 2012 melt event (e.g., up to 2014; Nilsson et al., 2015). The impact on the long term, especially following the 2019 melt (Tedesco and Fettweis, 2020), remain insufficiently monitored."*
in the revised manuscript.

L59: Provide a theoretical penetration depth for Ku-band frequencies. For Ku-band over the Greenland Ice Sheet (GrIS), penetration depth is typically 1-2 meters. Additionally, mention that the bias is retracker-dependent.
We have added
*"According to Ridley and Partington (1988) and Davis and Zwally (1993), the penetration depth may range from a few centimetres to several metres, depending on the firn status ((e.g. dry, wet, refrozen; Slater et al., 2019)) and the retracker (Michel et al., 2014; Simonsen and Sørensen, 2017; Li et al., 2022)."*
in Line 33 of the revised manuscript.

L85: Consider updating to Baseline-E, as it includes significant improvements in waveform processing compared to Baseline-D.
Throughout this document, the results are generated using Baseline E data (as mentioned in the comments above). Due to the higher data availability of Baseline E, the time series have been extended as well. These updated results have been presented in the revised manuscript.

L87: The 0.01 threshold for LeW seems too low; most studies use thresholds between 0.05 and 0.15 to account for noise. What is the impact of changing these values to 0.05 and 0.95? A more robust approach would be to compute LeW using OCOG parameters, which are less sensitive to noise.
(Now Line 110 onwards) When we aimed to observe the temporal variation of LeW, changing the 0.01 threshold to 0.05 did not result in essential changes. However, following the general comment and this specific comment of the referee, we have implemented the more conventional and robust approaches in the revised manuscript (using OCOG amplitude as the maximum and using 0.05 and 0.95 thresholds).

L91: The 50% threshold is appropriate for focusing on volume scattering rather than surface scattering. However, the LeW extraction algorithm needs to be redefined or better explained. OCOG-based methods would offer greater robustness.
This has been updated in the revised manuscript (as mentioned in the comments above).

L98: Include a map figure or inset to indicate pixel locations, as their current placement is unclear to the reader.
The pixel locations should be the same one as in Fig. 3 (Fig. 1 in the revised manuscript), which did not appear in L98 yet. However, we realised that the original Fig. 1 did not have added value in the overall readability, hence have removed it from the revised manuscript.

L103: Justify the use of a 50x50 km binning resolution, as it appears excessively coarse. Correlation length analysis could support this choice, or consider aligning the resolution with firn models, which typically have a 10 km resolution. If empty pixels result from a 10 km grid, they can be filled using gentle interpolation.

(Now Line 227) This choice was originally used in Li et al. (2022) to ensure that every inspected pixel should have sufficient (more than 10) data points to compute the reliable statistics, i.e. mean, median and standard deviation. We admit that this was an intuitive choice, therefore we have implemented different pixel sizes (50km x 50km, 25km x 25km, and 10km x 10km) to generate LeW time series, and show the results in Fig. R6. For each sub-plot, y-axis shows the distance along the north-south transect. The overall spatio-temporal patterns of using different resolutions are similar, while the 25km x 25km and 10km x 10km time series indeed show better details.

Finally, due to the higher correlation coefficients between dh and LeW using the 10km x 10km resolution (as assessed above) and the consistency with firn models, we have now adopted the 10km x 10km resolution in the revised manuscript.

[Figure]

Figure R6. Comparison of LeW time series along the north-south transect when different resolutions are adopted.

L117: The DEM resolution (100 m) and search radius (50 m) may not be optimal. Wouldn't this setup yield identical DEM values for adjacent locations? A higherresolution DEM (e.g., 10 m from REMA) would likely provide more accurate results, particularly in areas with complex topography.

We appreciate the suggestion. However, from Li et al. (2022), we found that loading the ArcticDEM with a resolution higher than 100 m was not feasible in MATLAB. One solution would be to crop the DEM, but this was not convenient for the processing chain, therefore we adopted 100 m as the finest resolution in our sensitivity analysis.

**Reference**
Bamber, J. L.: Ice sheet altimeter processing scheme, International Journal of Remote Sensing, 15, 925–938, https://doi.org/10.1080/01431169408954125, 1994.

Li, W., Slobbe, C., and Lhermitte, S.: A leading-edge-based method for correction of slope-induced errors in ice-sheet heights derived from radar altimetry, The Cryosphere, 16, 2225–2243, https://doi.org/10.5194/tc-16-2225-2022, 2022.

Lundin, J. M. D, Stevens, C. M., Arthern, R., Buizert, C., Orsi, A., Ligtenberg, S. R. M., Simonsen, S. B., Cummings, E., Essery, R., Leahy, W., Harris, P., Helsen, M. M., and Waddington, E., D.: Firn Model Intercomparison Experiment (FirnMICE), Journal of Glaciology, 63(239), 401-422, https://doi.org/10.1017/jog.2016.114, 2017.

Machguth, H., Tedstone, A., Kuipers Munneke, P., Brils, M., Noël, B., Clerx, N., Jullien, N., Fettweis, X., and van den Broeke, M.: Runoff from Greenland's firn area – why do MODIS, RCMs and a firn model disagree?, EGUsphere [preprint], https://doi.org/10.5194/egusphere-2024-2750, 2024.

Nilsson, J., Vallelonga, P., Simonsen, S. B., Sørensen, L. S., Forsberg, R., Dahl-Jensen, D., Hirabayashi, M., Goto-Azuma, K., Hvidberg, C. S., Kjaer, H. A., and Satow, K.: Greenland 2012 melt event effects on CryoSat-2 radar altimetry, Geophysical Research Letters, 42, 3919–3926, https://doi.org/10.1002/2015gl063296, 2015.

Roemer, S., Legrésy, B., Horwath, M., and Dietrich, R.: Refined analysis of radar altimetry data applied to the region of the subglacial Lake Vostok/Antarctica, Remote Sens. Environ., 106, 269–284, https://doi.org/10.1016/j.rse.2006.02.026, 2007.

Simonsen, S. B. and Sørensen, L. S.: Implications of changing scattering properties on Greenland ice sheet volume change from Cryosat-2 altimetry, Remote Sensing of Environment, 190, 207–216, https://doi.org/10.1016/j.rse.2016.12.012, 2017.

Slater, T., Shepherd, A., Mcmillan, M., Armitage, T. W. K., Otosaka, I., and Arthern, R., J.: Compensating Changes in the Penetration Depth of Pulse-Limited Radar Altimetry Over the Greenland Ice Sheet, IEEE Transactions on Geoscience and Remote Sensing, vol. 57, no. 12, 9633-9642, https://doi.org/10.1109/TGRS.2019.2928232, 2019.

Tedesco, M. and Fettweis, X.: Unprecedented atmospheric conditions (1948–2019) drive the 2019 exceptional melting season over the Greenland ice sheet, The Cryosphere, 14, 1209–1223, https://doi.org/10.5194/tc-14-1209-2020, 2020.

Vandecrux, B., Mottram, R., Langen, P. L., Fausto, R. S., Olesen, M., Stevens, C. M., Verjans, V., Leeson, A., Ligtenberg, S., Kuipers Munneke, P., Marchenko, S., van Pelt, W., Meyer, C. R., Simonsen, S. B., Heilig, A., Samimi, S., Marshall, S., Machguth, H., MacFerrin, M., Niwano, M., Miller, O., Voss, C. I., and Box, J. E.: The firn meltwater Retention Model Intercomparison Project (RetMIP): evaluation of nine firn models at four weather station sites on the Greenland ice sheet, The Cryosphere, 14, 3785–3810, https://doi.org/10.5194/tc-14-3785-2020, 2020.

---

## Referee Report (RR1)

egusphere-2024-3251

**General Comments**

The manuscript "Assessing spatio-temporal variability of melt-refreeze patterns in firn over Greenland with CryoSat-2" by Li et al. presents an interesting study looking at variations in CryoSat-2 LRM waveform leading edge widths (LeWs) and how they might be used to infer conditions in Greenland Ice Sheet (GrIS) firn. The manuscript begins with an Introduction into the intersection of melting and refreezing with firn and their combined influence on GrIS mass balance. This is followed by an overview of how firn conditions have been assessed previously by way of remote sensing. Section 2 presents the various datasets used throughout the study and Section 3 presents how they authors leverage these data to assess GrIS firn conditions. Section 4 presents the main study results, while Sections 5 and 6 discuss the implications of the results and the present the main study conclusions.

Overall, having reviewed an earlier version of this manuscript, I found this version to be more refined and easier to follow. I want to thank the authors for all their hardwork engaging with and considering previous reviewer comments. I believe the authors successfully managed to integrate a more consistent arc into the manuscript and the scope of the study, bringing together all the different datasets, is commendable. At the same time, there has been substantial change to the manuscript and I have some specific comments on this version that I think need to be addressed before the manuscript can be accepted for publication in The Cryosphere. These will be addressed in the enumerated comments presented below. I do not believe any one of them would take very long to address but their totality would improve the manuscript's coherence, clarity and impact. My hope is that the authors find them constructive as they continue revising their manuscript.

**Specific Comments**

1. My one larger comment is that I would recommend the authors more clearly outline what they view as the central implications/impact of their being able to link CryoSat-2 LeW to firn conditions. In the Introduction (Line 71), the authors state the main objective of their study but there is no definitive statement as to why this type of assessment is important (what is gained from linking LeW to firn conditions? How does this study link back to the mass balance considerations introduced at the start of the Introduction?). On line 417 in Section 5, they seem to suggest their LeW results could play a role in refining firn models but do not provide a path for doing this. Are they implying firn modelers simulate CryoSat-2 LeWs as part of their model validation activities? What would this look like knowing the discretized nature of modeled subsurface structures? I think what I am missing is how the authors see their results contributing to improving the current state-of-the-art. The authors present an interesting and seemingly rich dataset but the discussion about how they see it contributing to the community is underdeveloped. Including more in this direction would strengthen the studies impact and readership in The Cryosphere.

2. On line 7, the authors state LeW is the "most" sensitive parameter to changes in volume scattering. I do not think this is supported by what is presented in the manuscript. It has shown to be strongly sensitive, but confirming its primacy compared to other metrics is not established. I recommend the authors consider adjusting this language.

3. Line 31 (and line 468), please check the citations and attributions are correct as I don't think the Helm et al., 2014b DEM is actually used in the manuscript anymore.

4. I recommend the authors be clearer with what they mean by "… beyond the inflection point …" on line 38 knowing that not everyone who reads the manuscript will be familiar with radar altimetry waveforms or what they look like.

5. I recommend the authors clarify what they mean on line 38 with "… surface area becomes constant." as I am not sure I follow what they mean. Assuming the CryoSat-2 wavefront can be generally thought of as circular once 100's of kilometers from the spacecraft, where the wavefront intersects the surface will continually expand with time.

6. Please provide a reference supporting the statement on line 43 "Depending on thickness …".

7. I recommend the authors provide a version number and/or reference for the ArcticDEM when it is first mentioned on line 74. It comes eventually on line 91 but should appear earlier.

8. I recommend the authors clarify the spacing of the 1500m-3000m elevation groups mentioned on line 89. Are they evenly spaced? Evenly according to elevation or ice sheet area?

9. In Section 2.1, the authors state that they use two version of the ArcticDEM (100m and 1km). This is fine, though I would recommend the authors provide a quick justification for why the two different versions are used. For example, why is the 100m version appropriate for slope correction and 1km for macro-scale roughness and not and vice versa? It seems odd to use two versions without presenting a reason for why.

10. In Section 2.2, I'd also recommend the authors include the level of the LRM data products they are using (e.g., Level 1B, Level 2?). I would also ask the authors consider including a simple diagram showing the basics of LeW calculation and OCOG retracking for those reading the manuscript (e.g., firn modelers) who may be unfamiliar with what the radar waveform would look like. As it stands, the authors expect the reader to be well-versed in some nuanced aspects of radar altimetry (e.g., range bins, waveform shape, OCOG threshold) that may limit the reach of the manuscript in The Cryosphere.

11. I recommend the authors clarify the discrepancy between the number of LeWs extracted from the LRM dataset and the elevation estimates. How are their more LeWs compared to elevation estimates for, what I assume is, the same number of waveforms? Similarly, in Section 2.3, I'd recommend the authors qualify why there are two orders of magnitude less ICESat-2 elevations compared to CryoSat-2. ICESat-2 has not been flying as long as CryoSat-2 but does have a greater along-track data density and multiple beams, so this order of discrepancy is a little suprising.

12. I recommend moving lines 163-165 to the start of the preceding paragraph (i.e., combine with the paragraph starting on line 158).

13. Is there something missing at the end of the sentence starting on line 162 "When MWC>0 …"? Please check.

14. Is there a "the" on line 173 in "For consistency with *the* CryoSat-2 …"? Please check.

15. On line 200, the authors state it is critical that the in situ records they use contain the 2012 melt event. I recommend they state their logic for establishing that the density spikes in the in situ profiles are indeed from 2012 (I assume it is just because these spikes are the largest, but this should still be reflected in the manuscript).

16. I recommend the authors include references to the original implementations of RSR technique on line 212 (e.g., Grima et al., 2012 https://doi.org/10.1016/j.icarus.2012.04.017 and Grima et al., 2014 https://doi.org/10.1016/j.pss.2014.07.018).

17. I would recommend the authors include dates for each of the in situ measurements presented in Figure 1 as this is important context for what is discussed on lines 303 to 306.

18. I would recommend the authors be careful with the use of "validation" on line 217. I do not believe the Scanlan et al. (2023) study went as far as to fully validate their results.

19. Line 240 is where it is most evident, but it also applies to Figures 4, 6, 7, and 8 along with the surrounding discussions. The authors use a few different definitions for what is considered an "anomaly" in the different datasets they consider. On line 240 it is difference from a long-term mean, in Figure 8 it is a difference relative to Winter 2010/2011, but they are all just called "anomalies" and presented without qualification. This is confusing as it isn't always clear if what is being presented/discussed is immediately comparable. I recommend the authors either 1) define all "anomalies" the same way or 2) be specific and explicit with how the data are being represented. For the latter, a part of this could be something like replacing "anomaly" in Figures 6, 7, and 8 with "difference relative to October 2010-April 2011 mean". This would make things much clearer and understandable for the reader.

20. Please clarify if summer months are also removed from the annual means introduced on line 273.

21. I recommend the authors be explicit on line 284 in that the increase is ~0.5m per year (I assume this is what they meant) and not a cumulative increase (which would be trickier to immediately get out of Figure 3a).

22. On line 289, is it possible to provide support (e.g., a reference) for the "… strong melt events between June and December 2018 …" statement similar to the Tedesco and Fettweis (2020) reference for 2019?

23. For Figure 3b, much of the discussion surrounding the results is done by contrasting to 2012. With this in mind, I would recommend the authors consider if there'd be something to gain by referencing LeW changes to 2012 and not 2010. I think this would simplify the interpretation of the LeW timeseries by contrasting all changes relative to the state where they interpret there to be the least amount of volume scattering.

24. In Section 4.2, I'd recommend the authors consider restructuring to align with Figure 4 (or re-structure Figure 4). It is odd to me to discuss the bottom half of the figure (e.g., Figs. 4e and 4g on line 308) before the top. Also, I'd recommend the authors consider splitting Figure 4 between transect and elevation similar to Figures 6 and 7.

25. On line 338, the authors state correlation coefficients do not exceed 0.9 but this seems to be contradicted by what is shown in Figure 5d. Are the authors referring to an average correlation coefficient? Please clarify.

26. I do not completely follow the authors logic on lines 373-375. How would they explain an increase in model density if not caused by melt/refreezing? What other process are they envisioning that would operate in the models with such an annual timescale?

27. Should "annually" in line 376 be "annual"? Please check.

28. The sentence on lines 378-379 is confusing to me and I think it is the multiple uses of "past". I recommend the authors be more specific with the exact years they mean for "the past decade" and "in the past".

29. Is "revolution" on line 387 a typo? Please check.

30. I recommend the authors replace "… ArcticDEM data." on line 397 with "… ArcticDEM mosaics." to make it clear what specific date product they are referring to.

31. I recommend the authors expand a bit what they mean with "… up-to-date …" models on line 444. Are they referring to older versions used in some other uncited study?

---

## Author Response (AR2)

**Response to Referees on egusphere-2024-3251**
We appreciate the referee and the editor for their support in improving our manuscript. Please find our item-by-item responses below, with our replies highlighted in blue. The suggested changes have been implemented in the revised text.

**General Comments**
The manuscript "Assessing spatio-temporal variability of melt-refreeze patterns in firn over Greenland with CryoSat-2" by Li et al. presents an interesting study looking at variations in CryoSat-2 LRM waveform leading edge widths (LeWs) and how they might be used to infer conditions in Greenland Ice Sheet (GrIS) firn. The manuscript begins with an Introduction into the intersection of melting and refreezing with firn and their combined influence on GrIS mass balance. This is followed by an overview of how firn conditions have been assessed previously by way of remote sensing. Section 2 presents the various datasets used throughout the study and Section 3 presents how they authors leverage these data to assess GrIS firn conditions. Section 4 presents the main study results, while Sections 5 and 6 discuss the implications of the results and the present the main study conclusions.

Overall, having reviewed an earlier version of this manuscript, I found this version to be more refined and easier to follow. I want to thank the authors for all their hardwork engaging with and considering previous reviewer comments. I believe the authors successfully managed to integrate a more consistent arc into the manuscript and the scope of the study, bringing together all the different datasets, is commendable. At the same time, there has been substantial change to the manuscript and I have some specific comments on this version that I think need to be addressed before the manuscript can be accepted for publication in The Cryosphere. These will be addressed in the enumerated comments presented below. I do not believe any one of them would take very long to address but their totality would improve the manuscript's coherence, clarity and impact. My hope is that the authors find them constructive as they continue revising their manuscript.

We appreciate the comments of the referee. The following comments are also constructive and are really helpful for improving the manuscript.

**Specific Comments**
1. My one larger comment is that I would recommend the authors more clearly outline what they view as the central implications/impact of their being able to link CryoSat-2 LeW to firn conditions. In the Introduction (Line 71), the authors state the main objective of their study but there is no definitive statement as to why this type of assessment is important (what is gained from linking LeW to firn conditions? How does this study link back to the mass balance considerations introduced at the start of the Introduction?). On line 417 in Section 5, they seem to suggest their LeW results could play a role in refining firn models but do not provide a path for doing this. Are they implying firn modelers simulate CryoSat-2 LeWs as part of their model

validation activities? What would this look like knowing the discretized nature of modeled subsurface structures? I think what I am missing is how the authors see their results contributing to improving the current state- of-the-art. The authors present an interesting and seemingly rich dataset but the discussion about how they see it contributing to the community is underdeveloped. Including more in this direction would strengthen the studies impact and readership in The Cryosphere.

On Line 71, we have added:

*"By assessing how melt--refreezing processes affect the CryoSat-2 LeW, we aim to improve the understanding of the stability of the Greenland Ice Sheet and its response to climate change, and to explore the potential of using radar altimeters as a complementary tool for providing a comprehensive observation of Greenland firn properties."*

Line 417 onwards (now Line 423) has been elaborated as

*"This sensitivity suggests that LeW data could play a crucial role in refining firn models and improving radiative transfer models. For example, currently, radiative transfer modelling has been most successful in understanding firn property variations in Antarctic dry-snow zones (Adodo et al., 2018; Larue et al., 2021). How the refrozen layers in high-elevation zones in Greenland act as a reflective layer and hence affect the radar altimeter signal could be better represented in the modelling. Subsequently, the density and grain size changes following the melt–refreeze events and the potential new-snow deposition could also be derived with the combination of radiative transfer modelling and radar waveform information. Such a method has the potential of improving firn models through data assimilation (Weng, 2007), especially for higher elevations, where existing models may underestimate the impacts of melt events on volume scattering."*

2. On line 7, the authors state LeW is the "most" sensitive parameter to changes in volume scattering. I do not think this is supported by what is presented in the manuscript. It has shown to be strongly sensitive, but confirming its primacy compared to other metrics is not established. I recommend the authors consider adjusting this language.

We have changed the sentence from "...the parameter most sensitive..." to "...a parameter strongly sensitive".

3. Line 31 (and line 468), please check the citations and attributions are correct as I don't think the Helm et al., 2014b DEM is actually used in the manuscript anymore.

The reviewer is correct. We removed the references and attributions to Helm et al. (2014b) from the revised manuscript.

4. I recommend the authors be clearer with what they mean by "... beyond the inflection point ..." on line 38 knowing that not everyone who reads the manuscript will be familiar with radar altimetry waveforms or what they look like.

(Now Line 37) This statement is based on the model of satellite radar altimeter waveforms over ice sheets presented by Ridley and Partington (1988). They noted that *"after the area of the surface illuminated by the pulse becomes constant, surface scattering ceases to increase and this point is marked by an inflection point (or 'cusp') in the altimeter return."* To improve clarity, we have revised the sentence from: *"Surface scattering dominates the start of the waveform, while volume scattering becomes predominant beyond the inflection point, where the illuminated surface area becomes constant."* to *"As explained by Ridley and Partington (1988), surface scattering dominates the start of the waveform, while volume scattering becomes predominant beyond the point at which the illuminated surface area becomes constant (regarding the latter, see (Chelton et al., 2001, Sect. 2.4.1))."*

5. I recommend the authors clarify what they mean on line 38 with "... surface area becomes constant." as I am not sure I follow what they mean. Assuming the CryoSat-2 wavefront can be generally thought of as circular once 100's of kilometers from the spacecraft, where the wavefront intersects the surface will continually expand with time.
While it is true that the radar wavefront expands continuously as it propagates, the effective pulse-limited footprint area does not. For a thorough explanation, we refer to Chelton et al. (2001) section 2.4.1. In short, the pulse-limited footprint area grows linearly with time until the trailing edge of the pulse intersects the surface (assumed planar). Thereafter, the footprint becomes an expanding annulus which area remains constant. Indeed, the radii defining the outer and inner perimeters of the annulus continue to grow.

We have added the reference to Chelton et al. (2001), Sect. 2.4.1.

6. Please provide a reference supporting the statement on line 43 "Depending on thickness ...".
(Now Line 44) We have added references to Nilsson et al. (2015) and Otosaka et al. (2020) to the revised manuscript.

7. I recommend the authors provide a version number and/or reference for the ArcticDEM when it is first mentioned on line 74. It comes eventually on line 91 but should appear earlier.
The reference for the ArcticDEM has been added on Line 74 (now Line 77).

8. I recommend the authors clarify the spacing of the 1500m-3000m elevation groups mentioned on line 89. Are they evenly spaced? Evenly according to elevation or ice sheet area?
Yes, they are evenly spaced according to elevation. We have added the specification on Line 89 (now Line 92).

9. In Section 2.1, the authors state that they use two version of the ArcticDEM (100m and 1km). This is fine, though I would recommend the authors provide a quick justification for why the two different versions are used. For example, why is the 100m version appropriate for slope correction

and 1km for macro-scale roughness and not and vice versa? It seems odd to use two versions without presenting a reason for why.
We have switched to using the 100m ArcticDEM throughout the manuscript.

10. In Section 2.2, I'd also recommend the authors include the level of the LRM data products they are using (e.g., Level 1B, Level 2?). I would also ask the authors consider including a simple diagram showing the basics of LeW calculation and OCOG retracking for those reading the manuscript (e.g., firn modelers) who may be unfamiliar with what the radar waveform would look like. As it stands, the authors expect the reader to be well-versed in some nuanced aspects of radar altimetry (e.g., range bins, waveform shape, OCOG threshold) that may limit the reach of the manuscript in The Cryosphere.
We have added the information "L1b" on Line 111 of the revised manuscript.

Regarding the suggestion by the reviewer to include a simple diagram showing the basics of LeW calculation and OCOG retracking, we have added a new Fig. 1 to the revised manuscript.

11. I recommend the authors clarify the discrepancy between the number of LeWs extracted from the LRM dataset and the elevation estimates. How are their more LeWs compared to elevation estimates for, what I assume is, the same number of waveforms? Similarly, in Section 2.3, I'd recommend the authors qualify why there are two orders of magnitude less ICESat-2 elevations compared to CryoSat-2. ICESat-2 has not been flying as long as CryoSat-2 but does have a greater along-track data density and multiple beams, so this order of discrepancy is a little suprising.
We would like to begin by noting that we have updated the numbers in the revised manuscript. The figures in the previous version inadvertently included LRM data points collected over the oceans.

The higher number of elevation estimates compared to LeW estimates is due to an additional data editing step applied during the LeW determination. Specifically, we exclude all waveforms for which the normalized power in the initial part of the waveform (i.e., beyond range bin $n_1$, which is set to 10 in this study) exceeds 5% of the OCOG amplitude. In such cases, it is not possible to determine $b_{0.05}$ (see Eq. 1), and thus no LeW is derived. We have added this explanation to the revised manuscript (Line 123).

Regarding the difference between the number of CryoSat-2 elevation estimates and ICESat-2 elevations, we would like to clarify that the ICESat-2 point count mentioned in Section 2.3 refers only to those points that are in close proximity—both spatially (within 50 m) and temporally (within the same month)—to the selected CryoSat-2 measurements. For some months, the number of spatially and temporally coincident ICESat-2 observations is relatively low. For example, in January 2019, only about 3% of the CryoSat-2 points have a corresponding ICESat-2 measurement within 50 m (see Fig. R1). To make this clearer, we revised the sentence from (Line 153): *"The search for ICESat-2 points within a 50 m radius of each CryoSat-2*

*measurement yields a total of approximately 4.53 × 10⁵ points."* to *"The search for ICESat-2 points acquired within the same month and within a 50 m radius of each CryoSat-2 measurement yields a total of approximately 4.53 × 10⁵ points."*

[Figure]

Figure R1. Left: horizontal locations of CryoSat-2 (blue) and ICESat-2 (red) points obtained in January 2019 within a zoomed-in region. Right: histogram of distances between each CryoSat-2 point and the nearest ICESat-2 point.

12. I recommend moving lines 163-165 to the start of the preceding paragraph (i.e., combine with the paragraph starting on line 158).
(Now Line 170) Done.

13. Is there something missing at the end of the sentence starting on line 162 "When MWC>0 ..."? Please check.
(Now Line 173) Correct. The sentence has been revised into "When MWC>0, meltwater is present in the firn layer; thus, the altimeter-derived parameters are primarily influenced by meltwater content rather than firn properties."

14. Is there a "the" on line 173 in "For consistency with the CryoSat-2 ..."? Please check.
(Now Line 180) The "the" has been added.

15. On line 200, the authors state it is critical that the in situ records they use contain the 2012 melt event. I recommend they state their logic for establishing that the density spikes in the in situ profiles are indeed from 2012 (I assume it is just because these spikes are the largest, but this should still be reflected in the manuscript).
(Now Line 204) It is documented in the Schaller et al. (2016), the Otosaka et al. (2020) and the MacFerrin et al. (2022) studies that the density spikes are from the 2012 melt event. However, it is also true that we used more recent data, i.e. Vandecrux et al. (2023). In this case, our criterion has changed. Line 203 has been therefore revised as

*"We include available and published in situ density profiles in our analysis if they meet the following criteria: (i) the acquisition site falls within the CryoSat-2 LRM coverage; (ii) the acquisition time is within the CryoSat-2 operational*

*time, and (iii) the acquisition is vertically continuous rather than a single measurement at a specific depth."*

We also added another explanation regarding the density spikes on Line 211:

*"Among the adopted in situ density profiles, the inclusion of the 2012 melt layer is particularly important, as it offers strong evidence that the extreme melt event in that year produced a distinct high-density layer which can also be identified in modelled firn densities (Schaller et al., 2016a; Otosaka et al., 2020; MacFerrin et al., 2022). This layer becomes progressively buried in subsequent years, enabling the observed recovery in LeW. Similarly, recent melt events can appear in the recently acquired in situ density profiles (Vandecrux et al., 2023), as they show similar spikes (approximately 25 % higher than the average density over the top 5 m) as the 2012 melt event."*

16. I recommend the authors include references to the original implementations of RSR technique on line 212 (e.g., Grima et al., 2012 https://doi.org/10.1016/j.icarus.2012.04.017 and Grima et al., 2014 https://doi.org/10.1016/j.pss.2014.07.018).
(Now Line 221) The references have been added.

17. I would recommend the authors include dates for each of the in situ measurements presented in Figure 1 as this is important context for what is discussed on lines 303 to 306.
(Now Fig. 2) The dates of the acquisitions have been included.

18. I would recommend the authors be careful with the use of "validation" on line 217. I do not believe the Scanlan et al. (2023) study went as far as to fully validate their results.
(Now Line 226) Agreed. We have changed "validation" into "assessment".

19. Line 240 is where it is most evident, but it also applies to Figures 4, 6, 7, and 8 along with the surrounding discussions. The authors use a few different definitions for what is considered an "anomaly" in the different datasets they consider. On line 240 it is difference from a long-term mean, in Figure 8 it is a difference relative to Winter 2010/2011, but they are all just called "anomalies" and presented without qualification. This is confusing as it isn't always clear if what is being presented/discussed is immediately comparable. I recommend the authors either 1) define all "anomalies" the same way or 2) be specific and explicit with how the data are being represented. For the latter, a part of this could be something like replacing "anomaly" in Figures 6, 7, and 8 with "difference relative to October 2010-April 2011 mean". This would make things much clearer and understandable for the reader.
Thanks for pointing this out. We have adopted the suggestion (option 2) by the reviewer. Now in the captions and related descriptions of Figs. 8, 9, and 10, we have replaced "anomaly" with "difference relative to October 2010-April 2011 mean".

20. Please clarify if summer months are also removed from the annual means introduced on line 273.

(Now Line 282) Summer months are included. We have added "including summer months" to the revised manuscript.

21. I recommend the authors be explicit on line 284 in that the increase is ~0.5m per year (I assume this is what they meant) and not a cumulative increase (which would be trickier to immediately get out of Figure 3a).

(Now Line 293) We have clarified that the increase is ~0.5m per year.

22. On line 289, is it possible to provide support (e.g., a reference) for the "... strong melt events between June and December 2018 ..." statement similar to the Tedesco and Fettweis (2020) reference for 2019?

(Now Line 299) We have added a reference to Houtz et al. (2021).

23. For Figure 3b, much of the discussion surrounding the results is done by contrasting to 2012. With this in mind, I would recommend the authors consider if there'd be something to gain by referencing LeW changes to 2012 and not 2010. I think this would simplify the interpretation of the LeW timeseries by contrasting all changes relative to the state where they interpret there to be the least amount of volume scattering.

We appreciate the reviewer's thoughtful suggestion. While referencing the 2012 LeW as a baseline could indeed simplify interpretation related to the observed minimum in volume scattering, we chose to reference the 2010 LeW in order to highlight two key aspects: (i) the partial recovery of LeW (and thus the firn layer) following the 2012 melt event, and (ii) the fact that this recovery does not return to pre-2012 levels, as discussed on Line 298. Using 2010 as the reference year thus allows us to present a more comprehensive picture of both the impact and the long-term persistence of the 2012 melt-induced changes. For completeness, we have included the alternative version of the figure as Fig. R2.

[Figure]

Figure R2. Changes in non-melt season
average LeWs for the seasons 2010–2011 and 2013–2023 relative to the
2012 non-melt season average. Years refer to the start of each non-melt
season.

24. In Section 4.2, I'd recommend the authors consider restructuring to align with Figure 4 (or re- structure Figure 4). It is odd to me to discuss the bottom half of the figure (e.g., Figs. 4e and 4g on line 308) before the top. Also, I'd recommend the authors consider splitting Figure 4 between transect and elevation similar to Figures 6 and 7.
(Now Figs. 5 and 6) The orders of the subplots are restructured and the original figures are split.

25. On line 338, the authors state correlation coefficients do not exceed 0.9 but this seems to be contradicted by what is shown in Figure 5d. Are the authors referring to an average correlation coefficient? Please clarify.
(Now Line 352) We have changed it into *"most correlation coefficients remain below 0.9"* as the number of pixels that have correlation coefficients larger than 0.9 is smaller than the numbers of pixels with correlation coefficients between 0.3 and 0.8.

26. I do not completely follow the authors logic on lines 373-375. How would they explain an increase in model density if not caused by melt/refreezing? What other process are they envisioning that would operate in the models with such an annual timescale?
(Now Line 380) This was a wrong observation from our side. The annual cycle of LeW decrease does exist concurrently with the annual cycle of densities, although not as pronounced as the abrupt anomalies due to melt--refreezing. We have removed these statements from the manuscript.

27. Should "annually" in line 376 be "annual"? Please check.
(Now Line 382) This has been corrected.

28. The sentence on lines 378-379 is confusing to me and I think it is the multiple uses of "past". I recommend the authors be more specific with the exact years they mean for "the past decade" and "in the past".
(Now Line 384) This sentence has been revised into

*"Since 2012, the firn density of the upper 1.5m has been notably higher---by up to 50 kg m$^{-3}$---compared to the period prior to 2012.*

29. Is "revolution" on line 387 a typo? Please check.
(Now Line 393) Indeed. Changed into "evolution".

30. I recommend the authors replace "... ArcticDEM data." on line 397 with "... ArcticDEM mosaics." to make it clear what specific date product they are referring to.

(Now Line 403) This has been changed from "ArcticDEM data" into "ArcticDEM mosaics".

31. I recommend the authors expand a bit what they mean with "... up-to-date ..." models on line 444. Are they referring to older versions used in some other uncited study?
(Now Line 456) We intended to highlight that the GSFC-FDM is also available up to year 2023 (similar to the availability of MAR data). However, we agree that the phrase "up-to-date" may have been unclear in this context, so we have removed it from the manuscript to avoid confusion.

**Reference**
Adodo, F. I., Remy, F., and Picard, G.: Seasonal variations of the backscattering coefficient measured by radar altimeters over the Antarctic Ice Sheet, The Cryosphere, 12, 1767–1778, https://doi.org/10.5194/tc-12-1767-2018, 2018.

Chelton, D.B., Ries, J.C., Haines, B.J., Fu, L.L., Callahan, P.S., 2001. Chapter 1 satellite altimetry. In: International Geophysics, https://doi.org/10.1016/S0074-6142(01) 80146-7.

Houtz, D., Mätzler, C., Naderpour, R., Schwank, M., and Steffen, K.: Quantifying Surface Melt and Liquid Water on the Greenland Ice Sheet using L-band Radiometry, Remote Sensing of Environment, 256, 112341, https://doi.org/10.1016/j.rse.2021.112341, 2021.

Larue, F., Picard, G., Aublanc, J., Arnaud, L., Robledano-Perez, A., Meur, E. L., Favier, V., Jourdain, B., Savarino, J., and Thibaut, P.: Radar altimeter waveform simulations in Antarctica with the Snow Microwave Radiative Transfer Model (SMRT), Remote Sensing of Environment, 263, 112534, https://doi.org/10.1016/j.rse.2021.112534, 2021.

Nilsson, J., Vallelonga, P., Simonsen, S. B., Sørensen, L. S., Forsberg, R., Dahl-Jensen, D., Hirabayashi, M., Goto-Azuma, K., Hvidberg, C. S., Kjaer, H. A., and Satow, K.: Greenland 2012 melt event effects on CryoSat-2 radar altimetry, Geophysical Research Letters, 42, 3919–3926, https://doi.org/10.1002/2015gl063296, 2015.

Otosaka, I. N., Shepherd, A., Casal, T. G. D., Coccia, A., Davidson, M., Di Bella, A., Fettweis, X., Forsberg, R., Helm, V., Hogg, A. E., Hvidegaard, S. M., Lemos, A., Macedo, K., Kuipers Munneke, P., Parrinello, T., Simonsen, S. B., Skourup, H., and Sørensen, L. S.: Surface Melting Drives Fluctuations in Airborne Radar Penetration in West Central Greenland, Geophysical Research Letters, 47, https://doi.org/10.1029/2020gl088293, 2020.

Ridley, J. K. and Partington, K. C.: A model of satellite radar altimeter return from ice sheets, International Journal of Remote Sensing, 9, 601–624, https://doi.org/10.1080/01431168808954881, 1988.

Weng, F.: Advances in Radiative Transfer Modeling in Support of Satellite Data Assimilation, Journal of the Atmospheric Sciences, 64, 3799–3807, https://doi.org/10.1175/2007jas2112.1, 2007.